# VIDEOAGENT: ALL-IN-ONE AGENTIC FRAMEWORK FOR VIDEO UNDERSTANDING AND EDITING

## ABSTRACT

Video editing has become essential in digital media creation, yet existing automated systems are restricted to short segment processing and domain-specific tasks. They face two critical limitations: i) inability to handle diverse video comprehension and editing operations, and ii) lack of long-video understanding for coherent narrative creation. We propose **VideoAgent**, an all-in-one agentic framework addressing these challenges through two key innovations. First, we develop automated video shot creation with shot planning agents for coherent narratives and cross-modal retrieval for aligned visual content. Second, we design a multi-agent orchestration framework integrating over thirty specialized editing agents. Intent parsing filters relevant tools while self-reflective graph orchestration assembles complex editing pipelines. Extensive experiments on our newly-proposed **VideoEdit** benchmark and public datasets demonstrate VideoAgent's superiority over existing multimodal LLMs and agentic systems. VideoAgent achieves 87-98% orchestration success rates while reducing API costs by 60%. Human evaluation across six video categories shows VideoAgent produces professional-quality content approaching human-level performance, with ratings only 4% below human-created videos.

## 1 INTRODUCTION

Video editing has become indispensable in modern digital media creation (Esser et al., 2023), as evidenced by the explosive growth of platforms like YouTube and TikTok where millions of creators produce content daily (Chai et al., 2023). However, traditional video editing requires complex professional tools, specialized skills, and intricate multi-step workflows that create significant barriers for ordinary users. This gap between widespread content creation demand and technical accessibility has driven the need for automated video editing systems (Wang et al., 2025) that can interpret natural language instructions and execute sophisticated editing tasks without requiring professional expertise.

Recent advances in AI technology have made it increasingly feasible to leverage Large Language Models (LLMs) (Minaee et al., 2024; Jiang et al., 2024; Guo et al., 2025) and Vision Language Models (VLMs) (Zhang et al., 2024b; Li et al., 2022) for multimodal data understanding and editing. Deep Video Discovery (DVD) employs LLM-coordinated tool ensembles to enable autonomous video exploration through systematic "observe-reason-act" cycles (Zhang et al., 2025). VideoRAG introduces a sophisticated dual-channel architecture combining knowledge graphs with multimodal encoding for complex reasoning over long videos (Ren et al., 2025). mPLUG-2 demonstrates modular multimodal foundations using dual visual encoders for unified text, image, and video understanding (Xu et al., 2023b). Though these works have made pioneering explorations on multimodal content understanding and editing, none of them have adequately solved the complex requirements of automated video editing. These existing approaches suffer from two critical limitations: first, they typically focus on specific understanding or editing tasks and cannot handle the diverse range of video comprehension and editing operations required in real-world scenarios. Second, they often lack the long-video understanding and planning capabilities essential for human-level video editing, preventing them from creating videos with sufficient length and coherent narrative structure.

To address these limitations and achieve human-level automated video creation capabilities, we identify two fundamental challenges that must be overcome:

• **Coherent Long-Form Video Planning**. Professional video creation demands coherent narratives that span multiple shots and extended durations, requiring sophisticated top-down shot planning

with multi-step decomposition and strong text-based reasoning capabilities. For instance, creating a movie trailer requires understanding character arcs, identifying climactic moments, and sequencing emotional beats to build narrative tension, a process that involves complex hierarchical planning from story structure down to individual shot specifications. Critically, such shot planning must be conducted with comprehensive awareness of available visual materials, demanding global understanding and precise retrieval across extensive content libraries. Consider the task of creating a highlight reel from a 2-hour film: the system must possess deep comprehension of both granular visual details and overarching narrative elements to select and sequence appropriate clips. Current approaches predominantly focus on isolated segment understanding and editing, lacking the holistic story creation capabilities essential for professional-quality content. Addressing this challenge is crucial as it represents the fundamental difference between automated clip assembly and genuine creative video production that rivals human expertise.

- **Multi-Agent Workflow Orchestration**. Video editing requires orchestrating diverse tools spanning foundational utilities (audio extraction, rhythm detection) to sophisticated creative agents (dialogue creation, narrative structuring) across visual, auditory, and textual modalities with complex interdependencies. Effective production demands systems that can parse natural language instructions, identify creative patterns, and dynamically compose non-linear workflow networks coordinating multi-modal processing. For instance, creating a music video synchronized to beat drops requires seamlessly integrating audio analysis, rhythm detection, visual retrieval, and temporal alignment in orchestrated sequences where each component's output feeds subsequent operations. Existing methods are constrained to specific domains or predefined workflows, lacking comprehensive tool libraries and adaptive orchestration intelligence for general-purpose creation. This challenge is paramount as it determines whether AI systems can achieve general-purpose video creation or remain limited to domain-specific automated assistance.

To tackle the above challenges, this paper proposes an all-in-one agentic framework VideoAgent for video understanding and editing. Our approach first develops an automated video shot creation system that employs shot planning agents to generate coherent narrative structures, coupled with cross-modal retrieval mechanisms to identify and extract semantically aligned visual content from extensive material libraries, effectively addressing coherent long-form video planning. To achieve

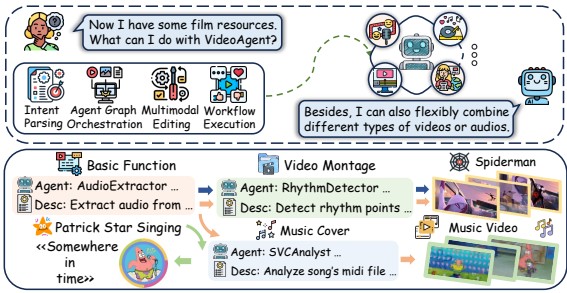

Figure 1: Automated video editing with VideoAgent.

general-purpose video creation across diverse genres, we further design a comprehensive multi-agent orchestration framework that integrates over thirty specialized editing agents through dynamic workflow composition, where intent parsing mechanisms efficiently filter relevant tools and self-reflective graph orchestration adaptively assembles complex editing pipelines, enabling seamless execution of sophisticated creative tasks that span multiple modalities and production workflows.

To summarize, this paper makes the following contributions:

- **All-in-one Agentic Framework**. We propose VideoAgent, a comprehensive agentic framework that unifies video understanding and editing capabilities within a single system, enabling automated video creation across diverse genres and production workflows.

- **Novel Technical Solutions**. We introduce global-aware video shot creation and self-reflective agent graph orchestration to address coherent long-form video planning and multi-agent workflow orchestration challenges that have hindered fully automated video editing and remaking tasks.

- **Comprehensive Evaluation**. We construct the VideoEdit benchmark with high-quality instructions and human-aligned evaluation. Experiments demonstrate VideoAgent's superiority over existing systems, achieving 87-98% success rates while reducing API costs by 60%. Human evaluation shows quality ratings only 4% below professional human-created videos. To facilitate reproducibility, we release our code at https://anonymous.4open.science/r/VideoAgent-DC33.

Figure 2: Automated video shot creation with shot planning, video retrieval and trimming.

## 2 METHODOLOGY

### 2.1 VIDEO EDITING TASK FORMALIZATION

We formally define the video editing task as follows. **Given i)** a natural language instruction $I$ describing the desired video editing objectives, and **ii)** a collection of multimodal content resources $\mathcal{M} = \{m_1, m_2, \ldots, m_{|\mathcal{M}|}\}$ (*e.g.*, raw video clips, audio tracks, and textual documents for narration), the system **produces** a complete edited video $V$ that satisfies the specified instruction $I$. The output video $V$ comprises a temporally ordered sequence of video shots $V = (v_1, v_2, \ldots, v_{|V|})$, where each shot $v_i$ represents a coherent video segment containing synchronized visual and audio components. In essence, the video editing task is to build an agentic system $f$ as follows:

$$f : (I, \mathcal{M}) \rightarrow V = (v_1, v_2, \ldots, v_{|V|}) \tag{1}$$

### 2.2 AUTOMATED VIDEO SHOT CREATION

#### 2.2.1 SHOT PLANNING AGENT

To generate the video shot sequence $(v_1, v_2, \ldots, v_{|V|})$, we develop a shot planning agent that produces shot-level storyboards in natural language. Each storyboard provides a textual description specifying the visual content for the corresponding shot, ensuring all shots collectively form a coherent narrative fulfilling the video creation instruction. This process requires global awareness: the shot planning agent must understand both the overall story structure specified by the user instruction $I$ and the available visual resources $\mathcal{M}$. Given these requirements, the shot planning agent operates as follows:

$$S = (s_1, s_2, \ldots, s_{|V|}) = \text{ShotPlanning}(I, \mathcal{M}, |V|) = \text{LLM}(I, C, |V|; p_s)$$

$$C = \text{LLM}(c_1, c_2, \ldots, c_{|\mathcal{M}|}; p_c), \quad c_i = \text{Caption}(\text{KeyFrames}(m_i)) \tag{2}$$

where $S$ is the generated shot sequence, $C$ is the compressed visual context summarizing all available materials, $c_i$ is the caption for the $i$-th visual material, $p_s$ and $p_c$ are prompts for shot planning and visual context generation respectively, and $|V|$ is the desired number of shots.

The shot planning agent operates through two sequential LLM processing steps. First, it processes all keyframe captions to generate a compressed visual summary $C$ that captures the available video materials. Second, it combines this visual summary with the user instruction $I$ to generate the required number of shot descriptions that form the overall narrative structure.

#### 2.2.2 SHOT VISUAL CONTENT RETRIEVAL

Given the shot-level storyboards $S$, VideoAgent employs paired video indexing and retrieval modules to identify and fetch appropriate video clips for each shot in an efficient manner. The indexing module first extracts keyframes from each video segment using the KeyFrames method mentioned above, then employs cross-modal representation methods such as ImageBind and CLIP to generate embeddings for these keyframes within a unified text-visual representation space. For retrieval, VideoAgent generates embeddings for each shot's storyboard $s_i$ using the same cross-modal representation approach, then selects the visual content with the highest cosine similarity to the storyboard embedding for use in the current shot. Formally, this indexing and retrieval process works as follows:

$$r_i = \arg\max_j \cos(e_j, \text{Embed}(s_i)), \quad e_i = \text{Embed}(\text{KeyFrames}(m_i)), \quad i = 1, 2, \ldots, |V| \tag{3}$$

where $r_i$ represents the retrieved material index for the $i$-th shot, $e_i$ denotes the embedding for the $i$-th material's keyframes, and $\cos(\cdot, \cdot)$ denotes cosine similarity. With this method, our visual content retrieval module fetches semantically aligned visual content to form a coherent narrative following the foregoing text-based shot planning.

Figure 3: Video editing with agent graph orchestration and execution.

### 2.2.3 FINE-GRAINED VIDEO TRIMMING

Though the previous modules fetch semantically aligned visual materials, these raw materials are trimmed with uniform length. In real creative work, shot lengths are variable to match the rhythm of background music and dialogue duration (see Listing 9 for shot length decision agents). To address this issue, VideoAgent further designs a fine-grained video trimming agent that employs Vision Language Models (VLMs) with visual understanding capabilities to select the most suitable sub-segment from the retrieved video clip $v_{r_i}$ based on the current shot's storyboard text, serving as the final visual material for that shot, formally as follows:

$$v_i = m_{r_i}[t_{\text{start}_i} : t_{\text{end}_i}], \quad [t_{\text{start}_i}, t_{\text{end}_i}] = \text{VLM}(m_{r_i}, s_i, t_{i+1} - t_i; p_t) \tag{4}$$

where $v_i$ is the final trimmed video segment for the $i$-th shot, $m_{r_i}$ is the retrieved raw material, $t_{\text{start}_i}$ and $t_{\text{end}_i}$ are the start and end timestamps for trimming, $t_{i+1} - t_i$ represents the target duration for the shot, and $p_t$ is the prompt for video trimming guidance. Through this adaptive trimming design, the system achieves precise temporal alignment between visual content and narrative requirements.

## 2.3 VIDEO EDITING WITH AGENT GRAPH ORCHESTRATION

### 2.3.1 MULTIMODAL EDITING AGENT LIBRARY

To enhance VideoAgent with advanced video editing capabilities beyond simple concatenation, we develop a multi-agent system for sophisticated video editing operations. This system enables complex creative transformations requiring deep understanding of content semantics, artistic styles, and technical production workflows. The system encompasses two primary categories of tool agents:

- **Foundational multimodal processing tools** that provide essential technical operations such as music beat extraction, voice synthesis, face swapping, lip-sync generation, and text translation.

- **Creation prompts with human expertise** that offer guidance for content generation including style-specific scriptwriting, lyrical composition, and meme-based humor creation. For example, the system can transform English stand-up routines into Chinese crosstalk, generate derivative meme content, or synchronize video cuts with background music rhythms.

These diverse tools are essential for enabling flexible and creative video production workflows, and to support this broad spectrum of editing requirements, we develop a comprehensive Agent Library covering a large array of creation genres. Notably, the functionalities used in aforementioned video shot creation (Section 2.2) are also wrapped as tool agents, ensuring all capabilities are **unified within this multi-agent system framework**. Details about all our tool agents are given in Appendix A.5.

### 2.3.2 EDITING AGENT GRAPH

To capture the complexity of editing workflows required by advanced video editing, our VideoAgent proposes to compose the tool agents into an agent graph. In this graph, nodes represent individual agents while edges capture information flow, representing the sequential dependencies of agents processing data. Formally, the tool agent library is denoted as $A = \{A_0, A_1, \ldots, A_n\}$, and the graph-structured agent workflow is represented as $\mathcal{G} = \{A, E, \delta\}$, where $E \subseteq A \times A$ denotes directed edges encoding inter-agent dependencies (*e.g.*, $A_i \to A_j$), and $\delta : E \to \Theta$ defines the state transition function for parameter passing between agents, where $\Theta$ denotes the parameter space.

### 2.3.3 EFFICIENT AGENT SELECTION VIA INTENT PARSING

To achieve efficient agent selection for each creation instruction $I$, VideoAgent employs an intent parsing mechanism to filter and prioritize relevant agents. Given instruction $I$, the system reads the agent registry $R$ containing parameter specifications and functional descriptions, and uses LLMs to decompose both explicit and latent sub-intents within $I$ based on a predefined intent-to-agent mapping table $M$. This process can be defined as:

$$\mathcal{S}_0 = \text{LLM}(I, R, M; p_{iparse}), \quad R = \Psi(A_i \mid A) \tag{5}$$

Here $\mathcal{S}_0$ denotes intents extracted from the user instruction, which correspond to a set of agents usually used to solve these intents. $p_{iparse}$ is the prompt for intent parsing, and $\Psi$ is the registry function that generates JSON schema formatted registration tables for LLM comprehension.

### 2.3.4 SELF-REFLECTIVE AGENT GRAPH ORCHESTRATION AND EXECUTION

To more accurately compose suitable agent graphs for solving user instruction $I$, VideoAgent adopts a self-reflective workflow for agent graph Orchestration. The process begins with the **intent parsing stage**, where it performs iterative self-reflection and revises its intent parsing outputs. This self-reflection process considers whether the extracted intents can form an agent graph sufficient to solve the user instruction. Formally, it works iteratively as follows:

$$\mathcal{S}_{i+1}, F_{i+1} = \text{LLM}(I, \mathcal{S}_i, F_i; p_{iref}) \tag{6}$$

where $\mathcal{S}_i$ represents the intent set at iteration $i$, $F_i$ denotes the feedback from the previous iteration, and $p_{iref}$ is the prompt for iterative reflection. Through this iterative self-reflection, more accurate intent parsing results are obtained for efficient agent pre-selection. The process has a maximum iteration limit, with the LLM controlling whether continued reflection is necessary.

Subsequently, for the **workflow planning** stage, we design a two-step self-evaluation mechanism: the first step assesses whether the current graph requires further modification, while the second step reflects and generates a new iteration of $\mathcal{G}$. This approach enables precise identification of graph deficiencies and addresses potential failure modes, which is defined as follows:

$$r_i = \text{LLM}(F_i', I, R, \mathcal{G}_i; p_{gref}), \quad \mathcal{G}_{i+1}, F_{i+1}' = \text{LLM}(I, \mathcal{S}, F_i', \mathcal{G}_i; p_{gref}') \tag{7}$$

where $r_i$ represents the assessment result at iteration $i$, $F_i'$ denotes the feedback for graph reflection, $R$ is the agent registry, and $p_{gref}, p_{gref}'$ are the prompts for graph assessment and revision respectively.

**Agent Graph Execution**. After constructing the agent graph $\mathcal{G}$, the execution follows the graph-defined workflow through sequential agent invocation. The execution process begins by identifying entry nodes (agents with no incoming edges) and proceeds through topological ordering of the graph structure. Formally, the execution process can be described as:

$$\mathcal{O}_j = A_j(\mathcal{I}_j, \theta_j), \quad \mathcal{I}_j = \bigcup_{(A_i, A_j) \in E} \delta((A_i, A_j))(\mathcal{O}_i) \tag{8}$$

where $A_i, A_j$ represent agents in the graph, $\mathcal{I}_*, \mathcal{O}_*$ refer to input and output data, respectively. $\theta_j$ represents the agent-specific configuration parameters. The state transition function $\delta((A_i, A_j))$ transforms the output from agent $A_i$ into the appropriate input format for agent $A_j$ according to the edge specification. The execution continues until all agents in the graph have been processed and the final video editing outputs are generated from the terminal nodes (agents with no outgoing edges).

## 3 EVALUATION

We conduct extensive experiments to validate VideoAgent's effectiveness across five research questions: **RQ1** evaluates VideoAgent's performance compared to various baselines, **RQ2** examines the effect of key modules, **RQ3** analyzes hyperparameter sensitivity, **RQ4** assesses consistency between self-evaluation and human assessment, and **RQ5** demonstrates superiority in real-world scenarios.

### 3.1 EXPERIMENTAL SETTINGS

**Datasets and Evaluation Protocols**. We evaluate VideoAgent on video understanding and workflow orchestration using two datasets: **Shot2Story** (Han et al., 2023) for video retrieval evaluation, and our

Table 1: Workflow orchestration performance comparison in terms of *Successs Rate*.

| Backbone | Claude-Sonnet-4 | | Claude-Sonnet-3.7 | | GPT-4o | | Deepseek-v3 | |
|---|---|---|---|---|---|---|---|---|
| Data | Audio | Video | Audio | Video | Audio | Video | Audio | Video |
| CoT | 0.78 | 0.77 | 0.80 | 0.72 | 0.68 | 0.65 | 0.69 | 0.65 |
| Debate | 0.66 | 0.52 | 0.67 | 0.66 | 0.76 | 0.69 | 0.61 | 0.50 |
| Step Back | 0.79 | 0.85 | 0.68 | 0.77 | 0.63 | 0.38 | 0.72 | 0.70 |
| ExpertPrompting | 0.71 | 0.76 | 0.78 | 0.73 | 0.69 | 0.68 | 0.75 | 0.70 |
| Intelligent Go-Explore | 0.69 | 0.63 | 0.70 | 0.75 | 0.88 | 0.86 | 0.58 | 0.53 |
| Flow | 0.62 | 0.64 | 0.84 | 0.83 | 0.68 | 0.60 | 0.66 | 0.61 |
| VideoAgent | **0.95** | **0.87** | **0.98** | **0.95** | **0.92** | **0.88** | **0.94** | **0.89** |

new **VideoEdit** benchmark for workflow orchestration with high-quality video creation instructions and human-aligned judge evaluation. See Appendix A.2 for details.

**Baseline Methods**. VideoAgent is compared with a comprehensive list of baseline methods, including **i) Pure-Language Agentic Systems:** Chain-of-Thought (Wei et al., 2022), LLM Debate (Du et al., 2023), Step Back (Zheng et al., 2023), ExpertPromptingXu et al. (2023a), Intelligent Go-ExploreLu et al. (2024), FlowNiu et al. (2025) **ii) Multimodal Understanding LLMs and Agents:** Qwen2.5-VL (Bai et al., 2025), VideoRAG (Ren et al., 2025), VideoMind (Liu et al., 2025) **iii) Video Generation Agents:** NoteBookLM (Google Labs, 2023), Director (VideoDB, 2024), FunClip (ModelScope, 2024), NarratoAI (linyqh, 2025). Details about baselines are provided in Appendix A.3.

**Implementation Details** of our VideoAgent and baseline methods are provided in Appendix A.4.

## 3.2 OVERALL PERFORMANCE COMPARISON (RQ1)

We first compare VideoAgent with baselines on orchestrating effective video editing workflows and video retrieval. The results are shown in Tables 1 and 2, respectively. We observe the following:

- **Superiority in Video Editing Orchestration.** VideoAgent consistently outperforms baselines across all backbone LLMs, achieving success rates of 0.87-0.98 compared to baselines including CoT-based prompting methods and general-purpose agent frameworks. Our method surpasses the best baseline by 2-25% across different modalities and models. This superiority stems from video editing's inherent complexity involving multiple modalities and diverse functionalities, which existing agentic methods struggle to identify and orchestrate. Our flexible agent graph design and self-reflective orchestration method effectively handles these complex multimodal requirements.

- **Improves Backbones in Video Understanding.** VideoAgent significantly enhances video retrieval performance across foundational models. For example, GPT-4o's Recall improves from 31.22% to 48.85%, and Gemini-2.5-flash from 30.04% to 44.93%. These improvements result from our shot planning agent's superior global information perception, generating coherent shot-level storyboards for accurate retrieval. Compared to existing RAG and agent frameworks (VideoRAG, VideoMind-7B), VideoAgent achieves stronger visual-language alignment and overall performance.

- **Cost-Efficiency of VideoAgent.** VideoAgent reduces MLLM API costs by 60% across backbone models while maintaining superior performance. Unlike multimodal LLM baselines that place all candidate shots in context for direct judgment, our strategic shot planning and targeted retrieval minimize redundant processing and focus computational resources on relevant content segments, achieving both better results and significantly lower costs.

Table 2: Video understanding and retrieval performance comparison in terms of Recall@1 (%), Embedding Matching score (EM) (%), Intersection over Union (IoU) (%), and API calling Costs.

| Method | Recall | EM | IoU | Cost | Method | Recall | EM | IoU | Cost |
|---|---|---|---|---|---|---|---|---|---|
| Claude-Sonnet-3.7 | **46.03** | 27.95 | 23.91 | 0.374 | Ours-Claude-Sonnet-3.7 | 44.27 | **28.18** | **24.81** | 0.147 |
| Claude-Sonnet-3.5 | 27.28 | 27.35 | 12.72 | 0.375 | Ours-Claude-Sonnet-3.5 | **38.70** | **28.45** | **21.32** | 0.147 |
| Gemini-2.5-pro | 45.98 | 27.78 | **25.91** | 0.349 | Ours-Gemini-2.5-pro | **47.24** | **28.21** | 25.74 | 0.136 |
| Gemini-2.5-flash | 30.04 | 28.04 | 16.69 | 0.070 | Ours-Gemini-2.5-flash | **44.93** | **28.25** | **25.07** | 0.028 |
| GPT-4o | 31.22 | 27.64 | 18.53 | 0.253 | Ours-GPT-4o | **48.85** | **28.26** | **26.99** | 0.099 |
| VideoRAG | 31.03 | 15.84 | 14.35 | 0.100 | Qwen-2.5-VL-72B-Instruct | 18.89 | 27.99 | 10.51 | - |
| VideoMind-7B | 38.26 | 27.75 | 19.67 | - | | | | | |

Table 3: Performance of different ablated VideoAgent, in terms of Success Rate (SR).

| Backbone | Claude-Sonnet-3.7 | | | | Deepseek-v3 | | | | GPT-4o | | | |
|---|---|---|---|---|---|---|---|---|---|---|---|---|
| Data | Audio | | Video | | Audio | | Video | | Audio | | Video | |
| Metric | SR | Cost | SR | Cost | SR | Cost | SR | Cost | SR | Cost | SR | Cost |
| -I | 0.96 | 0.15 | 0.95 | 0.19 | 0.90 | 0.16 | 0.87 | 0.10 | 0.91 | 0.15 | 0.85 | 0.17 |
| -G | 0.70 | 0.08 | 0.69 | 0.12 | 0.56 | 0.06 | 0.61 | 0.06 | 0.50 | 0.11 | 0.68 | 0.09 |
| -IG | 0.58 | 0.22 | 0.66 | 0.11 | 0.52 | 0.15 | 0.66 | 0.10 | 0.46 | 0.21 | 0.66 | 0.13 |
| Origin | 0.98 | 0.09 | 0.95 | 0.05 | 0.94 | 0.05 | 0.89 | 0.09 | 0.92 | 0.11 | 0.88 | 0.10 |

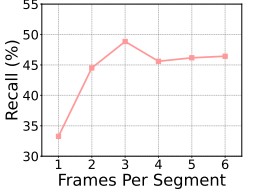 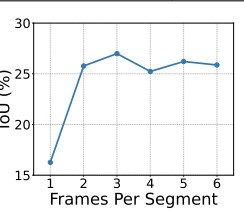 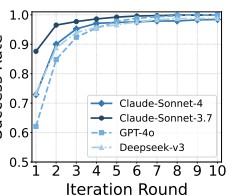 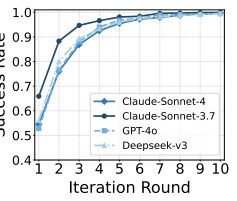

(a) Recall Evaluation     (b) IoU Evaluation     (c) Audio Dataset     (d) Video Dataset

Figure 4: Hyperparameter study for the proposed VideoAgent.

### 3.2.1 HUMAN EVALUATION ON CREATED VIDEO QUALITY

To assess the quality of videos produced by VideoAgent, we conduct a comprehensive human evaluation study. We generate creation instructions covering six diverse video types including tutorials, entertainment, and promotional content, such as "Please create a commentary video about the provided novel. The video should have a voiceover and appropriate visuals that match commentary content." A total of 26 participants evaluate the videos using a 5-point scoring system based on criteria including visual quality, content coherence, and overall effectiveness. We compare VideoAgent against four baseline methods and human-created videos. The results are shown in Figure 5.

VideoAgent successfully creates videos across all six categories, while existing methods often fail on certain video types, producing no evaluable output and demonstrating VideoAgent's superior **generalization capability**. Across all categories, VideoAgent achieves **significantly higher scores than baseline methods**, reflecting its ability to effectively orchestrate diverse editing tools with accurate visual content understanding. Notably, VideoAgent's performance **approaches or even exceeds human-created videos** in several categories, as VideoAgent consistently produces high-quality results while human creators exhibit varying skill levels.

### 3.3 ABLATION STUDY (RQ2)

We validate the effectiveness of key design components in VideoAgent through systematic ablation studies. We first assess our video shot creation component using the Shot2Story dataset (Figure 6). • When the shot planning agent is removed and original Shot2Story queries are used directly for retrieval, performance drops significantly across all metrics, with Recall declining from 48.85 to 45.01. This demonstrates the **importance of global shot planning** even when users provide shot-level descriptions. • Additionally, we replace our cross-modal representation (CM Rep.) method with a caption-then-encoding approach using a pure-language embedding model (all-MiniLM-L6-v2). Results show that our **cross-modal representation paradigm better preserves visual fidelity**, achieving superior retrieval performance at the shot level.

Furthermore, we evaluate the impact of Intent Parsing (-I) and Agent Graph (-G) on workflow orchestration, with results shown in Table 3. • Removing the graph-based agent orchestration substantially reduces the orchestration success rate from over 90% to below 55%, demonstrating the complexity of video editing workflows and the critical **importance of graph-based orchestration**. • While removing intent parsing does not significantly affect performance, it leads to a notable increase in computational cost, highlighting the **cost-efficiency of our intent parsing approach**.

### 3.4 HYPERPARAMETER STUDY (RQ3)

This section studies the impact of two key hyperparameters in our VideoAgent. The evaluation results are shown in Figure 4. From the results we make the following analysis:

**Frames per Segment** in shot planning affects the number of key frames extracted for each raw video resource $m_i$. Using only 1 frame results in a substantial decrease in recall performance, indicating insufficient visual information extraction capability. When the frame count is excessively increased, performance slightly declines, potentially due to over-sampling introducing noisy frames.

**Iteration Round** in workflow orchestration shows that increasing iteration rounds consistently improves the success rate of workflow orchestration, demonstrating typical performance scaling during inference. The results also indicate that, with our graph-based orchestration method, sufficient iterations can significantly reduce performance gaps between different underlying LLM models.

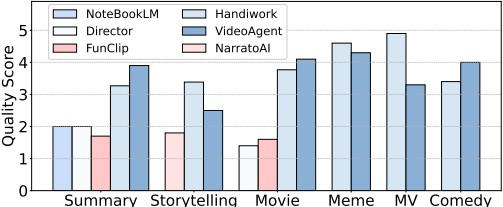

Figure 5: Human-rated video quality assessment.

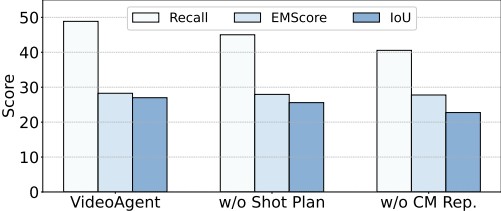

Figure 6: VideoAgent ablation study results.

## 3.5 LLM-HUMAN JUDGMENT CONSISTENCY (RQ4)

To validate the reliability of LLM-based evaluation on orchestration success rate, we compare the consistency between LLM judge results and human annotations. Specifically, we randomly generate 30 agent graphs using VideoAgent for different creation instructions with iteration rounds set to 1, manually annotate their success status, and calculate accuracy, precision, recall, and F1 scores for LLM judgments compared to human ground truth. Results are shown in Figure 7.

Our analysis reveals that Claude-Sonnet-3.7 demonstrates the most reliable evaluation performance, achieving scores between 0.85-1.0 across all metrics, indicating high consistency with human judgment. We observe that recall consistently exceeds precision across model-modality combinations, suggesting that LLM evaluators tend to adopt more permissive evaluation stances with higher rates of false positives than false negatives. This pattern indicates that our evaluation framework provides conservative assessments that rarely miss actual successful orchestrations.

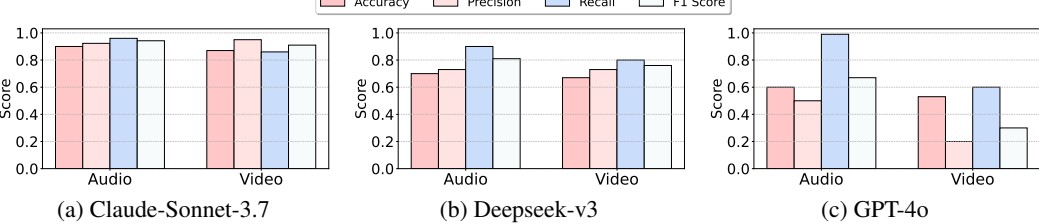

Figure 7: Consistency study on LLM self-evaluation

## 3.6 CASE STUDY ON CREATED VIDEOS (RQ5)

To demonstrate VideoAgent's practical capabilities and workflow flexibility, we conduct a comprehensive real-world case study. As illustrated in Figure 8, when a user provides the instruction about creating a rhythm-synced Spiderman movie montage video, the system autonomously parses the instruction's intent to efficiently orchestrate the appropriate agent graph, subsequently invoking agents such as RhythmDetector that identifies the high-energy, dynamic nature of the content to handle fast-paced visual editing requirements. This showcases VideoAgent's ability to handle creative scenarios through intelligent agent orchestration. Most notably, when addressing complex video montage creation, VideoAgent demonstrates advanced multimodal understanding, for example, synchronizing visual cuts with musical rhythms and retrieving target scenes or characters based on shot text descriptions. The case study particularly highlights VideoAgent's strength in multimodal integration and flexible agent combination. Rather than following rigid predefined workflows, the system dynamically adapts its agent graph structure based on user intents and available resources. This flexibility enables seamless transitions between different content types—from basic audio extraction to sophisticated rhythm-synchronized montages—all within a unified framework. The results

Figure 8: Case study: Creating a rhythm-synced Spiderman movie montage with VideoAgent.

demonstrate VideoAgent's practical utility in real-world creative scenarios, providing users with intuitive yet powerful tools for diverse video editing requirements.

## 4 RELATED WORK

**Automated Multimodal Content Editing** integrates textual, visual, and temporal modalities to enable intelligent content manipulation. Recent advances in diffusion-based models have facilitated text-guided image editing through cross-modal attention mechanisms (Shuai et al., 2024; Fang et al., 2025), while video editing research has addressed temporal consistency via spatiotemporal frameworks incorporating multiple conditional inputs (He et al., 2025; Yang et al., 2025). Multimodal large language models have further enabled systems that parse complex editing instructions and execute corresponding visual operations (Zhang et al., 2024a; Cheng et al., 2025). However, contemporary research primarily emphasizes editing granularity and temporal coherence while overlooking workflow orchestration challenges. Our work addresses comprehensive workflow orchestration by integrating over thirty specialized agents that adaptively assemble editing workflows based on user instructions, enabling seamless execution of complex multi-step tasks without manual coordination.

**Multi-Agent Systems**. Recent advances in multi-agent systems have shifted from monolithic AI models to collaborative frameworks where multiple autonomous agents interact to solve complex tasks (Zhuge et al., 2024a; Tang et al., 2025a). Key developments include the use of LLMs to enable natural language reasoning and tool invocation. This paradigm demonstrates remarkable flexibility through dynamic role specialization, where agents adaptively assume functions such as evaluation and refinement (Zhuge et al., 2024b), program synthesis and execution (Tang et al., 2025b), and seamless interfacing with external APIs, databases, and computational resources (Shi et al., 2024). However, existing multi-agent systems primarily target text-based applications like coding and mathematics. Our work extends multi-agent frameworks to multimodal video editing workflows.

**Video Understanding and Retrieval** are intrinsically connected pillars of visual intelligence, with advances in one domain driving progress in the other. Modern research in video understanding focuses on developing specialized models for challenges like visual question answering (VQA) (Qian et al., 2024; Sima et al., 2024; Li et al., 2025), leveraging complex fusion mechanisms to extract spatial and temporal semantics. Representative works such as VideoMind (Liu et al., 2025) and Qwen-2.5-VL (Bai et al., 2025) demonstrate strong capabilities in long-video scenarios, enabling precise temporal localization and spatio-temporal reasoning. Complementarily, retrieval-oriented methodologies leverage cross-modal video-text models that learn aligned representations in shared embedding spaces. Techniques like VideoRAG (Ren et al., 2025) construct comprehensive knowledge graphs integrating diverse long-video sources. Our work differs by designing a framework with strong video understanding and retrieval capabilities specifically for video editing workflows.

## 5 CONCLUSION

We presented VideoAgent, a comprehensive agentic framework that unifies video understanding and editing capabilities to enable automated video creation across diverse genres and production workflows. Our approach addresses two fundamental challenges in automated video editing: coherent long-form video planning through global-aware shot creation, and multi-agent workflow orchestration via self-reflective graph composition. VideoAgent integrates over thirty specialized editing agents and demonstrates significant superiority over existing methods, achieving 87-98% orchestration success rates while reducing MLLM API costs by 60%. Human evaluation confirms that VideoAgent produces high-quality videos comparable to human creators across multiple categories. These results establish VideoAgent as an effective solution for bridging the gap between widespread content creation demand and technical accessibility, paving the way for more sophisticated automated creative systems.

## ETHIC STATEMENT

VideoAgent democratizes video creation by lowering technical barriers, enabling broader creative expression and content production across diverse user communities. However, we acknowledge that automated video editing capabilities could potentially be misused to create misleading or manipulative content that spreads misinformation. To mitigate these risks, we recommend implementing content moderation frameworks and encourage responsible usage practices among users and platforms. LLMs were used sparingly for language refinement in the preparation of this manuscript.

## REPRODUCIBILITY STATEMENT

To facilitate reproducibility and open scientific research, we have open-sourced VideoAgent at an anonymous link: https://anonymous.4open.science/r/VideoAgent-DC33. The repository includes the complete implementation code, evaluation datasets with corresponding evaluation scripts, and all 30+ specialized tool agents comprising our multimodal editing framework. Beyond this submission, our project has been open-sourced on GitHub and has established a thriving community of over 200 users for deployment and collaboration. The codebase described in this paper has been successfully deployed and utilized by a substantial number of users in real-world scenarios, demonstrating its practical viability and robustness across diverse video editing workflows.

## USAGE OF LLMS

In this work, large language models (LLMs) serve as key components across multiple editing agents to enhance understanding, planning, and execution for adaptive video generation aligned with user goals. Specifically, 1) LLMs compress visual keyframe captions and integrate them with user instructions to generate coherent shot-level storyboards, ensuring global narrative consistency; 2) Vision-language models adaptively select precise video segments guided by storyboard text and target durations, achieving fine-grained temporal alignment with narrative rhythm. Additionally, 3) LLMs parse user instructions to extract intents that facilitate efficient agent selection and prioritization from a registry, while 4) dual-stage self-reflective agent graph orchestration iteratively refines intent understanding and evaluates workflows to improve agent coordination and mitigate failures. Furthermore, LLMs are employed in a limited capacity for language refinement during manuscript preparation.

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

# A APPENDIX

For a deeper understanding of the VideoAgent's operational mechanism, we provide a comprehensive technical description of the multi-modal agentic framework in the appendix.

## A.1 VIDEO SHOWCASE

Below are video showcases we created with VideoAgent, which can be grouped into six main categories: **Beat-synced Edits**, **Video Overviews**, **Storytelling**, **Meme Video Edits**, **Song Remixes**, and **Cross-lingual Adaptations** (including stand-up comedy and crosstalk adaptations).

**Beat-synced Edits**: The user provides background music and film footage. VideoAgent automatically detects beat and tempo changes in the music and aligns high-energy film visual shots with strong beat moments. **Video Overview**: The user provides video footage of a news item or event. VideoAgent transcribes the speech, writes a summary of the event, matches the meaning of each summary sentence to corresponding visual segments from the original video, and generates voiceover for each sentence to complete the edit. **Storytelling**: The user provides plain text of a novel or story along with video footage for visual matching. VideoAgent scripts the story in a specific narration style, matches each line of the script to suitable video segments based on semantic meaning, and generates voiceover for each script segment to produce the final video. **Meme Video Edits**: The user supplies a source video and a custom script or narrative. VideoAgent extracts and transcribes the original audio, generates new speech, precisely syncs the replacement audio to video frames, and outputs a professionally dubbed, meme-style edit. **Song Remixes**: The user provides a MIDI file, lyrics, background music, and a target voice sample. VideoAgent generates a cover, clones the target voice, aligns timing with the arrangement, and integrates the result into the video pipeline for a polished, synced remix. **Cross-lingual Adaptations**: The user provides source audio and target voice samples. VideoAgent adapts the script into the desired cultural format (e.g., Chinese crosstalk or English talk show), synthesizes target voices, applies appropriate effects, and integrates the result into the video editing pipeline for a polished, synced adaptation.

Listing 1: VideoAgent Showcase 1 with Prompts

```
Categories:
1. Beat-synced Edits
   Description:
      - User provides background music and film footage.
      - Detects beat/tempo changes and aligns high-energy shots to strong beats.
↪
   Prompt:
```

```
      Begin with Gwen, who has blond hair, sitting at a dining table in front of
↪  a window, then transition to her playing drums with pop textures and
↪ musical notes in the background. Include action sequences featuring Miguel
↪ O'Hara in a dark blue suit with red accents, sharp red claws, and black-and
↪ -red eye lenses; Spider-Gwen in a white-and-pink suit with a hood and
↪ ballet shoes; Miles Morales with curly hair and a red spider logo on his
↪ chest; and the Spot in a black suit covered in white spots using portal
↪ powers. Focus on the chase scene against a blue sky with trains, and
↪ emphasize high-quality motion throughout--web-swinging, combat, and vibrant
↪  special effects.

2. Video Overviews
   Description:
     - User provides a news/event video.
     - Transcribe speech, summarize, align summary sentences to visuals,
↪ generate voiceover.
   Prompt:
     - Short tech news, colloquial expression within 250 words, check the
↪ accuracy of key terms, e.g. the GPT model name should be 4o instead of 4.0

3. Storytelling
   Description:
     - User provides story/novel text and footage.
     - Script in chosen narration style, match lines to visuals, generate
↪ voiceover.
   Prompt:
     - A verbal interpretation copy of no less than 1,000 words

4. Meme Video Edits
   Description:
     - User supplies a source video and custom script.
     - Extract/transcribe audio, generate new speech, sync precisely to frames
↪ , output dubbed edit.
   Prompt:
     - Create a humorous narrative about two PhD students seeking advice from
↪ Master Ma. For the two PhD students, one of them is known for high citation
↪  counts and the other for numerous publications. Transform martial arts
↪ terms into AI research terminology while keeping phrase lengths similar (
↪ length difference should be less than two Chinese characters). The story
↪ highlights their academic rivalry and ends with Master Ma advising against
↪ "internal competition". Keep signature phrases like "wasn't cautious enough
↪ " and "achieving great results with minimal effort" while avoiding mentions
↪  of real institutions. The word combinations should be logical and
↪ appropriate for an academic context.

5. Song Remixes
   Description:
     - User provides MIDI, lyrics, background music, and a target voice sample.
↪
     - Generate cover, clone voice, align timing, integrate into video
↪ pipeline.
   Prompt:
     - The song is performed by Patrick Star, focusing on the theme of "the
↪ struggles of manuscript submission and dealing with overly critical
↪ reviewers", following the original lyrics' sentence structure while
↪ replacing specific content. It incorporates elements of reviewer nitpicking
↪  (e.g., questioning innovation, demanding redundant experiments) and
↪ expresses frustration with lines like "If only I could swap reviewers, this
```

```
↪    academic fate is too cruel" to highlight the emotional toll of peer review.
↪

6. Cross-lingual Adaptations
   Description:
     - User provides source audio and target voice samples.
     - Adapt script to cultural format (e.g., crosstalk/talk show), synthesize
↪  voices, apply effects, integrate into pipeline.
   Prompt:
     - Adapting to the cultural context of talk shows and localizing humor
```

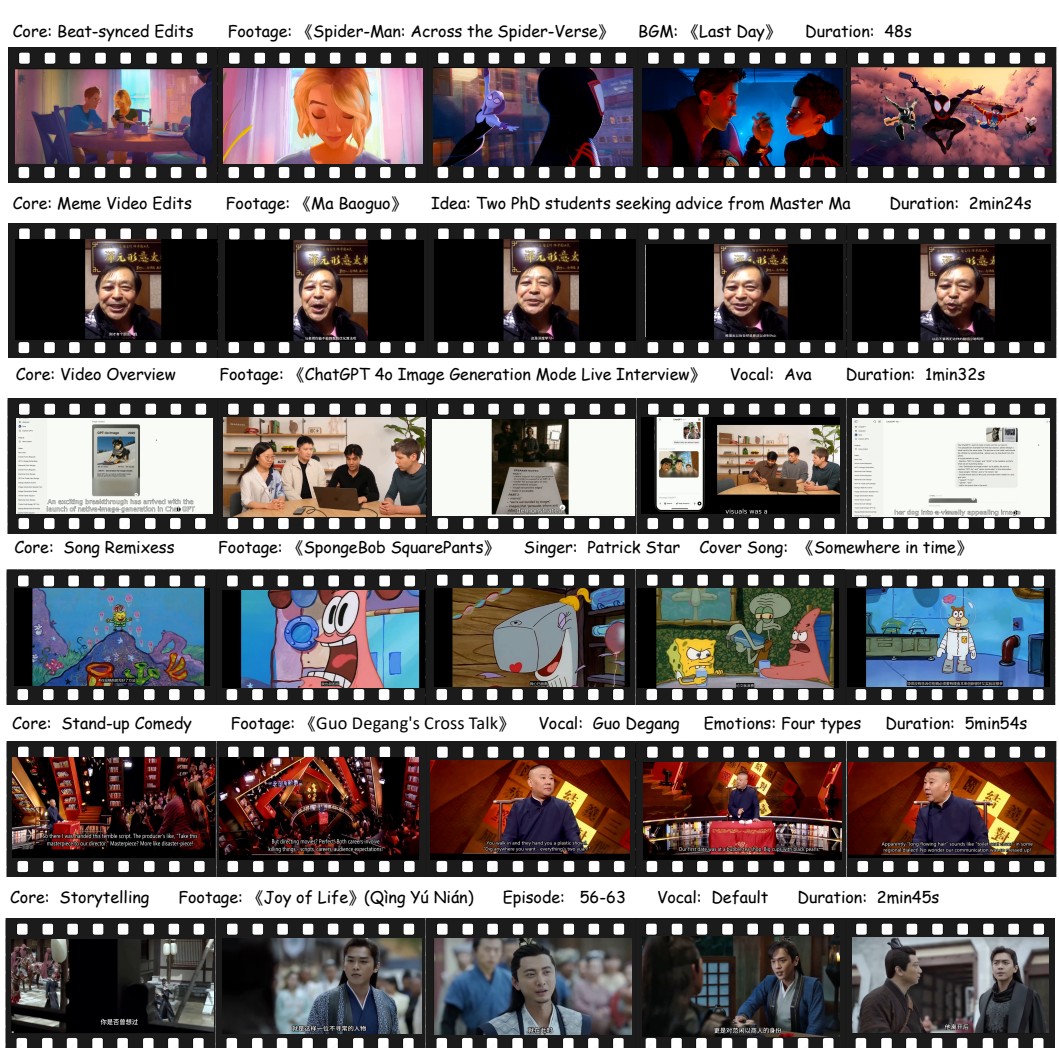

Figure 9: Video cases of VideoAgent in real-world scenarios - Case 1

Listing 2: VideoAgent Showcase 2 with Prompts

```
Categories:
1. Beat-synced Edits
   Description:
     - User provides background music and film footage.
     - Detects beat/tempo changes and aligns high-energy shots to strong beats.
↪
   Prompt:
     - Conflicts between Nezha, Dragon Prince Ao Bing (blue-robed and blue
↪ hair) and Shen Gongbao (black-robed).

2. Video Overviews
   Description:
     - User provides a news/event video.
     - Transcribe speech, summarize, align summary sentences to visuals,
↪ generate voiceover.
   Prompt:
     - Short movie podcast, colloquial expression within 300 words; identify
↪ which actor or host is speaking; do not mention movie-ticket availability.

3. Storytelling
   Description:
     - User provides story/novel text and footage.
     - Script in chosen narration style, match lines to visuals, generate
↪ voiceover.
   Prompt:
     - Write a fluent commentary script of 1,500 words.

4.1 Meme Video Edits (Fan Zhiyi)
   Description:
     - User supplies a source video and custom script.
     - Extract/transcribe audio, generate new speech, sync precisely to frames
↪ , output dubbed edit.
   Prompt:
     - Emphasize the positive impact of IShowSpeed's China tour, which is not
↪ only an online carnival but also a milestone event in cultural
↪ communication in the digital age.

4.2 Meme Video Edits (Xiao Ming Live)
   Description:
     - User supplies a source video and custom script.
     - Extract/transcribe audio, generate new speech, sync precisely to frames
↪ , output dubbed edit.
   Prompt:
     - Based on the following scenario, create an angry rebuttal from Zhuge
↪ Liang (me):
        - Speaker: Zhuge Liang (me)
        - Start with "Why don't you look at your own problems for the failure,"
↪  followed by "...look at your own problems" pattern sentences that all
↪ reference anime events
        - Anime examples must mention specific characters
        - Only the last "...look at your own problems" should return to the
↪ failure scenario
        - Use colloquial language and diverse anime references

5. Cross-lingual Adaptations
   Description:
     - User provides source audio and target voice samples.
```

```
      - Adapt script to cultural format (e.g., crosstalk/talk show), synthesize
↪  voices, apply effects, integrate into pipeline.
    Prompt:
      - To generate a crosstalk-style piece, the story must conform to
↪ objective reality and be about forty to fifty sentences.
```

Core: Beat-synced Edits     Footage: 《Ne Zha (2019)》     BGM: 《Ne Zha》     Duration: 58s

Core: Meme Video Edits     Footage: 《Fan Zhiyi》     Idea: Fan Zhiyi criticizes IShowSpeed's Chengdu trip     Duration: 1min11s

Core: Video Overview     Footage: 《Dune 2 Cast Podcast》     Vocal: Ava     Duration: 1min43s

Core: Meme Video Edits     Footage: 《Xiao Ming Live》     Idea: Angry rebuttal from Zhuge Liang     Duration: 33s

Core: Crosstalk Adaptation     Footage: 《TED Talk Show》     Vocal: Guo Degang/Fu Hang     Duration: 3min16s

Core: Storytelling     Footage: 《Joy of Life》 (Qìng Yú Nián)     Episode: 01-10     Vocal: Default     Duration: 3min24s

Figure 10: Video cases of VideoAgent in real-world scenarios - Case 2

Listing 3: VideoAgent Showcase 3 with Prompts

```
Categories:
1.1 Beat-synced Edits (Titanic)
    Description:
      - User provides background music and film footage.
      - Detects beat/tempo changes and aligns high-energy shots to strong beats.
↪
    Prompt:
      - A romantic and sweet love story about Jack and Rose meeting on the
↪ Titanic. It cannot include the part where the ship is in distress, nor the
↪ night scene. In the first section, Rose, wearing a purple hat and a white
↪ shirt, walks out of a white car with a purple umbrella, looking
↪ thoughtfully.
```

```
1.2 Beat-synced Edits (Interstellar)
   Description:
     - User provides background music and film footage.
     - Detects beat/tempo changes and aligns high-energy shots to strong beats.
↪
   Prompt:
     - Celebrate humanity's courage in space exploration. Include scenes
↪ featuring spaceships, wormholes, black holes, space station docking
↪ maneuvers, ocean planets, and glacial worlds. Show astronauts in their
↪ distinctive white spacesuits as they venture into the unknown, highlighting
↪  humanity's drive to explore the cosmos.

1.3 Beat-synced Edits (Interstellar)
   Description:
     - User provides background music and film footage.
     - Detects beat/tempo changes and aligns high-energy shots to strong beats.
↪
   Prompt:
     - Love can transcend time and space.

1.4 Beat-synced Edits (Ne Zha)
   Description:
     - User provides background music and film footage.
     - Detects beat/tempo changes and aligns high-energy shots to strong beats.
↪
   Prompt:
     - Capture more scenes of conflicts and battles between Nezha and Shen
↪ Gongbao (black-robed) and Dragon Prince Ao Bing (blue-robed).

2.1 Meme Video Edits (Xiao Ming Live)
   Description:
     - User supplies a source video and custom script.
     - Extract/transcribe audio, generate new speech, sync precisely to frames
↪ , output dubbed edit.
   Prompt:
     - Background: Mixue Ice Cream is a national chain brand focusing on ice
↪ cream and tea beverages. On March 15 (Consumer Rights Day), they were
↪ reported to be using overnight lemons. However, compared to other exposures
↪ , using overnight lemons is not considered a particularly serious violation
↪  and is somewhat understandable.
     - Speaker: Snow King (Mixue's representative)
     - Purpose: Emphasize that the "overnight lemon" situation is not too
↪ serious, highlighting Mixue's good reputation
     - Must preserve the phrases "Look in my eyes tell me why" and "tell me"
     - Must end with the word "say it"
     - If the original text contains awkward phrasing, such as redundant words
↪  or confused semantics, do not imitate that style or sentence structure
     - Ensure natural and fluent sentences

2.2 Meme Video Edits (Xiao Ming Live)
   Description:
     - User supplies a source video and custom script.
     - Extract/transcribe audio, generate new speech, sync precisely to frames
↪ , output dubbed edit.
   Prompt:
     - Based on the following scenario, create an angry rebuttal from Zhuge
↪ Liang (me):
         - Speaker: Zhuge Liang (me)
```

```
        - Zhuge Liang (me) is challenged about why a certain Three Kingdoms
↪ character has a higher rating than him and launches a fierce rebuttal
        - Later rating comparisons should show stark differences (can be
↪ exaggerated)
        - Use colloquial language, align with historical facts, and only
↪ replace specific content
```

Core: Beat-synced Edits    Footage: 《Titanic》    BGM: 《My Heart Will Go On》    Duration: 1min35s

Core: Meme Video Edits    Footage: 《Xiao Ming Live》    Idea: Mixue Ice Cream    Duration: 28s

Core: Meme Video Edits    Footage: 《Xiao Ming Live》    Idea: Angry rebuttal from Zhuge Liang    Duration: 45s

Core: Beat-synced Edits    Footage: 《Interstellar》    BGM: 《Memory Reboot》    Duration: 1min01s

Core: Beat-synced Edits    Footage: 《Interstellar》    BGM: 《Cornfield Chase》    Duration: 1min28s

Core: Beat-synced Edits    Footage: 《Ne Zha (2019)》    BGM: 《Once Upon a Time》    Duration: min47s

Figure 11: Video cases of VideoAgent in real-world scenarios - Case 3

## A.2 EVALUATION DATASETS AND PROTOCOLS

To evaluate VideoAgent in terms of both video understanding and workflow orchestration, we conduct experiments from the following two perspectives.

- **Video Understanding and Retrieval Evaluation**. We utilize the **Shot2Story** dataset to evaluate whether VideoAgent can effectively retrieve appropriate video clips according to user inputs. This dataset contains video clips paired with corresponding queries that describe each clip's content. We randomly select 100 videos with more than 5 queries each, using the midpoint frames as keyframes for retrieval evaluation. In total, there are 599 individual queries. Note that achieving such global understanding and precise retrieval across multiple video segments is computationally intensive. Processing all queries for a multimodal LLM or agent typically requires around 3-9 hours. The distribution of evaluation video durations in our sampled dataset is shown in Figure 12. Videos are grouped into fixed intervals 10–15s, 15–20s, 20–25s, 25–30s, and 30s+ based on their total segment lengths. The caption queries per video distribution in our dataset is illustrated in Figure 13. Each video in the dataset contains between 5 and 8 captions, providing multiple query perspectives for evaluation. For video categories distribution in our sampled dataset is shown in Figure 14. The dataset exhibits a diverse range of content categories, with Entertainment, Film & Animation, Autos & Vehicles, Sports , News & Politics, Howto & Style, Science & Technology, Comedy, Education, Travel & Events, People & Blogs, Gaming and Pets & Animals of the sampled videos.

  We use three metrics to compare retrieved video clips and groundtruth shots: **i) Recall:** This is the percentage of groundtruth videos being retrieved with the first rank for each query. **ii) Embedding Matching Score:** Following Shi et al. (2021); Yang et al. (2024); Cheng et al. (2025), we compute the coarse-grained embedding matching score between retrieved video content and LLM-generated captions (see List 6) using ImageBind Girdhar et al. (2023) cross-modal encoder. This evaluates whether retrieved videos align with user intents at a semantic level. **iii) Intersection over Union:** Following Team et al. (2025), we compute temporal IoU between predicted and ground-truth video segments. IoU measures the overlap between retrieved and target clips, ranging from 0 to 1.

- **Workflow Orchestration Evaluation**. To assess the model's performance in orchestrating workflows for video editing instructions, we construct the **VideoEdit** dataset containing 2000 creation instructions encompassing diverse visual and audio editing requirements. Starting with authentic video editing demands, such as creating prompts for ChatGPT-4o image generation live stream overviews like "Short movie podcast, colloquial expression within 300 words, notice to identify which actor or host is talking, don't mention movie tickets available issue", details of prompts can see Appendix A.1. We then use Claude-Sonnet-3.7 for data augmentation to generate additional creative requirements, the audio editing and video editing system prompts are presented in List 4 and List 5. For audio editing, examples include requirements such as "I want to create a funny stand-up comedy audio with audience reactions. The content should be adapted from a script I provide, and I'd like the comedian's voice to match a specific target vocal style." For video editing, examples include "I have a video that I need to edit with a different script while maintaining the original speaker's voice. I want to keep the visuals exactly the same, but change what the person is saying to better match my needs. The new content should sound natural and match the original speaker's voice and speaking style." Through this approach, we yield 2000 distinct instructions covering the full spectrum of video editing tasks.

  We employ Claude-Sonnet-3.7 to build a **Judge Agent** that evaluates whether VideoAgent and baseline models successfully orchestrate agent workflows for these complex requirements. This judge agent verifies instruction fulfillment and validates data flow compatibility across agents, ensuring outputs from preceding agents are properly formatted as inputs for subsequent agents. The judge agent demonstrates high alignment with human evaluators in comparative testing (see Section 3.5). This evaluation uses **Success Rate** as the metric, indicating the percentage of samples where the orchestrated workflow fulfills the user instruction.

Listing 4: Audio Editing Instruction Prompt

```
You are an Autonomous Agent System Designer. I will provide Registered Agents
↪ (Name, Description and Parameters information):

Your task is to:
1. Generate Feasible User Requirement:
```

```
1134      - Only consider generating feasible User Requirement for **audio-related**
1135  ↪ aspects (Finally generate audio instead of video) based on metadata of
1136  ↪ Registered Agents.
1137      - Avoid mentioning specific agent names in User Requirement, but clearly
1138  ↪ describe the needs.
1139      - Avoid mentioning the concrete steps and details of implementation.
1140      - Reflect real-world needs, and be phrased conversationally.
1141  2. Design Executable Agent Graph:
1142      - Format: List
1143      - Agent Graph shall contain metadata for each Agent Node including:
1144        * name: (string)
1145        * inputs: (list of input parameter objects with):
1146          - parameter: input parameter name
1147          - description: brief parameter description
1148        * outputs: (list of output parameter objects with):
1149          - parameter: output parameter name
1150          - description: brief parameter description
1151          - links: (list of dictionaries) where each dictionary specifies:
1152            - key of dictionaries: target agent name
1153            - value of dictionaries: target agent's input parameter name that
1154  ↪ this output connects to
        - Agent Graph case:
          [
            {
              "node": Agent Name,
              "inputs": [
                  {
                  "name": input parameter name1,
                  "description": ...
                  },
                  {
                  "name": input parameter name2,
                  "description": ...
                  },
              ]
              "outputs": [
                  {
                  "name": output parameter name1,
                  "description": ...,
                  "links": [
                      {"next_agent1": next_agent1's input parameter name"},
                      {"next_agent2": next_agent2's input parameter name"}
                      ]
                  },
                  {
                  "name": output parameter name2,
                  "description": ...,
                  "links": []
                  },
                  ...
              ]
            },
            ...
          ]
  3. Generate Agent Chain:
        - Format: List
        - Based on the description of the Agent and the sequential information
  ↪ contained in the designed Agent Graph, generate the Agent Chain
        - Agent Chain case:
```

```
        ["agent1", "agent2", ...]
4. Generate User Input Graph
   - Format: List
   - Parameter nodes with no in-degree (no incoming edges) are uniformly
↪ considered to require user input.
   - Parameter nodes with no in-degree may have different names but share the
↪ same user input, meaning a single user input parameter can point to
↪ multiple such nodes.
   - Parameter nodes with no in-degree that are linked to user input should be
↪  represented in the format **AgentName.input_parameter**
   - Generate the User Input Graph based on the Agent descriptions and
↪ parameter passing information in the designed Agent Graph.
   - User Input Graph case:
      [
        {
          "node": User input parameter name,
          "description": brief description of the parameter
          "links": [
                    {"agent1": agent1.input_parameter1},
                    {"agent2": agent2.input_parameter2}
                ]
        },
        ...
      ]
5. Output Reasoning:
   - Provide concise reasoning (<200 words) explaining the entire workflow
↪ logic

In addition to the above formatting requirements, please also note the
↪ following:
1. For each element of **outputs** in each Agent Node:
   - Ensure that the **links** in the **outputs** point to an input parameter
↪ that actually exists in the next Agent Node.
   - Also, make sure the description of the output parameter matches the
↪ description of the input parameter in the next Agent Node.
2. Final JSON Output Format Specification:
{
    "User Requirement": ...,
    "Agent Graph": ...,
    "Agent Chain": ...,
    "User Input Graph": ...,
    "Reasoning": ...
}
Strictly follow JSON output format!

Metadata of registered agents:
{registry_agents}

Real-life Requirement Examples:
{real-life_examples}
```

Listing 5: Video Editing Instruction Prompt

```
You are an Autonomous Agent System Designer. I will provide Registered Agents
↪ (Name, Description and Parameters information):

Your task is to:
1. Generate Feasible User Requirement:
```

```
1242      - Only consider generating feasible User Requirement for **video-related**
1243  ↪ aspects based on metadata of Registered Agents.
1244      - Avoid mentioning specific agent names in User Requirement, but clearly
1245  ↪ describe the needs.
1246      - Avoid mentioning the concrete steps and details of implementation.
1247      - Reflect real-world needs, and be phrased conversationally.
1248  2. Design Executable Agent Graph:
1249      - Format: List
1250      - Agent Graph shall contain metadata for each Agent Node including:
1251        * name: (string)
1252        * inputs: (list of input parameter objects with):
1253          - parameter: input parameter name
1254          - description: brief parameter description
1255        * outputs: (list of output parameter objects with):
1256          - parameter: output parameter name
1257          - description: brief parameter description
1258          - links: (list of dictionaries) where each dictionary specifies:
1259            - key of dictionaries: target agent name
1260            - value of dictionaries: target agent's input parameter name that
1261  ↪ this output connects to
1262      - Agent Graph case:
1263        [
1264          {
1265            "node": Agent Name,
1266            "inputs": [
1267                {
1268                "name": input parameter name1,
1269                "description": ...
1270                },
1271                {
1272                "name": input parameter name2,
1273                "description": ...
1274                },
1275            ]
1276            "outputs": [
1277                {
1278                "name": output parameter name1,
1279                "description": ...,
1280                "links": [
1281                    {"next_agent1": next_agent1's input parameter name"},
1282                    {"next_agent2": next_agent2's input parameter name"}
1283                    ]
1284                },
1285                {
1286                "name": output parameter name2,
1287                "description": ...,
1288                "links": []
1289                },
1290                ...
1291            ]
1292          },
1293          ...
1294        ]
1295  3. Generate Agent Chain:
      - Format: List
      - Based on the description of the Agent and the sequential information
  ↪ contained in the designed Agent Graph, generate the Agent Chain
      - Agent Chain case:
          ["agent1", "agent2", ...]
```

```
4. Generate User Input Graph
   - Format: List
   - Parameter nodes with no in-degree (no incoming edges) are uniformly
↪ considered to require user input.
   - Parameter nodes with no in-degree may have different names but share the
↪ same user input, meaning a single user input parameter can point to
↪ multiple such nodes.
   - Parameter nodes with no in-degree that are linked to user input should be
↪  represented in the format **AgentName.input_parameter**
   - Generate the User Input Graph based on the Agent descriptions and
↪ parameter passing information in the designed Agent Graph.
   - User Input Graph case:
      [
        {
          "node": User input parameter name,
          "description": brief description of the parameter
          "links": [
                      {"agent1": agent1.input_parameter1},
                      {"agent2": agent2.input_parameter2}
                   ]
        },
        ...
      ]
5. Output Reasoning:
   - Provide concise reasoning (<200 words) explaining the entire workflow
↪ logic

In addition to the above formatting requirements, please also note the
↪ following:
1. For each element of **outputs** in each Agent Node:
   - Ensure that the **links** in the **outputs** point to an input parameter
↪ that actually exists in the next Agent Node.
   - Also, make sure the description of the output parameter matches the
↪ description of the input parameter in the next Agent Node.
2. Final JSON Output Format Specification:
{
    "User Requirement": ...,
    "Agent Graph": ...,
    "Agent Chain": ...,
    "User Input Graph": ...,
    "Reasoning": ...
}
Strictly follow JSON output format!

Metadata of registered agents:
{registry_agents}

Real-life Requirement Examples:
{real-life_examples}
```

Listing 6: Video Caption to High-Level Summary Transformation Prompt

```
prompt = f"""
Transform the following detailed video caption into a concise, high-level
↪ summary that captures the main themes, setting, and overall atmosphere in
↪ no more than 50 words, no point form, use pure sentence.

Focus on:
- Setting identification: Where does the action take place?
```

```
1350   - Key themes: What are the main activities or purposes being shown?
1351   - Atmosphere and tone: What mood or feeling does the scene convey?
1352   - Overall narrative: What story or message emerges?
1353
1354   Instructions:
1355   1. Condense specific details into broader concepts
1356   2. Identify recurring elements that suggest themes
1357   3. Synthesize individual actions into a cohesive narrative about the
1358   ↪ environment
1359   4. Use descriptive language that captures essence rather than listing events
1360   5. Maintain the core message while abstracting from granular details
1361   6. Preserve emotional tone and workplace/environmental culture
1362   7. Keep response under 50 words
1363
1364   Input format: whole_caption"[detailed description]"
1365   Output format: high_level_caption"[concise thematic summary]"
1366
1367   Rules:
1368   - Extract thematic elements from specific actions
1369   - Focus on overarching narrative rather than individual events
1370   - Capture the purpose
1371   - Ensure output maintains fidelity to original content intent
1372
1373   Transform the provided detailed caption following these guidelines."""
```

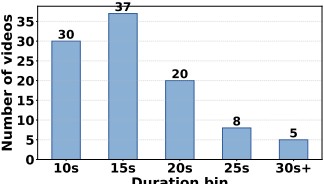

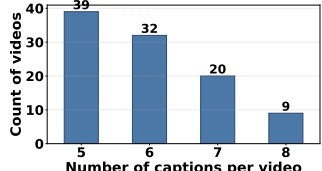

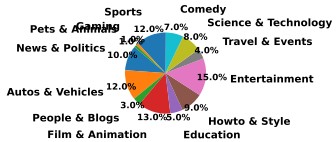

Figure 12: Evaluation video durations distribution.

Figure 13: Evaluation video captions/queries distribution.

Figure 14: Evaluation video categories distribution.

### A.3 BASELINE METHOD

To ensure the comprehensiveness of the benchmark, we conducted a multi-dimensional comparison with n baselines encompassing different mechanisms.
**Agentic Systems on Different Domains.**

- **Chain-of-Thought** Wei et al. (2022):This method proposes a chain of thought to improve LLMs' ability to perform complex reasoning by integrating explicit, structured intermediate steps.

- **LLM Debate** Du et al. (2023): It thoughtfully and systematically integrates debate results from multiple independent language model instances across several iterative rounds of refinement.

- **Step Back** Zheng et al. (2023): This method enables LLMs to abstract from instances containing specific details to generate high-level concepts and foundational first principles efficiently.

- **ExpertPrompting** Xu et al. (2023a): It leverages the potential of LLMs by delegating complex tasks to domain-specific expert agents and automatically synthesizing their respective opinions.

- **Intelligent Go-Explore** Lu et al. (2024): This method is designed to replace handcrafted heuristics with intelligence, iteratively returning to observed states and searching for optimal solutions.

- **Flow** Niu et al. (2025): This framework introduces modular, parallel workflows to boost efficiency and adaptability, enabling real-time updates that preserve global coherence during challenges.

**MLLM and Agents.**

- **GPT-4o** OpenAI (2024): GPT-4o is a unified multimodal model for text, audio, image, and video, with near-human latency, Turbo-level text/code, better non-English, faster, and stronger vision.

- **Deepseek-v3** Liu et al. (2024): This large language model excels in code generation, mathematical reasoning, and multilingual understanding, combining high performance with efficient inference for a wide range of technical and general-purpose applications.

- **Gemini-2.5** Gemini Team (2025): Gemini 2.X spans Pro and Flash models with strong multimodal reasoning; 2.5 Pro offers SoTA coding/reasoning and 3-hour video context, while 2.5/2.0 Flash and Flash-Lite deliver high performance at much lower latency/cost, enabling agentic workflows.

- **Claude-Sonnet-3.7** Anthropic (2025a): This efficient multimodal language model features hybrid reasoning and extended thinking capabilities, optimized for coding, customer service, and general-purpose tasks with strong performance at low latency.

- **Claude-Sonnet-4** Anthropic (2025b): This high-performance language model supports up to 1 million tokens of context, excelling in complex reasoning, code generation, and long-context analysis, making it suitable for large-scale document understanding and planning tasks.

- **Qwen2.5-VL** Bai et al. (2025): This vision-language model is designed for long-video comprehension spanning hours, with precise second-level event localization and multimodal reasoning.

- **VideoRAG** Ren et al. (2025): It introduces a RAG framework for long-context video understanding with graph-grounded and multimodal encoding, enabling unlimited retrieval and beating SOTA.

- **VideoMind** Liu et al. (2025): This method offers a role-based workflow for temporally grounded video understanding, using Chain-of-LoRA for efficient role-switching via lightweight adapters.

**Video Generation Agents.**

- **NoteBookLM** Google Labs (2023): NotebookLM is a personalized AI knowledge assistant that leverages advanced LLMs to analyze and interact with user-provided documents collaboratively.

- **Director** VideoDB (2024): This framework is capable of reasoning through video-related tasks while simultaneously streaming the processed results in real-time to facilitate video synthesis.

- **FunClip** ModelScope (2024): FunClip is an open-source tool for automated video clipping. Users can quickly extract video segments by selecting recognized text or speaker segments.

- **NarratoAI** linyqh (2025): NarratoAI is a video narration platform, combining script writing, editing, voice-over generation, and subtitles into a unified workflow for content production.

## A.4 Implementation Details

This section will provide detailed descriptions of the implementation details and parameter configurations for each baseline in different experiments.

All baseline methods are configured using their default or recommended parameter settings and implemented following the original papers. All experiments were conducted on NVIDIA RTX 3090 GPUs. **Workflow Orchestration Evaluation**: In the workflow orchestration performance comparison and workflow orchestration ablation study, for fairness, we use Claude-Sonnet-3.7 as the Judge Agent to evaluate Agent Graphs and Agent Chains from multiple perspectives, including execution sequence validation, parameter routing correctness, agent functional redundancy assessment, and requirement fulfillment analysis. We also set the number of reflection rounds to 2 for all self-reflective methods. In the LLM-human judgment consistency experiment, to reduce the error rate of human judgment, we first execute agent graph checks for execution sequence validation and parameter routing correctness issues, then proceed with agent functional redundancy assessment and requirement fulfillment analysis. **Video Editing Evaluation**: All video inputs are segmented into three-second segments, under fps1, one frame is sampled per second. For recall calculations, the midpoint timestamp of each retrieved video segments serves as the reference point. Subsequently, two-second clips are extracted, one second before and one second after the midpoint timestamp, yielding a total duration of two seconds per retrieved clip. These extracted clips are then concatenated sequentially to generate the final output video for computing the Embedding Matching score and Intersection over Union metrics, all ours VLM agent/caption use MiniCPM-V 2.6 int4 Yao et al. (2024). In the video editing ablation study and hyperparameter experiments, all our base models utilize GPT-4o, and continue to keep the video segments divided into three seconds for input.

## A.5 TOOLUSE AGENTS

To accommodate diverse user needs, we have designed a variety of multimodal tool-use agents capable of handling tasks such as audio preprocessing, video overview, storytelling, beat-synced edits, meme video remaking, song remixes, cross-lingual adaptations as well as video edits. The agent names and detailed descriptions of these tools are presented in Table 4.

Table 4: The detailed information of tooluse agents.

| Agent Name | Type | Description |
|---|---|---|
| AudioExtractor | Audio Preprocessing | To extract audio from a single video or all videos in a directory |
| LoudnessNormalizer | Audio Preprocessing | Audio loudness normalization tool |
| Merge | Audio Preprocessing | To merge video and audio tracks |
| Mixer | Audio Preprocessing | To mix audio with the BGM |
| Resampler | Audio Preprocessing | Audio Resampling tool |
| Separator | Audio Preprocessing | Audio source separation tool |
| VoiceGenerator | Speech Generation | To generate speech based on scene content. |
| CrossTalkAdapter | Cross-lingual Adaptations | Adapt a reference script into segmented cross talk. |
| CrossTalkSynth | Cross-lingual Adaptations | Segment-by-segment cross talk audio synthesis with final merge. |
| StandUpAdapter | Cross-lingual Adaptations | Adapt a reference script into segmented stand-up comedy. |
| StandUpSynth | Cross-lingual Adaptations | Segment-by-segment stand-up comedy audio synthesis with final merge. |
| SVCAdapter | Song Remixess | Adapt the original lyrics. |
| SVCAnalyst | Song Remixess | Analyze the original song's MIDI file to extract information such as notes, note durations, etc. |
| SVCAnalyst | Song Remixess | Analyze the original song's MIDI file to extract information such as notes, note durations, etc. |
| SVCCoverist | Song Remixess | Source-to-target voice timbre cloning for singing voice synthesis. |
| SVCSingle | Song Remixess | Split the adapted lyrics into segments, then synthesize each segment with the default vocal singing voice. |
| TTSInfer | Meme Video | Take the sliced audio clips as target voice references, then conduct Text-To-Speech synthesis using their rewritten text segments. |
| TTSReplace | Meme Video | Replace audio of the original video with derivative audio segments from the sliced clips. |
| TTSSlicer | Meme Video | Slice the audio into segments. The audio segments often need to be transcribed afterward. |
| TTSWriter | Meme Video | Rewrite the transcript of the sliced audio segments based on user requirements. |
| CommentaryContent Generator | Storytelling | To generates commentary content based on user ideas and text source materials with specialized formatting for video presentations. |
| NewsContentGenerator | Video Overview | To generate news summary based on user ideas and reference materials. |

| Agent Name | Type | Description |
|---|---|---|
| RhythmDetector | Beat-synced Edits | To create cut points for video editing based on music rhythms. |
| RhythmDetector | Beat-synced Edits | To create cut points for video editing based on music rhythms. |
| RhythmContent Generator | Beat-synced Edits | To extract video segment content, create scene-focused narrative summaries, and generate rhythm-aware storyboards. |
| VideoConversion | Video Edits | converts audio content with JSON timestamps into visual scene descriptions for video generation. |
| VideoEditor | Video Edits | Ultimately merging the clips and adding audio. |
| VideoPreloader | Video Edits | To initialize environment and preprocess video files. |
| VideoSearcher | Video Edits | To retrieve matching video clips based on timestamp file. |
| VideoQA | Video Interaction | Transcribes all videos in a directory and provides interactive Q&A session. |
| VideoSummarization Generator | Video Interaction | Agent that generates summarization content based on user ideas and reference materials. |
| FaceSwapping | Video Processing | Replace the faces appearing in the video with those of the target person. |
| LipSynchronization | Video Processing | Synchronize the speaker's lip movements in the video with the audio track. |

### A.6 VISUAL PROCESSING AGENTS

Visual processing agents are specialized components that handle video content analysis, scene caption, and visual element manipulation for multimedia production workflows. These agents bridge the gap between textual content and visual representation by converting visual scenes into actionable narrative descriptions, managing video file operations, and coordinating the integration of visual and audio elements.

#### A.6.1 VIDEOPRELOADER

The VideoPreloader serves as the foundational initialization agent for video processing pipelines, establishing the necessary directory structure and preprocessing operations required for video editing workflows. It integrates with VideoRAG systems to enable efficient video content indexing and retrieval.

Listing 7: VideoPreloader Structure

```
Input: video_directory
Output: preprocessing_status

Algorithm:
1. System initialization:
   a. Configure multiprocessing with spawn method
   b. Set up logging and suppress warnings
   c. Initialize project root and tools directory paths
   d. Add necessary modules to system path

2. Directory structure setup:
```

```
a. Create dataset base directory structure
b. Initialize video_edit working directory
c. Create required subdirectories:
   - audio_analysis: for audio processing outputs
   - scene_output: for scene analysis results
   - videosource-workdir: for VideoRAG operations
   - writing_data: for text and script storage
   - video_output: for final video products
d. Set working directory for VideoRAG operations

3. Video source validation:
   a. Verify video source directory existence
   b. Check for MP4 video files in source directory
   c. Generate list of valid video file paths
   d. Report video discovery statistics

4. VideoRAG module loading:
   a. Dynamically import VideoRAG and QueryParam classes
   b. Handle import errors with fallback mechanisms
   c. Initialize VideoRAG content processing system

5. Video preprocessing pipeline:
   a. Initialize VideoRAG with working directory
   b. Execute video insertion process:
      - Extract video metadata and features
      - Generate content embeddings for retrieval
      - Create searchable video index database
      - Store preprocessed data in working directory
   c. Validate preprocessing completion

6. Status reporting and cleanup:
   a. Generate processing statistics
   b. Report successful video preprocessing
   c. Return execution status for pipeline coordination
   d. Prepare for subsequent VideoSearcher operations

Error Handling:
- Directory creation failures with permission checks
- VideoRAG import failures with module path resolution
- Video file access errors with format validation
- Processing interruption with graceful cleanup
```

### A.6.2 VIDEOSEARCHER

The VideoSearcher retrieves matching video clips from preprocessed video databases based on scene semantic descriptions. It leverages VideoRAG content indexing to perform intelligent video segment matching for automated video assembly workflows.

Listing 8: VideoSearcher Structure

```
Input: video_scene_path
Output: search_status

Algorithm:
1. Path and directory setup:
   a. Initialize dataset and video_edit directory paths
   b. Set scene_output and working directory locations
   c. Create necessary directories if not existing
   d. Add tools directory to system path
```

```
2. VideoRAG module initialization:
   a. Dynamically import VideoRAG and QueryParam classes
   b. Handle import failures with error logging
   c. Configure VideoRAG content processing system

3. Scene file processing:
   a. Load JSON scene file with UTF-8 encoding
   b. Extract segment_scene content for queries
   c. Validate scene content availability
   d. Prepare query string for VideoRAG processing

4. Query parameter configuration:
   a. Initialize QueryParam with videoragcontent mode
   b. Configure reference inclusion settings:
      - wo_reference = False: include video clip references
      - wo_reference = True: exclude references from response
   c. Set multiprocessing spawn method for compatibility

5. VideoRAG content search:
   a. Initialize VideoRAG with working directory
   b. Execute semantic query against video database:
      - Process scene descriptions into embeddings
      - Perform similarity matching against indexed videos
      - Retrieve relevant video segments with timestamps
   c. Generate search response with matching clips

6. Result processing and validation:
   a. Validate VideoRAG query completion
   b. Log search statistics and performance metrics
   c. Handle search failures with appropriate error messages
   d. Return structured search status

Error Handling:
- JSON file not found or invalid format errors
- VideoRAG import and initialization failures
- Empty or malformed scene content handling
- Query processing exceptions with detailed logging
```

### A.6.3   VIDEOEDITOR

The VideoEditor performs final video assembly by integrating matched video segments with synchronized audio tracks. It utilizes multimodal analysis to optimize clip selection and ensures temporal alignment between visual content and audio/storyboards narratives.

Listing 9: VideoEditor Structure

```
Input: video_directory, audio_path, timestamp_path
Output: final_video_path

Algorithm:
1. Initialization and data loading:
   a. Configure directory structure and model paths
   b. Load MiniCPM-V-2_6-int4 model for multimodal analysis
   c. Parse visual_retrieved_segments.json for video segments
   d. Load kv_store_video_segments.json for timing metadata
   e. Initialize beat synchronization and storyboard processing

2. Time period generation:
```

```
   a. Parse rhythm_points.json for beat timestamps
   b. Create time periods from consecutive beat intervals
   c. Load video_scene.json for storyboard descriptions
   d. Align time periods with narrative segments

3. Frame extraction and preprocessing:
   For each video segment:
   a. Extract frames at 1fps starting from segment start time
   b. Include exact start timestamp plus subsequent whole seconds
   c. Convert frames to RGB PIL Images (224x224 resolution)
   d. Handle RGBA to RGB conversion for consistent processing

4. Multimodal scene matching with VLM prompt and frame insertion:
   System context: Video segment analysis for optimal clip selection

   Message structure construction:
   msgs = [{'role': 'user', 'content': [prompt_text]}]
   For each extracted frame:
       msgs[0]['content'].append(frame_image)

   User prompt template embedded with frames:
   "You are analyzing a video segment from {start_time:.3f}s to {end_time:.3f}
↪ s.
   You see {frame_count} consecutive video frames, each representing 1 second.
   The required clip duration is {duration:.3f} seconds.
   Find the best sequence matching this description: '{description}'

   Requirements:
   1. Analyze ALL {frame_count} frames for best consecutive sequence
   2. Choose starting frame (0-{max_frame}) allowing {duration:.3f}s clip
   3. Maximum starting frame: {max_start} to fit duration constraints
   4. Return ONLY single number - starting frame number
   5. Select frames with optimal visual quality and scene consistency
   6. Prioritize alignment with scene description content"

   Frame insertion process:
   - Text prompt inserted as first content element
   - Sequential frame images appended to content array
   - Model processes text prompt with visual frame sequence
   - VLM analyzes prompt context alongside embedded frame data

5. Clip extraction and temporal alignment:
   a. Parse VLM response to extract optimal starting frame
   b. Calculate precise clip timing: start_time + frame_offset
   c. Validate clip boundaries within segment constraints
   d. Extract video clips with exact duration matching beat intervals
   e. Maintain audio track based on keep_original_audio parameter

6. Audio integration pipeline:
   a. Load background audio and resize to match video duration
   b. Configure audio mixing strategy:
      - keep_original_audio=True: composite with background at mix_ratio
      - keep_original_audio=False: replace with background music only
   c. Apply volume adjustments and create composite audio tracks
   d. Synchronize audio timeline with video concatenation

7. Final video assembly and export:
   a. Concatenate video clips using compose method for seamless transitions
   b. Apply composite audio track with proper codec configuration
```

```
    c. Export with optimized encoding parameters:
       - fps=24, codec=libx264, audio_codec=aac
       - preset=medium, threads=4 for performance optimization
    d. Clean up temporary resources and close video handles

Error handling and fallback mechanisms:
- Frame extraction failures with segment boundary validation
- VLM response parsing with regex-based number extraction
- Video file access errors with path verification
- Audio synchronization failures with default timeline alignment
- Memory management with proper resource cleanup
```

### A.6.4 FACESWAPPING

The FaceSwapping agent is capable of replacing a specified face in a video with that of a target person. It uses the Viggle AI tool to achieve precise face replacement, enhancing the overall viewing experience of the video.

Listing 10: FaceSwapping Agent Structure

```
Input: source_video_path, target_face_image
Output: swapped_video_path

Algorithm:
1. Initialization:
   a. Set up file paths and working directories
   b. Initialize access to Viggle AI face swapping tool

2. Source video processing:
   a. Load source video and extract frames
   b. Detect and track the specified face across frames
   c. Prepare frames containing the target face for swapping

3. Face swapping operation:
   For each frame with detected target face:
     a. Send frame and target_face_image to Viggle AI tool
     b. Receive the frame with the specified face replaced by the target face
     c. Replace the original frame with the swapped output

4. Video reconstruction:
   a. Reassemble processed frames preserving original video properties
   b. Synchronize original audio track if available
   c. Output the final face-swapped video file

5. Error handling and cleanup:
   a. Handle failures in calling Viggle AI or frame processing gracefully
   b. Log issues and fall back to original frames if needed
   c. Release resources and temporary files

Notes:
- Input video and target face image are required inputs
- Output is a video with the specified face replaced by the target
- Reliant on Viggle AI for core face swapping capabilities
- Designed for seamless integration within video pipelines
```

### A.6.5 LIPSYNCHRONIZATION

The LipSynchronization agent is capable of synchronizing the speaker's lip movements in a video with a target audio track. It uses the Kling AI tool to achieve precise lip-sync alignment, enhancing the realism and coherence of the video's audio-visual experience.

Listing 11: LipSynchronization Agent Structure

```
Input: source_video_path, target_audio_path
Output: synced_video_path

Algorithm:
1. Initialization:
   a. Configure input/output file paths and working directories
   b. Initialize connection and access to Kling AI lip synchronization tool

2. Data preparation:
   a. Load source video and extract frames and original audio if present
   b. Load target audio track for synchronization

3. Lip synchronization process:
   a. Send source video frames and target audio to Kling AI tool
   b. Receive processed video frames with speaker's lip movements synchronized
↪  to target audio
   c. Replace original frames with synchronized frames

4. Video reconstruction:
   a. Combine synchronized frames preserving original video resolution and
↪ frame rate
   b. Integrate the target audio track as the output video's audio
   c. Output final lip-synced video file

5. Error handling and cleanup:
   a. Handle communication errors with Kling AI tool gracefully
   b. Provide fallback to original video if synchronization fails
   c. Log all processing steps and encountered issues
   d. Release temporary resources and close file streams

Notes:
- Supports video and separate audio inputs
- Output video shows visually consistent lip motions aligned with target audio
- Relies fully on Kling AI external service for synchronization
- Designed for smooth integration with video editing workflows
```

### A.7 AUDIO PROCESSING AGENTS

Audio processing agents constitute a fundamental component of the multimedia production pipeline, handling diverse audio manipulation tasks. These agents leverage advanced signal processing techniques and machine learning models Du et al. (2024); Liao et al. (2024); Liu (2024); Radford et al. (2022); McFee et al. (2015); Liu et al. (2021) to ensure high-quality audio output while maintaining compatibility across different formats and sampling rates. The audio processing workflow encompasses both basic preprocessing operations and sophisticated content analysis capabilities, with agents designed to handle everything from simple format conversions to complex multi-track audio synthesis.

### A.7.1 AUDIOEXTRACTOR

The AudioExtractor agent provides comprehensive audio extraction capabilities from video sources, supporting both single file and batch directory processing. It utilizes FFmpeg for high-quality audio conversion while maintaining consistent output format specifications across all processed files.

Listing 12: AudioExtractor Structure

```
Input: video_path (file or directory)
Output: audio_paths, data_directory

Algorithm:
1. Input validation and path analysis:
   a. Validate video_path existence and accessibility
   b. Determine input type (single file vs directory)
   c. Initialize supported video format extensions:
      {.mp4, .avi, .mov, .mkv, .wmv, .flv, .webm, .m4v}

2. Video file discovery:
   For directory input:
   a. Scan directory for video files with supported extensions
   b. Filter files by extension validation (case-insensitive)
   c. Generate sorted list of video file paths
   d. Validate file accessibility and integrity

3. Audio extraction process:
   For each video file:
   a. Generate output audio path: video_name.wav
   b. Configure FFmpeg command with parameters:
      - Input: source video file
      - Video processing: disabled (-vn flag)
      - Audio codec: PCM 16-bit LE (pcm_s16le)
      - Sample rate: 44.1 kHz (-ar 44100)
      - Channels: stereo (-ac 2)
      - Error handling: error-level logging only

4. FFmpeg execution pipeline:
   Command structure:
   ffmpeg -y -i input_video -vn -acodec pcm_s16le
   -ar 44100 -ac 2 -loglevel error output_audio.wav

   a. Execute subprocess with error handling
   b. Monitor conversion progress and status
   c. Handle CalledProcessError for conversion failures
   d. Detect FFmpeg availability with FileNotFoundError handling

5. Batch processing workflow:
   a. Process each video file sequentially
   b. Track successful vs failed extractions
   c. Generate progress reporting for each file
   d. Collect all successful audio file paths
   e. Report final statistics: success_count/total_count

6. Output generation and validation:
   a. Return audio_paths as list (batch) or string (single)
   b. Provide data_dir as containing directory path
   c. Handle empty results for failed extractions
   d. Ensure output path consistency and accessibility

Error Handling:
```

```
- FFmpeg availability detection and user guidance
- Individual file conversion error isolation
- Directory access permission validation
- Graceful degradation for partial batch failures
- Comprehensive error reporting with specific failure reasons
```

### A.7.2 RHYTHMDETECTOR

The RhythmDetector agent provides comprehensive music rhythm analysis for rhythm-cut video creation workflows. It employs advanced signal processing techniques using librosa and scipy to detect beat patterns, analyze temporal distributions, and generate precise cut points for video editing synchronization with musical rhythms.

Listing 13: RhythmDetector Structure

```
Input: audio_file_path
Output: rhythm_analysis_directory

Algorithm:
1. Audio loading and preprocessing:
   a. Load audio file using librosa.load() with original sample rate
   b. Store audio data, sample rate, and extract base filename
   c. Configure analysis parameters:
      - frame_length=2048, hop_length=512 for STFT analysis
      - Initialize analysis state variables for processing

2. RMS energy calculation and normalization:
   a. Calculate RMS energy using librosa.feature.rms():
      - Apply frame_length and hop_length for temporal resolution
      - Extract energy envelope from audio signal
   b. Normalize RMS values: rms_normalized = rms / max(rms)
   c. Apply smoothing with convolution kernel:
      - kernel = ones(smoothing_window) / smoothing_window
      - Smooth energy curve to reduce noise artifacts

3. Peak detection with temporal constraints:
   Function _detect_rhythm_points():
   a. Apply scipy.signal.find_peaks() with parameters:
      - height=energy_threshold (default 0.4) for minimum peak amplitude
      - distance=min_samples_interval for minimum beat separation
   b. Convert peak indices to timestamps using librosa.frames_to_time()
   c. Apply masking for unwanted detection regions (e.g., intro/outro)

4. Mask-based rhythm filtering:
   a. Define mask_ranges for temporal exclusion zones
   b. Filter detected timestamps within masked periods:
      - Check each timestamp against mask start/end boundaries
      - Separate filtered and masked timestamps for reporting
   c. Generate rhythm points with sequential ID and precise timing

5. Comprehensive visualization generation:
   Function _plot_rhythm_detection():
   a. Create three-panel visualization:
      - Waveform with rhythm point markers and masked regions
      - RMS energy curve with threshold line and detection points
      - Spectrogram with frequency-domain rhythm visualization
   b. Apply color coding: red for rhythm points, gray for masked areas
   c. Export high-resolution plots (300 DPI) for documentation
```

```
6. Rhythm interval analysis and statistics:
   Function _analyze_rhythm_distribution():
   a. Calculate inter-beat intervals using np.diff(timestamps)
   b. Generate statistical measures:
      - mean, median, min, max interval durations
      - standard deviation for rhythm consistency analysis
   c. Create distribution visualizations:
      - Histogram of interval frequencies
      - Temporal plot showing interval variations over time

7. Output generation and data export:
   a. Structure rhythm data in JSON format:
      beat_data: {
         count: total_rhythm_points,
         beats: [{id: index, timestamp: precise_time}]
      }
   b. Include mask information if applied for analysis transparency
   c. Export to dataset/video_edit/audio_analysis/ directory:
      - cut_points.json: rhythm timing data
      - rhythm_detection.png: comprehensive analysis visualization
      - rhythm_distribution.png: statistical analysis plots

Technical Specifications:
- Signal processing: librosa with STFT-based energy analysis
- Peak detection: scipy.signal.find_peaks with configurable thresholds
- Temporal resolution: Frame-based analysis with hop_length=512 samples
- Visualization: matplotlib with multi-panel comprehensive displays
- Precision: 3 decimal places for timestamp accuracy

Rhythm Analysis Features:
- Adaptive energy thresholding for various music styles
- Temporal masking for intro/outro exclusion
- Smoothing filters for noise reduction in energy detection
- Statistical analysis for rhythm consistency evaluation
- Multi-domain visualization (time, frequency, energy)

Error Handling:
- Audio file format validation with librosa compatibility
- Path existence verification for input and output directories
- Graceful handling of insufficient rhythm points for analysis
- Exception management for file I/O operations
- Comprehensive error reporting for debugging workflows
```

### A.7.3 LOUDNESSNORMALIZER

The LoudnessNormalizer agent provides comprehensive audio loudness normalization capabilities for maintaining consistent audio levels across multiple files. It serves as a critical preprocessing component in audio production workflows, utilizing the FAP (Fast Audio Processing) tool for batch loudness standardization.

Listing 14: LoudnessNormalizer Structure

```
Input: data_directory
Output: processing_status

Algorithm:
1. Input validation and path processing:
   a. Validate data_dir parameter using InputSchema
   b. Convert input path to resolved Path object
```

```
   c. Verify directory existence and accessibility
   d. Ensure input is directory (not single file)
   e. Generate absolute path for consistent processing

2. Path preprocessing workflow:
   a. Apply Path.resolve() for absolute path conversion
   b. Validate directory permissions and read access
   c. Handle path existence verification with detailed error reporting
   d. Ensure target directory contains processable audio files

3. FAP command construction:
   a. Build command array: ["fap", "loudness-norm", input_dir,
      output_dir, "--overwrite", "--recursive"]
   b. Configure overwrite mode for existing file replacement
   c. Enable recursive processing for subdirectory traversal
   d. Filter empty arguments from command array

4. Subprocess execution with real-time monitoring:
   a. Initialize subprocess.Popen with pipe configuration:
      - stdout=PIPE, stderr=PIPE for output capture
      - bufsize=1 for line-buffered output
   b. Create threading.Thread for stdout and stderr monitoring
   c. Apply real-time output reading with UTF-8 decoding
   d. Handle decoding errors with 'replace' error handling

5. Real-time output processing:
   Function _read_output(pipe):
   a. Iterate through pipe.readline() until empty
   b. Decode bytes to UTF-8 with error replacement
   c. Strip whitespace and format with [FAP] prefix
   d. Print real-time progress updates to console
   e. Ensure proper pipe closure after processing

6. Process completion and result handling:
   a. Wait for subprocess completion with process.wait()
   b. Join stdout and stderr threads for cleanup
   c. Evaluate return_code for success/failure determination
   d. Generate success status for return_code == 0
   e. Raise RuntimeError for non-zero return codes

Error Handling:
- Path existence and type validation with specific error messages
- Directory permission verification and access control
- FAP tool availability detection with installation guidance
- Real-time error capture and user feedback
- Graceful subprocess termination and resource cleanup
- Comprehensive exception handling with status reporting
```

### A.7.4  MERGE

The Merge agent provides direct video and audio track combination capabilities without intermediate video clip processing. It distinguishes itself from other multimedia agents by performing complete file merging rather than segmented video editing, utilizing FFmpeg for high-quality encoding and synchronization.

Listing 15: Merge Structure

```
Input: video_path, audio_path
Output: merged_video_path
```

```
Algorithm:
1. Input validation and parameter processing:
   a. Validate video_path and audio_path using InputSchema
   b. Verify file existence and accessibility for both inputs
   c. Set output_path to overwrite original video file
   d. Ensure input files are in compatible formats

2. FFmpeg command construction:
   a. Configure dual input streams:
      - Primary input: source video file (-i video_path)
      - Secondary input: replacement audio file (-i audio_path)
   b. Set video encoding parameters:
      - Codec: libx264 for H.264 compression
      - Preset: fast for speed/quality balance
      - CRF: 23 for optimal quality (18-28 range)
   c. Set audio encoding parameters:
      - Codec: aac for broad compatibility
      - Bitrate: 192k for high-quality audio

3. Stream mapping configuration:
   a. Video stream selection: -map 0:v:0 (first video from input 0)
   b. Audio stream selection: -map 1:a:0 (first audio from input 1)
   c. Duration control: -shortest flag for sync with shorter input
   d. Optimization: +faststart movflags for streaming compatibility

4. Encoding execution pipeline:
   Command structure:
   ffmpeg -i video_path -i audio_path -c:v libx264 -preset fast
   -crf 23 -c:a aac -b:a 192k -map 0:v:0 -map 1:a:0
   -shortest -movflags +faststart -y output_path

   a. Execute subprocess.run with check=True for error detection
   b. Monitor process completion and return code validation
   c. Handle real-time progress feedback through console output

5. Quality assurance and output validation:
   a. Verify successful merge completion through return code
   b. Validate output file creation and size consistency
   c. Ensure audio-video synchronization maintenance
   d. Generate success confirmation with output path

6. Error handling and recovery:
   a. Capture CalledProcessError for FFmpeg execution failures
   b. Handle file permission and access errors gracefully
   c. Provide detailed error reporting for troubleshooting
   d. Ensure partial file cleanup on processing failure

Technical Specifications:
- Video encoding: H.264 with CRF 23 quality level
- Audio encoding: AAC at 192 kbps bitrate
- Synchronization: Shortest input duration matching
- Output optimization: Fast-start flag for progressive download
- File handling: Automatic overwrite with -y flag
```

### A.7.5 MIXER

The Mixer agent provides sophisticated audio mixing capabilities for combining foreground audio with background music tracks. It utilizes PyDub for comprehensive audio processing, supporting dynamic volume adjustment, length synchronization, and high-quality audio overlay operations.

Listing 16: Mixer Structure

```
Input: bgm_path, audio_path
Output: mixed_audio_path

Algorithm:
1. Input validation and path processing:
   a. Validate bgm_path and audio_path using InputSchema
   b. Verify file existence and format compatibility
   c. Generate output filename: "mixed_" + original_audio_name
   d. Set output directory to match input audio directory
   e. Determine output format from file extension or default to WAV

2. Audio loading and preprocessing:
   a. Load BGM file using AudioSegment.from_file(bgm_path)
   b. Load vocal audio using AudioSegment.from_file(audio_path)
   c. Support multiple audio formats (WAV, MP3, FLAC, etc.)
   d. Handle audio format conversion internally through PyDub

3. Volume adjustment and normalization:
   a. Apply BGM volume reduction: bgm_audio + bgm_volume
   b. Default BGM volume adjustment: -1 dB for balanced mixing
   c. Maintain vocal audio at original volume level
   d. Report volume adjustment settings to user

4. Length synchronization and BGM extension:
   a. Compare audio lengths: len(bgm_audio) vs len(vocal_audio)
   b. For shorter BGM:
      - Initialize repeated_bgm = bgm_audio
      - Loop BGM extension: repeated_bgm += bgm_audio
      - Continue until repeated_bgm >= vocal_audio length
      - Trim to exact vocal length: repeated_bgm[:len(vocal_audio)]
   c. For longer BGM: use original BGM without modification

5. Audio overlay and mixing process:
   a. For extended BGM case:
      - Apply overlay: vocal_audio.overlay(repeated_bgm)
      - Vocal audio maintains prominence in mix
   b. For original BGM case:
      - Apply overlay: bgm_audio.overlay(vocal_audio)
      - BGM provides extended background beyond vocal length
   c. Overlay operation preserves audio quality and dynamic range

6. Export and output generation:
   a. Determine output format from file extension analysis
   b. Export mixed audio: mixed_audio.export(output_path, format)
   c. Support format options: WAV, MP3, FLAC, AAC, OGG
   d. Maintain original audio quality settings during export
   e. Return output path for downstream processing

Technical Specifications:
- Volume adjustment: -1 dB default BGM reduction
- Length handling: Automatic BGM looping for shorter backgrounds
- Overlay method: Vocal-priority mixing with background preservation
```

```
- Format support: Multi-format input/output through PyDub
- Quality preservation: No quality degradation during mixing process

Error Handling:
- File format compatibility validation
- Audio loading error detection and reporting
- Export format verification with fallback to WAV
- Exception handling with detailed error messaging
- Graceful degradation for unsupported audio formats
```

### A.7.6   RESAMPLER

The Resampler agent provides comprehensive audio resampling capabilities for standardizing sample rates across multiple audio files within a directory. It leverages the FAP (Fast Audio Processing) framework to perform batch resampling operations with real-time progress monitoring and threaded output handling.

Listing 17: Resampler Structure

```
Input: data_directory
Output: processing_status

Algorithm:
1. Input validation and path processing:
   a. Validate data_dir parameter using InputSchema
   b. Convert input path to resolved Path object using Path.resolve()
   c. Verify directory existence with detailed error reporting
   d. Ensure input is directory type (not single file)
   e. Generate absolute path for consistent processing

2. Path preprocessing workflow:
   Function _process_path(input_path):
   a. Initialize Path object from input string
   b. Validate path.exists() with FileNotFoundError handling
   c. Verify path.is_dir() with type validation
   d. Return resolved absolute path for processing
   e. Handle permission and access control validation

3. FAP command construction and configuration:
   a. Build command array: ["fap", "resample", input_dir,
      output_dir, "--overwrite"]
   b. Configure input and output directories (same directory)
   c. Enable overwrite mode for in-place file replacement
   d. Filter empty arguments from command array
   e. Generate command string for execution logging

4. Subprocess execution with threading:
   a. Initialize subprocess.Popen with pipe configuration:
      - stdout=PIPE, stderr=PIPE for output capture
      - bufsize=1 for line-buffered real-time output
   b. Create separate threads for stdout and stderr monitoring
   c. Start threads for concurrent output processing
   d. Execute process.wait() for completion synchronization

5. Real-time output monitoring system:
   Function _read_output(pipe):
   a. Implement iter(pipe.readline, b'') for line iteration
   b. Decode bytes to UTF-8 with 'replace' error handling
   c. Strip whitespace and apply [FAP] prefix formatting
```

```
     d. Print real-time progress updates to console
     e. Ensure proper pipe closure in finally block

6. Process completion and result validation:
     a. Wait for subprocess completion with return_code capture
     b. Join stdout_thread and stderr_thread for cleanup
     c. Evaluate return_code for success/failure determination
     d. Return success status for return_code == 0
     e. Raise RuntimeError with detailed code for failures

Technical Specifications:
- Resampling engine: FAP framework with optimized algorithms
- Processing mode: In-place file replacement with --overwrite
- Output handling: Real-time threaded monitoring system
- Error detection: Return code validation with detailed reporting
- Concurrency: Separate threads for stdout/stderr processing

Error Handling:
- Path existence and type validation with specific messages
- Directory permission verification and access control
- FAP tool availability detection with installation guidance
- Process execution error capture with return code analysis
- Thread synchronization and resource cleanup
- Comprehensive exception handling with status reporting
```

### A.7.7  SEPARATOR

The Separator agent provides advanced audio source separation capabilities for isolating vocal and instrumental components from mixed audio tracks. It utilizes machine learning-based separation algorithms through the FAP framework to perform high-quality vocal extraction and background music isolation across multiple audio files.

Listing 18: Separator Structure

```
Input: data_directory
Output: processing_status

Algorithm:
1. Input validation and directory processing:
     a. Validate data_dir parameter using InputSchema
     b. Convert input path to resolved Path object
     c. Verify directory existence and accessibility
     d. Ensure input is directory type for batch processing
     e. Generate absolute path for consistent file handling

2. Path preprocessing and validation:
     Function _process_path(input_path):
     a. Initialize Path object from input string
     b. Validate path.exists() with detailed error messaging
     c. Verify path.is_dir() for directory type confirmation
     d. Return path.resolve() for absolute path generation
     e. Handle access permissions and directory structure

3. FAP separation command construction:
     a. Build command array: ["fap", "separate", input_dir,
        output_dir, "--overwrite", "--recursive"]
     b. Configure input and output directories (same location)
     c. Enable overwrite mode for existing file replacement
     d. Enable recursive processing for subdirectory traversal
```

```
   e. Filter empty arguments and validate command structure

4. Machine learning-based separation execution:
   a. Initialize subprocess.Popen with pipe configuration:
      - stdout=PIPE, stderr=PIPE for output monitoring
      - bufsize=1 for real-time line-buffered output
   b. Create threading.Thread for concurrent output handling
   c. Execute source separation algorithms on audio files
   d. Monitor processing progress through threaded output

5. Real-time progress monitoring system:
   Function _read_output(pipe):
   a. Implement line-by-line output reading with iter()
   b. Decode subprocess output with UTF-8 and error handling
   c. Format output with [FAP] prefix for identification
   d. Display real-time separation progress to user
   e. Ensure proper pipe closure and resource management

6. Separation result validation and output:
   a. Wait for subprocess completion with return_code capture
   b. Synchronize stdout_thread and stderr_thread completion
   c. Evaluate return_code for separation success/failure
   d. Generate separated audio files (vocals, instrumentals)
   e. Return processing status with detailed error reporting

Technical Specifications:
- Separation engine: Machine learning-based source separation
- Processing scope: Recursive directory traversal capability
- Output formats: Separated vocal and instrumental tracks
- Quality preservation: High-fidelity separation algorithms
- Real-time monitoring: Threaded progress feedback system

Separation Outputs:
- Vocal tracks: Isolated human voice components
- Instrumental tracks: Background music without vocals
- Original preservation: Source files maintained with --overwrite
- Directory structure: Maintains original file organization

Error Handling:
- Directory validation with specific error messages
- FAP tool availability verification and guidance
- Separation algorithm failure detection and reporting
- Thread synchronization and cleanup procedures
- Resource management for large audio file processing
- Comprehensive exception handling with status codes
```

A.7.8 TRANSCRIBER

The Transcriber agent provides comprehensive audio-to-text transcription capabilities using advanced speech recognition models. It processes multiple audio files within a directory structure and generates accurate transcription outputs with real-time progress monitoring and robust error handling.

Listing 19: Transcriber Structure

```
Input: data_directory
Output: processing_status

Algorithm:
1. Input validation and directory processing:
```

```
        a. Validate data_dir parameter using InputSchema
        b. Convert input path to resolved Path object
        c. Verify directory existence and read permissions
        d. Ensure input is directory type for batch processing
        e. Generate absolute path for consistent file access

2. Path preprocessing and validation:
   Function _process_path(input_path):
   a. Initialize Path object from input string
   b. Validate path.exists() with FileNotFoundError handling
   c. Verify path.is_dir() for directory type confirmation
   d. Return path.resolve() for absolute path generation
   e. Handle access control and permission validation

3. FunASR model configuration and command construction:
   a. Build transcription command: ["fap", "transcribe",
      "--model-type", "funasr", "--recursive", audio_dir]
   b. Configure FunASR model for high-accuracy transcription
   c. Enable recursive processing for subdirectory traversal
   d. Set model parameters for optimal speech recognition
   e. Prepare command array for subprocess execution

4. Subprocess execution with real-time monitoring:
   a. Initialize subprocess.Popen with pipe configuration:
      - stdout=PIPE, stderr=PIPE for output capture
      - bufsize=1 for line-buffered real-time feedback
   b. Create threading.Thread for stdout and stderr handling
   c. Execute transcription process with concurrent monitoring
   d. Handle multiple audio file processing simultaneously

5. Real-time progress tracking system:
   Function _read_output(pipe):
   a. Implement iter(pipe.readline, b'') for line iteration
   b. Decode subprocess output with UTF-8 encoding
   c. Apply error handling with 'replace' for corrupted chars
   d. Format output with [FAP] prefix for identification
   e. Display transcription progress and file completion status

6. Transcription completion and result handling:
   a. Wait for subprocess completion with return_code validation
   b. Synchronize stdout_thread and stderr_thread completion
   c. Evaluate return_code for transcription success/failure
   d. Generate transcription files in same directory as audio
   e. Return processing status with detailed error information

Technical Specifications:
- Speech recognition model: FunASR for high-accuracy transcription
- Processing scope: Recursive directory traversal capability
- Output format: Text files (.txt) with same basename as audio
- Model features: Multi-language support and noise robustness
- Real-time feedback: Threaded progress monitoring system

Transcription Outputs:
- Text files: Generated with matching audio file basenames
- Directory structure: Maintains original audio file organization
- Encoding: UTF-8 text files for broad compatibility
- Accuracy: High-quality speech-to-text conversion
- Timestamps: Optional timestamp generation for synchronization
```

```
Error Handling:
- Directory validation with specific error messaging
- FunASR model availability verification and installation guidance
- Audio format compatibility checking and conversion recommendations
- Thread synchronization and resource cleanup procedures
- Comprehensive exception handling with return code analysis
- Graceful degradation for unsupported audio formats
```

### A.7.9 VOICEGENERATOR

The VoiceGenerator agent provides advanced text-to-speech synthesis capabilities for generating high-quality voice audio from scene content. It utilizes the CosyVoice2 model for zero-shot voice cloning and produces synchronized audio with precise timestamp tracking for video editing workflows.

Listing 20: VoiceGenerator Structure

```
Input: video_scene_path, target_vocal_path
Output: synthesized_audio_path, timestamp_path

Algorithm:
1. Model initialization and dependency management:
   a. Import CosyVoice2 and load_wav from cosyvoice.cli.cosyvoice
   b. Load CosyVoice2 model from pretrained_models/CosyVoice2-0.5B
   c. Configure model parameters: load_jit=False, load_trt=False, fp16=False
   d. Load target vocal prompt using load_wav at 16kHz sample rate
   e. Initialize prompt_speech_16k for zero-shot voice synthesis

2. Scene content processing with multilingual support:
   Function _process_with_timestamps(json_file_path):
   a. Load JSON file with UTF-8 encoding for Chinese text support
   b. Extract content_created field from scene JSON structure
   c. Parse content using '/////\n' delimiter pattern
   d. Create segment list with sequential IDs and content mapping
   e. Generate clean JSON output with ensure_ascii=False

3. Text segmentation for optimal TTS processing:
   Function _split_into_sentences(text, max_length=200):
   a. Handle Chinese punctuation
   b. Split by punctuation markers with '|' separator insertion
   c. Apply length-based chunking for sentences exceeding max_length
   d. Further split by commas and Chinese commas
   e. Maintain semantic coherence while respecting length limits

4. Audio generation with timestamp tracking:
   Function _generate_audio_for_segments():
   a. Process each segment through sentence chunking
   b. Apply zero-shot voice synthesis for each text chunk:
      cosyvoice.inference_zero_shot(generator, prompt_text,
      prompt_speech_16k, stream=False)
   c. Concatenate chunk waveforms using torch.cat()
   d. Track cumulative timestamps: current_time += segment_duration
   e. Generate timestamp data structure with precise timing

5. Zero-shot voice synthesis pipeline:
   a. Create single_sentence_generator() for individual text chunks
   b. Apply inference with target voice characteristics
   c. Extract tts_speech waveform from audio_data response
   d. Concatenate waveforms maintaining audio continuity
   e. Handle synthesis errors with graceful chunk skipping
```

```
6. Audio consolidation and export:
   Function _combine_audio_files():
   a. Combine all segment waveforms using torch.cat()
   b. Export final audio using torchaudio.save() with sample_rate
   c. Generate cut_points.json with UTF-8 encoding
   d. Structure timestamp data: sentence_data.chunks with IDs
   e. Clean up temporary segment files after combination

Technical Specifications:
- Voice model: CosyVoice2-0.5B for high-quality synthesis
- Sample rate: 16kHz for prompt audio, model native for output
- Text processing: Max 200 characters per chunk for optimal quality
- Audio format: WAV output with torch audio tensor processing
- Timestamp precision: Millisecond-level accuracy for video sync

Multilingual Features:
- Chinese text support with proper punctuation handling
- UTF-8 encoding throughout processing pipeline
- ensure_ascii=False for character preservation
- Bilingual punctuation recognition and processing
- Cultural text formatting considerations

Error Handling:
- Model loading verification with dependency checking
- File existence validation for scene and prompt inputs
- Chunk-level error isolation with processing continuation
- Resource cleanup for temporary files and memory management
- Comprehensive exception handling with detailed error reporting
```

### A.7.10 TTSINFER

The TTSInfer agent provides sophisticated text-to-speech synthesis for rewritten video content, utilizing sliced audio clips as voice references. It implements a three-stage Fish-Speech pipeline combining VQGAN encoding, text-to-semantic conversion, and audio generation to produce high-quality voice clones matching original speaker characteristics.

Listing 21: TTSInfer Structure

```
Input: audio_path, speech_text_path
Output: derivative_audio_directory

Algorithm:
1. Input processing and text segmentation:
   a. Load rewritten text from speech_path with UTF-8 encoding
   b. Split content into paragraph segments using line breaks
   c. Validate audio_path and extract slice directory structure
   d. Create derivative directory for synthesized output files
   e. Initialize Fish-Speech model paths and dependencies

2. Audio reference preparation and duration analysis:
   Function get_audio_duration(wav_file_path):
   a. Open WAV file using wave.open() with contextlib.closing()
   b. Calculate duration: frames / sample_rate for timing reference
   c. Handle audio format errors with exception management
   d. Return precise duration for reference audio segments

3. LAB file processing and audio segment mapping:
   a. Scan slice directory for .lab files with numerical sorting
```

```
   b. Load LAB content with UTF-8 encoding for text reference
   c. Map corresponding .wav files using Path.with_suffix()
   d. Create lab_files_with_content structure with paths and content

4. Duration-based audio combination strategy:
   For each text paragraph:
   a. Start with corresponding LAB file and WAV audio
   b. Calculate combined_duration from audio segments
   c. If duration < 1 second, aggregate additional segments:
      - Cycle through available LAB files using modulo indexing
      - Concatenate LAB content and WAV file references
      - Continue until minimum 1-second duration achieved
   d. Handle multiple WAV concatenation using scipy.io.wavfile

5. Three-stage Fish-Speech synthesis pipeline:
   Stage 1 - VQGAN Encoding:
   Command: fish_speech/models/vqgan/inference.py
   - Input: combined reference WAV file
   - Checkpoint: firefly-gan-vq-fsq-8x1024-21hz-generator.pth
   - Output: encoded audio features (.npy tokens)

   Stage 2 - Text-to-Semantic Conversion:
   Command: fish_speech/models/text2semantic/inference.py
   - Parameters: --text (target text), --prompt-text (LAB content)
   - Input: prompt tokens from Stage 1
   - Checkpoint: fish-speech-1.5 semantic model
   - Output: semantic codes (temp/codes_0.npy)

   Stage 3 - Audio Generation:
   Command: fish_speech/models/vqgan/inference.py
   - Input: semantic codes from Stage 2
   - Checkpoint: firefly-gan-vq-fsq-8x1024-21hz-generator.pth
   - Output: final synthesized WAV file

6. Batch processing and audio consolidation:
   a. Process each paragraph through complete synthesis pipeline
   b. Monitor subprocess execution with stdout/stderr capture
   c. Validate return codes for each pipeline stage
   d. Collect generated WAV files in derivative directory
   e. Perform final audio concatenation using numpy arrays

7. Final audio merging and cleanup:
   a. Sort generated WAV files by numerical stem for proper ordering
   b. Load audio data using scipy.io.wavfile.read()
   c. Validate consistent sample rates across all segments
   d. Concatenate audio arrays using numpy.concatenate()
   e. Export final merged audio as derivative/final.wav

Technical Specifications:
- Voice synthesis: Fish-Speech 1.5 with VQGAN-based generation
- Audio processing: WAV format with sample rate consistency
- Text encoding: UTF-8 support for multilingual content
- Duration threshold: Minimum 1-second reference for quality synthesis
- Pipeline stages: Sequential VQGAN --> Text2Semantic --> VQGAN

Error Handling:
- Subprocess return code validation for each synthesis stage
- Audio duration calculation with wave format error handling
- Sample rate consistency validation across concatenated segments
```

```
- Temporary file cleanup with graceful error recovery
- Exception isolation for individual paragraph processing failures
- Directory creation and permission handling for output paths
```

### A.7.11 TTSReplace

The TTSReplace agent provides comprehensive video-audio synchronization for replacing original video audio with derivative synthesized speech segments. It implements precise temporal alignment, video speed adjustment, and seamless audio-video merging to produce coherent rewritten video content while maintaining visual continuity.

Listing 22: TTSReplace Structure

```
Input: video_path
Output: final_video_path

Algorithm:
1. Input validation and directory structure setup:
   a. Validate video_path existence and file accessibility
   b. Extract slice_dir from video filename without extension
   c. Verify derivative audio directory and metadata.json existence
   d. Create final output directory structure for processed clips
   e. Configure UTF-8 encoding for stdout/stderr handling

2. Metadata processing and clip mapping:
   a. Load metadata.json containing temporal clip information
   b. Extract clip data: file names, start times, end times, durations
   c. Map derivative audio files to corresponding video segments
   d. Validate derivative audio availability for each clip
   e. Initialize processed_files array for concatenation tracking

3. Video clip extraction with temporal precision:
   For each metadata clip:
   a. Execute FFmpeg video extraction command:
      ffmpeg -y -ss start_time -i video_path -to duration
      -c:v libx264 -preset fast -vf scale=iw:ih clip_path
   b. Apply precise temporal trimming using start/end timestamps
   c. Maintain video quality with libx264 fast preset
   d. Preserve original resolution with scale=iw:ih filter

4. Audio duration analysis and speed calculation:
   a. Use FFprobe to determine derivative audio duration:
      ffprobe -v error -show_entries format=duration
      -of default=noprint_wrappers=1:nokey=1 derivative_audio
   b. Calculate speed adjustment factor: target_duration / clip_duration
   c. Determine video playback speed modification requirements
   d. Handle duration mismatches between audio and video segments

5. Video speed adjustment for audio synchronization:
   a. Apply temporal adjustment using setpts filter:
      ffmpeg -y -i clip_path -filter:v setpts=speed_factor*PTS
      -an adjusted_path
   b. Remove audio track (-an) to prepare for replacement
   c. Modify video playback speed to match derivative audio duration
   d. Maintain frame rate consistency throughout adjustment

6. Audio-video merging with quality preservation:
   a. Combine adjusted video with derivative audio:
      ffmpeg -y -i adjusted_path -i derivative_audio
```

```
    -c:v copy -c:a aac -map 0:v:0 -map 1:a:0
    -shortest merged_path
  b. Use copy codec for video to avoid quality loss
  c. Encode audio to AAC for broad compatibility
  d. Apply shortest stream duration for synchronization

7. Batch concatenation and final assembly:
  a. Generate filelist.txt with absolute paths for FFmpeg concat
  b. Write UTF-8 encoded file list for proper path handling
  c. Execute FFmpeg concatenation:
    ffmpeg -y -f concat -safe 0 -i filelist.txt
    -c copy final_output
  d. Use copy codec for lossless segment joining

8. Cleanup and resource management:
  a. Iterate through output directory for intermediate file removal
  b. Preserve only final.mp4 output while removing temporary clips
  c. Handle file deletion errors with graceful error reporting
  d. Display cleanup progress and completion status

Technical Specifications:
- Video encoding: H.264 with fast preset for efficiency
- Audio encoding: AAC for universal compatibility
- Temporal precision: Frame-accurate clip extraction and alignment
- Speed adjustment: setpts filter for smooth temporal modification
- Concatenation: FFmpeg concat demuxer for seamless joining

Synchronization Features:
- Dynamic speed adjustment based on audio duration differences
- Precise temporal alignment using FFprobe duration analysis
- Quality preservation through copy codec usage during concatenation
- Frame rate consistency maintenance throughout processing pipeline

Error Handling:
- File existence validation for video, audio, and metadata
- Subprocess execution monitoring with return code checking
- Graceful handling of missing derivative audio segments
- Resource cleanup with error isolation for individual file operations
- UTF-8 encoding configuration for international character support
```

### A.7.12 TTSSLICER

The TTSSlicer agent provides intelligent audio segmentation for preparing audio content for text-to-speech processing and transcription workflows. It implements advanced silence detection algorithms with RMS-based energy analysis to create optimal audio chunks while maintaining temporal precision and content coherence.

Listing 23: TTSSlicer Structure

```
Input: audio_path
Output: processing_status, segmented_audio_files, metadata

Algorithm:
1. Configuration and parameter initialization:
  a. Set duration constraints: min_duration=6.0s, max_duration=8.0s
  b. Configure silence detection: min_silence_duration=0.5s, top_db=-35
  c. Set analysis parameters: hop_length=10ms, max_silence_kept=0.3s
  d. Initialize merge_short=False for individual segment processing
```

```
2. Audio loading and preprocessing:
   a. Load audio using librosa.load() with original sample rate preservation
   b. Handle mono/stereo conversion: expand mono to 2D array format
   c. Validate audio data integrity and duration constraints
   d. Create output directory structure based on audio filename

3. RMS-based silence detection pipeline:
   Class _Slicer implementation:
   a. Calculate RMS energy using librosa.feature.rms():
      - frame_length=win_size, hop_length=hop_size
      - Apply threshold: 10^(top_db/20.0) for dB to linear conversion
   b. Analyze energy profile for silence identification
   c. Track silence_start and clip_start positions for segmentation

4. Intelligent segmentation logic:
   For each RMS frame:
   a. Detect silence regions where rms < threshold
   b. Apply segmentation rules:
      - Leading silence: silence_start == 0 and i > max_sil_kept
      - Middle silence: i - silence_start >= min_interval and
        i - clip_start >= min_length
   c. Calculate optimal cut points using argmin() on RMS values
   d. Generate sil_tags array with (start, end) silence boundaries

5. Chunk extraction with temporal precision:
   Function _apply_slice(waveform, begin, end):
   a. Convert frame indices to sample indices: idx * hop_size
   b. Extract audio slice maintaining original channel structure
   c. Calculate precise timestamps: start_idx/sr, end_idx/sr
   d. Return structured chunk with audio data and timing information

6. Duration-based post-processing:
   a. Short chunk merging (if merge_short enabled):
      - Combine consecutive chunks until max_duration reached
      - Maintain temporal continuity through concatenation
   b. Long chunk subdivision:
      - Split chunks exceeding max_duration using _slice_by_max_duration()
      - Calculate optimal chunk_size: ceil(total_samples / n_chunks)
      - Ensure uniform distribution across subdivided segments

7. Audio export and metadata generation:
   a. Save each chunk using soundfile.write():
      - Transpose multi-channel audio for proper format
      - Maintain original sample rate and bit depth
   b. Generate metadata with precise timing:
      - filename: sequential numbering (0000.wav, 0001.wav, etc.)
      - start/end timestamps rounded to 3 decimal places
      - duration calculation: end - start with precision
   c. Export metadata.json with UTF-8 encoding for compatibility

Technical Specifications:
- Silence threshold: -35 dB for robust speech detection
- Temporal resolution: 10ms hop length for precise timing
- Segment constraints: 6-8 second optimal duration range
- RMS analysis: Frame-based energy calculation with librosa
- Output format: WAV files with original sample rate preservation

Segmentation Features:
- Adaptive silence detection with configurable thresholds
```

```
- Intelligent cut point selection using minimum RMS values
- Leading/trailing silence handling with preservation limits
- Long audio subdivision for consistent chunk sizes
- Metadata tracking for downstream processing integration

Error Handling:
- Audio format validation with librosa compatibility checking
- Empty audio detection and graceful handling
- Directory creation with permission validation
- Sample rate preservation across processing pipeline
- Robust file I/O with exception management for large files
```

### A.7.13   SVCANALYZER

The SVCAnalyzer agent provides comprehensive MIDI file analysis for music cover creation workflows, extracting detailed musical information including note sequences, timing data, and tempo variations. It performs intelligent lyrics-to-music alignment by matching character counts with actual note sequences while handling rest periods and tempo changes.

Listing 24: SVCAnalyzer Structure

```
Input: midi_file_path, lyrics_file_path
Output: song_name, analysis_results_path

Algorithm:
1. Input validation and file processing:
   a. Validate MIDI file extension (.mid) and accessibility
   b. Validate lyrics file extension (.txt) and encoding
   c. Extract song name from lyrics filename without extension
   d. Load lyrics content with UTF-8 encoding for character analysis

2. MIDI file parsing and track analysis:
   a. Load MIDI file using mido.MidiFile() for comprehensive parsing
   b. Iterate through all tracks to identify musical content
   c. Extract track names or assign default identifiers
   d. Initialize note tracking structures for temporal analysis

3. Tempo change detection and BPM calculation:
   Function get_tempo_changes(mid):
   a. Scan all tracks for 'set_tempo' messages
   b. Record timestamp and tempo value for each change
   c. Sort tempo changes chronologically by time
   d. Insert default tempo (500000 microseconds) if none at start
   e. Calculate BPM: 60000000 / tempo_microseconds

4. Note extraction with temporal precision:
   For each MIDI track:
   a. Process note_on messages with velocity > 0 for note starts
   b. Process note_off messages or note_on with velocity = 0 for ends
   c. Calculate note duration: end_time - start_time in ticks
   d. Detect rest periods: gaps > ticks_per_beat / 8 threshold
   e. Group simultaneous notes occurring within 0.01 time units

5. Time conversion with tempo awareness:
   Function ticks_to_seconds():
   a. Handle variable tempo throughout song duration
   b. Calculate duration for each tempo segment separately
   c. Apply formula: (ticks * tempo) / (ticks_per_beat * 1000000)
   d. Accumulate total duration across tempo changes
```

```
   e. Ensure precise timing for musical synchronization

6. Note name conversion and formatting:
   Function note_to_name(note_number):
   a. Convert MIDI note numbers to musical notation
   b. Calculate octave: note_number // 12 - 1
   c. Determine note name using chromatic scale array
   d. Format as: "{note}{octave}" (e.g., "C4", "F#3")
   e. Handle chord notation for simultaneous notes

7. Lyrics-to-music alignment and validation:
   a. Count actual notes excluding rest periods
   b. Compare note count with lyrics character count
   c. For matching counts:
      - Insert "AP" markers for rest periods in lyrics
      - Map each character to corresponding note
      - Maintain temporal alignment throughout song
   d. Generate processed lyrics with rest markers

8. Analysis output generation and export:
   a. Create structured JSON output with:
      - text: processed lyrics with AP markers
      - notes: pipe-separated note sequence
      - notes_duration: corresponding timing data
      - input_type: "word" for character-level processing
   b. Save analysis to /analysis/{song_name}.json
   c. Ensure UTF-8 encoding with ensure_ascii=False

Technical Specifications:
- MIDI parsing: mido library for comprehensive format support
- Temporal resolution: Tick-based timing with tempo-aware conversion
- Note grouping: 0.01 time unit threshold for simultaneous detection
- Rest detection: ticks_per_beat / 8 minimum gap threshold
- Precision: 6 decimal places for duration measurements

Musical Features:
- Multi-track analysis with automatic track selection
- Chord detection and simultaneous note handling
- Variable tempo support throughout song duration
- Rest period insertion for natural speech rhythm
- Character-to-note mapping for lyrics synchronization

Error Handling:
- MIDI file format validation and parsing error recovery
- Lyrics encoding detection with UTF-8 fallback
- Track selection validation for meaningful musical content
- Character count mismatch detection and reporting
- Directory creation with permission handling for output files
```

A.7.14 SVCCONVERSION

The SVCConversion agent provides intelligent audio segmentation and timestamp generation for music cover workflows, converting adapted lyrics into precisely timed JSON format suitable for video generation. It handles complex temporal alignment by parsing musical structure markers and generating seamless transition points for multimedia editing.

Listing 25: SVCConversion Structure

```
Input: adapted_lyrics_string, midi_analysis_path
```

```
Output: timestamp_json_path

Algorithm:
1. Input validation and data loading:
   a. Validate adapted lyrics string and analysis file path
   b. Load MIDI analysis JSON with UTF-8 encoding
   c. Update analysis data with adapted lyrics content
   d. Extract duration array from notes_duration field (pipe-separated)

2. Text parsing and temporal alignment:
   Function parse_text_to_segments(text, durations):
   a. Initialize timeline array and AP time ranges tracker
   b. Process character-by-character with duration mapping:
      - Detect "AP" markers for musical phrase boundaries
      - Map individual characters to corresponding note durations
      - Track cumulative time progression through song
   c. Generate timeline entries: ("AP", start, end) or ("CHAR", char, start,
→ end)

3. Segment boundary detection and grouping:
   a. Process timeline to identify lyrical segments between AP markers
   b. Group consecutive characters into coherent text segments
   c. Calculate segment boundaries:
      - start: end of previous AP marker or beginning
      - end: start of next AP marker or final character
   d. Handle edge cases for segments at song boundaries

4. AP marker processing for instrumental breaks:
   a. Extract AP time ranges for instrumental/rest periods
   b. Identify extended AP durations exceeding 12-second threshold
   c. For long AP segments (duration > 12s):
      - Generate intermediate chunk points every 12 seconds
      - Create "bgm" content markers for background music
      - Ensure smooth transitions for video editing workflows

5. Timestamp generation and chunk creation:
   a. Generate text chunks from lyrical segments:
      - timestamp: segment end time for completion marking
      - content: complete text content of segment
      - type: "text" for lyrical content identification
   b. Generate AP chunks for instrumental breaks:
      - timestamp: calculated break points within long AP sections
      - content: "bgm" for background music indication
      - type: "ap" for instrumental section identification

6. Temporal sorting and priority handling:
   a. Combine text and AP chunks into unified entry array
   b. Apply sorting criteria: (timestamp, content_type_priority)
      - Primary sort: chronological by timestamp
      - Secondary sort: text entries before AP entries at same timestamp
   c. Ensure temporal consistency for seamless playback

7. JSON output generation and export:
   a. Create structured output format:
      sentence_data: {
        count: total_chunk_number,
        chunks: [
           {id: sequential_index, timestamp: precise_time, content: text_or_bgm
→ }
```

```
        ]
    }
  b. Round timestamps to 3 decimal places for precision
  c. Export to dataset/video_edit/voice_gen/gen_audio_timestamps.json
  d. Ensure UTF-8 encoding with ensure_ascii=False

Technical Specifications:
- Temporal precision: 3 decimal place timestamp accuracy
- Chunk segmentation: 12-second maximum for extended instrumental breaks
- Character mapping: One-to-one correspondence with MIDI note durations
- Priority system: Text content prioritized over background music markers
- Format compatibility: JSON structure optimized for video editing workflows

Musical Structure Features:
- AP marker recognition for phrase and verse boundaries
- Instrumental break detection and subdivision
- Character-level timing precision for lip-sync accuracy
- Background music insertion points for extended instrumental sections
- Seamless transition management between lyrical and instrumental content

Error Handling:
- Duration array validation with length consistency checking
- Timeline boundary validation for segment integrity
- File I/O error management for JSON export operations
- Character encoding preservation for international lyrics
- Directory creation with permission handling for output paths
```

### A.7.15 SVCCOVERIST

The SVCCoverist agent provides advanced voice timbre cloning and conversion capabilities for singing voice synthesis in music cover production. It utilizes the Seed-VC (Voice Conversion) framework to perform source-to-target vocal transformation while maintaining musical characteristics and timing precision for professional-quality audio output.

Listing 26: SVCCoverist Structure

```
Input: source_audio_path, target_vocal_path
Output: synthesized_audio_path

Algorithm:
1. Input validation and path processing:
   a. Validate source audio and target vocal file accessibility
   b. Convert file paths to absolute paths for cross-directory execution
   c. Extract source and target filenames without extensions
   d. Generate output filename: "{source_name}_{target_name}.wav"

2. Directory structure setup and navigation:
   a. Calculate final output directory: "../../final" relative to source
   b. Create output directory structure with proper permissions
   c. Store original working directory for restoration
   d. Navigate to tools/seed-vc for model execution environment

3. Environment configuration for Seed-VC execution:
   a. Store original PYTHONPATH environment variable
   b. Set PYTHONPATH to seed-vc absolute directory path
   c. Configure Python module import resolution for seed-vc
   d. Ensure model dependencies and checkpoints accessibility

4. Voice conversion command construction:
```

```
    a. Build subprocess command array:
       [python, "inference.py", "--source", source_path,
        "--target", target_path, "--output", output_dir,
        "--f0-condition", "True"]
    b. Enable F0 conditioning for pitch characteristic preservation
    c. Configure input/output paths for proper file handling

5. Subprocess execution with encoding management:
    a. Execute inference.py with UTF-8 text capture
    b. Handle UnicodeDecodeError with fallback decoding:
       - Capture output as bytes if UTF-8 fails
       - Apply decode with 'replace' error handling
       - Ensure readable output for debugging and monitoring
    c. Monitor stdout and stderr for process feedback

6. Voice conversion processing pipeline:
    a. Load source audio for vocal content extraction
    b. Load target audio for timbre characteristic analysis
    c. Apply neural voice conversion with F0 conditioning:
       - Preserve pitch patterns from source audio
       - Transfer vocal timbre from target speaker
       - Maintain temporal alignment and musical timing
    d. Generate converted audio with target voice characteristics

7. Output validation and cleanup:
    a. Verify successful command execution (return_code == 0)
    b. Validate output file creation in final directory
    c. Restore original working directory and environment:
       - Change back to original directory
       - Restore original PYTHONPATH or remove if empty
       - Clean up temporary environment modifications
    d. Return synthesized audio path for downstream processing

Technical Specifications:
- Voice conversion model: Seed-VC with F0 conditioning enabled
- Audio preservation: Pitch pattern and timing maintenance
- Output format: WAV files with high-quality audio encoding
- Environment isolation: Sandboxed execution with path restoration
- Error handling: Unicode decoding with graceful fallback

Voice Conversion Features:
- Timbre transfer: Source-to-target vocal characteristic mapping
- F0 conditioning: Pitch pattern preservation during conversion
- Musical timing: Temporal alignment maintenance for singing voice
- Quality preservation: High-fidelity audio output for professional use
- Batch processing: Support for multiple source-target combinations

Error Handling:
- File path validation with absolute path conversion
- Directory creation with permission error management
- Subprocess execution monitoring with return code validation
- Unicode encoding error recovery with 'replace' strategy
- Environment restoration with original state preservation
- Comprehensive exception handling with detailed error reporting
```

A.7.16   SVCSINGLE

The SVCSingle agent provides comprehensive singing voice synthesis for music cover production, converting adapted lyrics into high-quality vocal audio with precise temporal alignment. It utilizes the DiffSinger framework for neural singing voice generation and implements sophisticated audio processing techniques to ensure accurate timing and seamless segment concatenation.

Listing 27: SVCSingle Structure

```
Input: adapted_lyrics_string, midi_analysis_path, song_name
Output: synthesized_vocal_audio_path

Algorithm:
1. Input validation and MIDI analysis loading:
   a. Validate adapted lyrics string and analysis file path
   b. Load MIDI analysis JSON with UTF-8 encoding
   c. Extract notes and duration data from analysis structure
   d. Prepare input structure with text, notes, and timing information

2. Lyrics segmentation and alignment processing:
   Function _split_single_annotation():
   a. Split text by "AP" markers for phrase boundary detection
   b. Filter empty segments and preserve AP delimiters
   c. Convert non-AP segments to character-level arrays
   d. Generate aligned notes_list and notes_duration_list
   e. Validate index correspondence between text and musical data

3. Segment creation with minimum duration optimization:
   Function _create_segment_with_min_duration():
   a. Apply duration threshold: 0.15s minimum per segment
   b. For segments below threshold, aggregate consecutive characters:
      - Combine text content until minimum 0.5s duration
      - Concatenate notes using " | " separator
      - Merge duration values for unified segment timing
   c. Preserve AP markers as individual segments for rest periods

4. Batch audio generation with DiffSinger integration:
   a. Create temporary JSON files for each vocal segment
   b. Generate filename format: "{song_name}_part_{start}-{end}.json"
   c. Execute run_diffsinger() for neural voice synthesis:
      - Input: JSON annotation files with text and musical data
      - Output: WAV files with synthesized singing voice
   d. Handle batch processing for efficient resource utilization

5. Temporal precision audio processing:
   Function _phase_vocoder_stretch():
   a. Load generated audio using librosa.load() at 44.1kHz
   b. Calculate stretch factor: current_duration / target_duration
   c. Apply time stretching with librosa.effects.time_stretch()
   d. Perform sample-level alignment for precise timing:
      - Truncate excess samples if audio too long
      - Pad with zeros if audio too short
      - Ensure exact sample count matching target duration

6. Audio segment consolidation and timing verification:
   a. Initialize combined_audio with 44.1kHz AudioSegment
   b. Process each segment with millisecond-level precision:
      - Convert numpy arrays to 16-bit PCM format
      - Apply clip normalization: np.clip(audio, -1.0, 1.0) * 32767
      - Force duration alignment with target millisecond values
```

```
   c. Handle AP segments with precise silence generation
   d. Concatenate segments maintaining temporal continuity

7. Quality assurance and output generation:
   a. Validate total duration against expected timing
   b. Report timing errors and adjustment statistics
   c. Export final audio to dataset/mad_svc/cover directory
   d. Clean up temporary files and intermediate outputs
   e. Return absolute path to synthesized vocal audio

Technical Specifications:
- Synthesis engine: DiffSinger neural singing voice model
- Audio format: 44.1kHz WAV with 16-bit PCM encoding
- Temporal precision: Millisecond-level alignment accuracy
- Segment processing: Character-level with minimum duration optimization
- Quality control: Sample-level timing verification and correction

Audio Processing Features:
- Phase vocoder time stretching for duration matching
- Automatic silence insertion for AP markers
- Dynamic segment aggregation for optimal synthesis quality
- Real-time duration monitoring and error reporting
- Batch processing optimization for multiple segment synthesis

Error Handling:
- JSON file validation with UTF-8 encoding support
- Audio file existence verification after synthesis
- Duration mismatch detection with automatic correction
- Temporary file cleanup with exception safety
- Sample rate consistency validation throughout pipeline
- Comprehensive error reporting for debugging and optimization
```

### A.7.17  STANDUPCONVERSION

The StandUpConversion agent provides precise temporal mapping for stand-up comedy audio segments, converting individual audio files into structured timestamp format suitable for video generation workflows. It analyzes audio duration and creates synchronized timing data for seamless multimedia editing and video synchronization.

Listing 28: StandUpConversion Structure

```
Input: segment_directory, metadata_path
Output: timestamp_json_path

Algorithm:
1. Input validation and metadata loading:
   a. Validate segment directory and metadata file paths
   b. Load metadata JSON containing script analysis results:
      - tone information for each segment
      - text content without formatting markers
      - reaction indicators for audience response integration
   c. Initialize timing tracking variables and chunk array

2. Audio file analysis and duration calculation:
   For each metadata entry:
   a. Construct audio file path: {seg_dir}/{index}.wav
   b. Load audio data using soundfile.read():
      - Extract audio samples and sample rate
      - Handle various audio formats with sf compatibility
```

```
   c. Calculate precise duration: len(audio_samples) / sample_rate
   d. Track cumulative timing for sequential segments

3. Timestamp generation with cumulative timing:
   a. Maintain current_time tracker for running total
   b. For each segment:
      - Calculate end_time = current_time + segment_duration
      - Round timestamp to 3 decimal places for precision
      - Update current_time for next segment calculation
   c. Handle timing gaps and overlaps with precise calculation

4. Chunk data structure creation:
   a. Generate structured chunk entries:
      {
        id: sequential_index (1-based),
        timestamp: cumulative_end_time,
        content: original_text_content
      }
   b. Preserve original text content from metadata analysis
   c. Maintain sequential ordering for proper video synchronization

5. Error handling and file validation:
   a. Validate audio file existence for each expected segment
   b. Handle corrupted or missing audio files gracefully:
      - Log specific file errors with detailed messages
      - Continue processing remaining segments
      - Maintain timing accuracy despite missing segments
   c. Apply robust audio format compatibility checking

6. JSON output generation and export:
   a. Structure final timestamp data:
      sentence_data: {
        count: total_valid_chunks,
        chunks: [chunk_array_with_timing_data]
      }
   b. Export to timestamps.json in metadata directory
   c. Ensure UTF-8 encoding with ensure_ascii=False
   d. Apply proper JSON formatting with 2-space indentation

7. Timing validation and quality assurance:
   a. Verify sequential timestamp progression
   b. Check for negative durations or timing anomalies
   c. Validate total duration against expected performance length
   d. Report timing statistics for quality control

Technical Specifications:
- Audio analysis: soundfile library for precise duration calculation
- Timing precision: 3 decimal places for millisecond-level accuracy
- Format compatibility: Multi-format audio support through soundfile
- Data structure: JSON with nested sentence_data organization
- Error tolerance: Graceful handling of missing or corrupted segments

Timestamp Features:
- Cumulative timing calculation for proper sequence alignment
- Sample-rate aware duration computation for accuracy
- Sequential ID assignment for video editing reference
- Content preservation from original script analysis
- Robust error handling for production workflow reliability
```

```
Video Synchronization Preparation:
- Compatible format for VideoConversion agent integration
- Precise timing data for frame-accurate video editing
- Content mapping for automated video scene generation
- Seamless transition support for multimedia workflows
- Quality assurance validation for downstream processing

Error Handling:
- Audio file existence verification with detailed error reporting
- Sample rate consistency checking across segments
- Metadata correlation validation with audio file availability
- JSON export error management with fallback procedures
- Comprehensive exception handling for production stability
```

### A.7.18 CROSSTALKCONVERSION

The CrossTalkConversion agent provides precise temporal mapping for crosstalk audio segments, converting dual-performer dialogue into structured timestamp format suitable for video generation workflows. It analyzes audio duration and creates synchronized timing data with performer identification for seamless multimedia editing and video synchronization.

Listing 29: CrossTalkConversion Structure

```
Input: segment_directory, metadata_path
Output: timestamp_json_path

Algorithm:
1. Input validation and metadata loading:
   a. Validate segment directory and metadata file paths
   b. Load metadata JSON containing crosstalk analysis results:
      - role information for each dialogue segment
      - tone data for delivery style reference
      - text content without formatting markers
      - optional reaction indicators for audience response
   c. Initialize timing tracking variables and chunk array

2. Audio file analysis and duration calculation:
   For each metadata entry:
   a. Construct audio file path: {seg_dir}/{index}.wav
   b. Load audio data using soundfile.read():
      - Extract audio samples and sample rate information
      - Handle various audio formats with soundfile compatibility
   c. Calculate precise duration: len(audio_samples) / sample_rate
   d. Track cumulative timing for sequential dialogue segments

3. Timestamp generation with cumulative timing:
   a. Maintain current_time tracker for running total duration
   b. For each dialogue segment:
      - Calculate end_time = current_time + segment_duration
      - Round timestamp to 3 decimal places for precision
      - Update current_time for next segment calculation
   c. Handle timing gaps and overlaps with precise calculation

4. Content formatting with performer identification:
   a. Generate performer-tagged content structure:
      "[{performer_role}] {dialogue_text}"
   b. Preserve original role information from metadata:
      - Funny guy (dou_gen) performer identification
      - Setup guy (peng_gen) performer identification
```

```
   c. Maintain dialogue attribution for video scene generation

5. Chunk data structure creation:
   a. Generate structured chunk entries:
      {
        id: sequential_index (1-based),
        timestamp: cumulative_end_time,
        content: "[role] dialogue_text"
      }
   b. Preserve performer role tags for video editing reference
   c. Maintain sequential ordering for proper dialogue flow

6. Error handling and file validation:
   a. Validate audio file existence for each expected segment
   b. Handle corrupted or missing audio files gracefully:
      - Log specific file errors with detailed messages
      - Continue processing remaining dialogue segments
      - Maintain timing accuracy despite missing segments
   c. Apply robust audio format compatibility checking

7. JSON output generation and export:
   a. Structure final timestamp data:
      sentence_data: {
        count: total_valid_chunks,
        chunks: [chunk_array_with_timing_and_roles]
      }
   b. Export to timestamps.json in metadata directory
   c. Ensure UTF-8 encoding with ensure_ascii=False
   d. Apply proper JSON formatting with 2-space indentation

8. Timing validation and quality assurance:
   a. Verify sequential timestamp progression across dialogue
   b. Check for negative durations or timing anomalies
   c. Validate total duration against expected performance length
   d. Report timing statistics and performer distribution

Technical Specifications:
- Audio analysis: soundfile library for precise duration calculation
- Timing precision: 3 decimal places for millisecond-level accuracy
- Format compatibility: Multi-format audio support through soundfile
- Data structure: JSON with nested sentence_data organization
- Role preservation: Performer identification tags in content field

Crosstalk-Specific Features:
- Dual-performer role tracking for video scene generation
- Performer-tagged content for automated video editing
- Sequential dialogue timing for natural conversation flow
- Error tolerance for production workflow reliability
- Metadata correlation with original script analysis

Video Synchronization Preparation:
- Compatible format for VideoConversion agent integration
- Performer identification for automated scene switching
- Precise timing data for frame-accurate video editing
- Content mapping for dual-performer video generation
- Seamless transition support for dialogue-based multimedia

Error Handling:
- Audio file existence verification with detailed error reporting
```

```
- Sample rate consistency checking across dialogue segments
- Metadata correlation validation with audio file availability
- JSON export error management with fallback procedures
- Comprehensive exception handling for production stability
- Performer role validation for consistent content formatting
```

### A.8 KNOWLEDGE ANALYSIS AGENTS

Knowledge analysis agents are specialized components that process and synthesize multimedia content to generate coherent narratives and creative outputs. These agents combine video content extraction, narrative summarization, and rhythm-aware storyboard generation to create comprehensive multimedia knowledge representations. The knowledge analysis pipeline incorporates semantic understanding, temporal synchronization, and creative synthesis to transform raw content into structured creative materials suitable for video production workflows.

#### A.8.1 RHYTHMCONTENTGENERATOR WITH GLOBAL-AWARE MECHANISM

The RhythmContentGenerator extracts video segment content and creates scene-focused narrative summaries that incorporate user creative requirements. The Global-Aware Mechanism enhances video retrieval by refining raw user input into captions-aware queries through a two-stage process: first building and analyzing a comprehensive caption bank from video content, then integrating this with rhythm analysis to generate fine-grained, contextually grounded storyboard-based subqueries.

Listing 30: RhythmContentGenerator Structure

```
Input: user_requirements, rhythm_analysis_directory, video_segments_path
Output: video_scene_path, video_summary_path

Algorithm:
1. Content extraction and caption bank construction:
   a. Load video segments from kv_store_video_segments.json
   b. Extract content from all video segments across videos
   c. Transform raw captions to numbered video segments format:
      'Caption:' > 'Video Segments {id}:'
   d. Build comprehensive caption bank with sequential numbering

2. Video content contextualization:
   System: "You are a creative beat sync video producer who is good at
   write scenes from ground truth video segments, strictly following
   the user's requirements."

   Video context prompt template:
   "Here is the visual description of all video segments from source video:

   {video_summary}

   Each video segment represents a scene from the ground truth video.
   Later, you'll need to use these video segments content to write
   storyboards."

3. Rhythm analysis integration:
   a. Parse rhythm_points.json for beat count extraction
   b. Load rhythm visualization from rhythm_detection.png
   c. Generate rhythm reference context:
      "Background Music Visualization with Rhythm Points for Reference:
       - Plot shows musical intensity and rhythm patterns over time
       - Peaks represent high-energy moments suitable for impactful scenes
       - Valleys indicate calmer segments for transitional content
       - Use visualization to guide scene pacing and emotional intensity"
```

```
4. Global-aware storyboard generation with conversation state:
   Conversation state management:
   - Initialize conversation tracking array
   - Maintain context across multi-turn interactions
   - Store user and assistant messages for continuity

   Storyboard creation prompt with global awareness:
   "Build rhythm-synchronized video storyboards from ground truth video
   segments content, aligning with user's requirements

   {rhythm_reference}

   Total Scenes Required: {sections_num}
   User's creative requests (high priority): '{user_idea}'

   Video Storyboards Guidelines:
   1. Scene Structure: Begin each with /////, number 1 to {sections_num}
   2. Visual Requirements: Detailed character appearances, rich motion
      descriptions, no dialogue required
   3. Rhythm Integration: Match scene intensity with visualization patterns
   4. Content Rules: Max two sentences per scene, focus on visual elements,
      maintain narrative flow, base on previous grounded video segments

   Format Output: /////\n[Scene description]\n\n/////\n[Scene description]"

5. API interaction with retry mechanism:
   @tenacity.retry configuration:
   - wait=exponential(multiplier=1, min=2, max=60)
   - stop=after_attempt(5)
   - before_sleep=logging with attempt number

   Conversation-aware API calls:
   def _call_gpt_api(user_prompt, temperature=0.7):
       self.conversation.append({"role": "user", "content": user_prompt})
       if len(self.conversation) > 1:
           initial_message = "In previous conversation, you said: " +
                             assistant_messages[-1] + "\n\n"
       combined_prompt = initial_message + user_prompt
       response = gpt(model=self.model, system=self.system_message,
                      user=combined_prompt)
       assistant_response = response.choices[0].message.content
       self.conversation.append({"role": "assistant", "content":
↪  assistant_response})

6. Output generation and validation:
   a. Create structured JSON with user_idea, video_summary, segment_scene
   b. Save storyboard to video_scene.json with UTF-8 encoding
   c. Generate preview with truncation for large content
   d. Return file paths for downstream video processing

Error Handling:
- Tenacity retry logic for robust API interactions
- Fallback mechanisms for missing rhythm analysis files
- Graceful degradation when video content unavailable
- Boolean conversion for string-based configuration parameters
- Default value assignment for missing beat count data
```

### A.8.2 NEWSCONTENTGENERATOR

The NewsContentGenerator creates news summary content by implementing a dual-agent pipeline that processes reference video transcripts and adapts them according to user requirements and presentation styles. The algorithm employs a presenter agent for content adaptation and a judger agent for structural formatting.

Listing 31: NewsContentGenerator Structure

```
Input: user_requirements, news_presentation_style, video_directory
Output: video_scene_json

Algorithm:
1. Transcript discovery and loading:
   a. Scan video directory for .lab files with multiple encoding support
   b. Parse .lab file format: [start_time end_time transcription]
   c. Extract transcription text while preserving content integrity
   d. Handle encoding fallbacks: utf-8, gb18030, gbk, gb2312, cp1252

2. Presentation style processing:
   a. Load presentation methodology from file or direct string input
   b. Parse formatting guidelines and style requirements
   c. Prepare template for content adaptation pipeline

3. Presenter agent processing with embedded prompt:
   System: "You are an experienced expert in news writing skit review
   copy. Pay special attention to user's words count requirements."

   User prompt template:
   "Create skit narration copy, strictly following user's ideas and
   presentation methods.

   User's idea: '{user_idea}'
   Grounded text content: {content}
   Follow this presentation method: {present_content}

   Requirements:
   1. Format: Remove section numbers, min 11 words per sentence
   2. Content: Use original dialogues, focus on plot elements
   3. Language: Third-person, convert numbers to words, separate
      abbreviations (ChatGPT --> Chat GPT)"

4. Judger agent formatting with embedded prompt:
   System: "You are a content formatting specialist with expertise
   in following guidelines"

   User prompt template:
   "Format content with requirements:
   - Remove all commas
   - Start with /////\n
   - Chunk each period with \n\n/////\n
   - Keep original content, separate sentences

   Example: /////\nGood morning everyone nice to meet you again.
   \n\n/////\nThe weather is very nice today."

5. Visual scene generation:
   a. Process formatted content for scene extraction
   b. Generate English visual-scene keywords and descriptions
   c. Apply scene translation with embedded prompt:
```

```
3402        "Deduce visual-scene keywords, each section some scene keywords
3403         (proper nouns: iPhone 16, SWE Arena Benchmark)
3404         Keep same paragraph separators, max 1 sentence per section"
3405
3406   6. Content validation and output:
3407      a. Count content sections using ///// markers
3408      b. Generate structured JSON with user_idea, content_created,
3409         segment_scene
3410      c. Save to video_scene.json with UTF-8 encoding
3411      d. Provide section statistics and preview
3412
3413   Error Handling:
3414   - Multi-encoding file reading with graceful fallbacks
3415   - API retry logic with exponential backoff (5 attempts)
3416   - Content truncation for oversized inputs (15K character limit)
3417   - Fallback formatting for judger agent failures
```

### A.8.3  NEWS PRESENTATION STYLE SYSTEM PROMPT

This subsection outlines the standardized presentation methodology for video event overview content generation, ensuring consistent narrative structure and professional delivery across all multimedia production workflows.

Listing 32: Video Event Overview Presentation Style

```
Core Requirements:
1. Third-person narrative perspective throughout content
2. Factual accuracy: avoid fabricated specifications (e.g., iPhone 15, not
↪ iPhone 15.0)
3. Proper subject attribution: understand main characters (Apple released
↪ iPhone 4,
   not "iPhone 4 released new product")
4. No greeting/farewell formalities (avoid "good morning," "thank you for
↪ watching")
5. Focus: comprehensive video event overview generation

OPENING SECTION (First Three Sentences):
Objective: Create compelling hook with core concept presentation
- Immediately establish central narrative tension
- Highlight primary innovation or conflict
- Engage audience with compelling opening statement

Example Structure:
"Meta just announced revolutionary virtual reality equipment. It can visualize
immersive environments in real time that would require fastest GPUs several
↪ hours
to render. This breakthrough addresses accelerated processing challenges
↪ called [specific technology]."

Alternative Opening Approaches:
- Situational context establishment
- Hypothetical scenario presentation
- Generational impact framing
- Direct technical revelation

CONTENT BODY SECTION (Primary Narrative):
Structure: Timeline-based plot advancement with clear progression
- Unfold events chronologically with logical sequence
- Maintain clear narrative thread throughout
```

```
- Continuously advance plot without stagnation
- Avoid generalized event summaries or character abstractions
- Incorporate key character dialogue appropriately
- Use accessible language: "really," "and," "how," "if," "because," "but"

Prohibited Elements:
- Analytical reflections or thematic summaries
- Macro-level content summarization
- Theme or character development overviews
- Generalized terminology: "The event shows how...", "showing the...",
  "episodes...", "reveals...", "demonstrates..."
- Direct mention of "event" within body text
- Literary analysis or interpretive commentary

Required Elements:
- Tight pacing with smooth scene transitions
- Constant plot advancement and momentum
- Strong character development and interaction
- Factual content delivery without speculation

CLOSING SECTION (Final Sentences):
Objective: Thematic elevation without content summarization
- Refine deeper meaning and significance
- Sublimate central themes naturally
- Avoid high-level content recapping
- Connect to broader implications or impact
- Maintain narrative flow to conclusion

Quality Assurance Standards:
- Grammatical accuracy and professional language use
- Consistent third-person perspective maintenance
- Factual verification of all technical specifications
- Character attribution accuracy verification
- Prohibition compliance monitoring for banned phrases
- Timeline coherence and logical progression validation
```

### A.8.4 COMMENTARYCONTENTGENERATOR

The CommentaryContentGenerator creates commentary content from text source materials with specialized formatting for video presentations. It employs a dual-agent architecture to transform source texts into structured video-ready narratives while maintaining user-specified creative requirements.

Listing 33: CommentaryContentGenerator Structure

```
Input: user_requirements, source_text_path, commentary_presentation_style
Output: video_scene_json

Algorithm:
1. Source text processing:
   a. Load source text with multi-encoding support (utf-8, gb18030, gbk, etc.)
   b. Validate content size and apply truncation if exceeding 30K characters
   c. Generate text statistics for processing optimization
   d. Handle encoding fallbacks with binary reading approach

2. Presenter agent processing with embedded prompts:
   System: "You are an experienced expert in writing review copy.
   Pay special attention to user's words count requirements."

   User prompt template:
```

```
    "Create narration copy, strictly following user's ideas and
    presentation methods.

    User idea: '{user_idea}'
    Grounded text content: {content}
    Follow this presentation method: {present_content}

    Requirements:
    1. Format: Response in source language, remove chapter numbers,
       max 3 commas per sentence
    2. Content: Follow word count requirements, use original dialogues,
       focus on plot elements
    3. Language: Clear narrative flow, no user requirement mentions"

3. Judger agent formatting with embedded prompts:
   System: "You are a content formatting specialist with expertise
   in following guidelines"

   User prompt template:
   "Format content with requirements:
   - Start each sentence with /////
   - Remove chapter numbers and punctuation after segmentation
   - Keep original content, separate each sentence
   - Purpose: sentence-level segmentation

   Example: /////\nsentence one \n\n/////\nsentence two"

4. Visual scene description generation:
   a. Process formatted content for scene inference
   b. Apply scene description generation with embedded prompt:
      "Convert to English visual-scene descriptions:
       - Keep ///// markers unchanged
       - Max 1 sentence per section
       - Replace low-quality descriptions with conflict scenes
       - Describe character appearances when names mentioned
       - Generate high-ignition story moments"

5. Content validation and structuring:
   a. Count content sections using ///// markers
   b. Validate scene coherence and visual consistency
   c. Generate structured JSON output:
      - reqs: user requirements
      - content_created: formatted narrative content
      - segment_scene: visual scene descriptions
   d. Save to video_scene.json with UTF-8 encoding

6. Quality assurance and reporting:
   a. Display section statistics and preview
   b. Validate content structure integrity
   c. Report processing completion status

Error Handling:
- Multi-encoding file reading with graceful degradation
- Content truncation for oversized inputs (30K limit)
- API retry logic with exponential backoff (5 attempts)
- Fallback formatting for agent processing failures
- Binary reading fallback for encoding issues
```

A.8.5 COMMENTARY PRESENTATION STYLE SYSTEM PROMPT

This subsection provides comprehensive guidelines for creating engaging narrative content with proper structure, pacing, and audience engagement techniques.

Listing 34: Content Writing Methodology

```
OPENING SECTION: First three sentences with compelling attraction, quickly
↪ highlighting core elements

MAIN CONTENT SECTION (Majority of content, maintain plot progression in latter
↪  half, strictly adhere to word count requirements):
- Unfold according to plot timeline with clear main thread
- Smooth chapter transitions, continuous plot advancement
- Avoid generalizing events and characters
- Appropriately use key character dialogue
- Tight pacing, smooth scene transitions, continuous plot development,
↪ distinct character portrayal
- Maximum 3 commas per sentence

PROHIBITED ELEMENTS:
- Literary analysis, reflection, and thematic summaries
- Macro-level generalizations about story themes or character development
- Forbidden phrases: "The story shows...", "demonstrates...", "plot...", "
↪ reveals...", "someone's story..."
- Cannot mention "story" in main content

OPENING SUMMARY SECTION:
First three sentences provide content overview, followed by 2-3 introductory
↪ sentences before narrative begins

Example 1: Situational and Immersive Openings
Identity Immersion:
"His name is Wang Xiaoming. You think he's just an ordinary college student?
↪ No. He's a time traveler from the future, shouldering the mission to change
↪  history."

Knowledge-Guided:
"Do you know what the world's most dangerous job is? Turns out it's
↪ maintenance engineers at Antarctic research stations, dancing with death
↪ daily in minus 70-degree environments."

Hypothetical Scenarios:
"If someone gave you 100 million dollars, but the condition was never using
↪ social media again, would you accept? For modern people, this might be the
↪ cruelest multiple choice question."

Role-Playing:
"If you were a newly graduated medical student suddenly facing an
↪ unprecedented pandemic outbreak, what would you do? This was exactly Dr. Li
↪ 's situation in early 2020."

Time-Travel Hypothetical:
"If you traveled back to the Titanic in 1912, knowing the ship would sink in
↪ four days, how would you save over 2000 lives without being labeled insane
↪ ?"

Example 2: Theme-First Openings
Real Event Revelation:
```

```
"This isn't fiction; this is real. An engineer was trapped in Mount Everest's
↪ death zone for 72 hours straight-no oxygen, no supplies. Doctors said he
↪ couldn't survive. However, a miracle happened."

Ability Paradox:
"If one day you could suddenly see others' death times, this wouldn't be a
↪ superpower but torture. Julian possesses this 'curse,' watching his lover's
↪  countdown decrease daily while remaining powerless."

Expectation Reversal:
"He's two meters tall, muscular, with piercing eyes. Is this a professional
↪ boxer? No, he's a world-famous children's book author whose gentle stories
↪ have accompanied generations."

Extreme Situations:
"Minus 40 degrees, no food, pitch black, with only aurora as light source. How
↪  does one survive 30 days in Arctic wilderness? This was extreme explorer
↪ Zhang San's challenge, and his choices completely changed survival theory."

Suspenseful Questions:
"What could make a billionaire abandon all wealth to live as a hermit in deep
↪ mountains for 20 years? Today, we reveal the shocking truth behind this
↪ Silicon Valley legend's decision."

MAIN CONTENT SECTION (Majority of content, maintain plot progression in latter
↪  half):
Unfold according to plot timeline with clear main thread, ensuring continuous
↪ advancement in latter sections

PROHIBITED ELEMENTS:
- Literary analysis, reflection, and thematic summaries
- Macro-generalizations about story themes or character development
- Forbidden phrases: "The story shows...", "demonstrates...", "plot...", "
↪ reveals...", "someone's story..."
- Cannot mention "story" in main content

REQUIRED TRANSITION WORDS:
Use common transitional words: "unexpectedly," "suddenly," "turns out," "but,"
↪  "however," "yet," "as a result," "finally," "until," "if," "while," "
↪ indeed," "discovered," "only," "surprisingly," "afterward," "exactly," "not
↪  only," "nevertheless," "little did they know," "moreover," "of course," "
↪ because," "therefore," etc.

NARRATIVE TECHNIQUES:
- Dense use of transition words with high frequency of "but," "yet," "however"
- Unexpected twists for dramatic effect using "turns out," "little did they
↪ know"
- Plot advancement after each transition bringing new developments
- Concise action descriptions with short sentences for complex actions
- Sensory descriptive words enhancing immersion
- Frequent emotional adjectives
- Emphatic words highlighting characteristics
- Contrasting words creating dramatic tension
- Narrative connective words for smooth scene transitions
- Conversational tone with natural, approachable language
- Short sentences for rapid pacing at key moments
- Repetitive sentence structures for emphasis
- Rhetorical devices including metaphors and hyperbole
- Strategic suspense placement
```

```
3672    - Dramatic contrast throughout
3673
3674    CLOSING SECTION (Final sentence):
3675    Extract deeper meaning, elevate themes without mentioning "story ending" or "
3676    ↪ story beginning" to avoid breaking audience immersion
3677
```

3678

### A.8.6 TTSWRITER

The TTSWriter agent provides intelligent text rewriting capabilities for creating derivative video content based on sliced audio transcripts. It employs a dual-LLM approach using Claude for creative content generation and DeepSeek for precise text extraction, enabling coherent content transformation while maintaining original speech patterns and timing constraints.

Listing 35: TTSWriter Structure

```
Input: user_requirements, audio_path
Output: rewritten_speech_path

Algorithm:
1. Input validation and file path resolution:
   a. Validate user requirements and audio file path
   b. Generate lab_path: audio_filename + '.lab' extension
   c. Extract slice_lab_dir from audio path without extension
   d. Verify transcription file availability and accessibility

2. Transcript aggregation and segmentation analysis:
   a. Scan slice_lab_dir for .lab files with sorted ordering
   b. Load each lab file content with UTF-8 encoding
   c. Aggregate slice_lab array with individual segment transcripts
   d. Load complete original transcript from main .lab file
   e. Generate structured output format for LLM processing

3. Creative content generation with Claude LLM:
   System context: "Parody text recreation expert, specializing in generating
   text suitable for new scenarios"

   User prompt structure:
   "Perform creative recreation based on the following points and user
   requirements:
   1. Ensure smooth text flow between slices while recreating each slice
   2. Imitate the language style and sentence structure of each slice,
      only replace specific content
   3. When replacing original text vocabulary, length variation should
      not exceed two characters

   User requirements: {user_requirements}
   Original text: {original_transcript}
   Output format: {structured_format}"

4. Structured content generation workflow:
   a. Format template creation for each audio slice:
      "{index}. Original slice content: {original_content}\n   Recreation:"
   b. Apply user requirements to content transformation guidelines
   c. Maintain linguistic style and sentence structure consistency
   d. Enforce character length constraints (+-2 characters maximum)
   e. Generate comprehensive rewritten content with original context

5. Text extraction and refinement with DeepSeek LLM:
   System context: "Text extraction expert"
```

```
   User prompt structure:
   "Extract the **recreation** content of each slice and output line by line
   Output format:
   Recreation of slice 1
   Recreation of slice 2

   Text to be extracted: {generated_content}"

6. Dual-stage output processing and file generation:
   a. Save raw generation output to 'raw_speech.txt'
   b. Apply extraction LLM for clean content isolation
   c. Generate final 'speech.txt' with line-separated segments
   d. Ensure UTF-8 encoding for multilingual content support
   e. Return speech_path for downstream TTS processing

Content Transformation Guidelines:
- Linguistic style preservation: Maintain original speech patterns
- Structural consistency: Preserve sentence construction and rhythm
- Length constraints: Maximum +-2 character variation per replacement
- Semantic adaptation: Transform content while preserving meaning flow
- Temporal alignment: Ensure rewritten segments fit original timing

Quality Assurance Features:
- Dual-LLM validation for content accuracy and extraction precision
- Structured format enforcement for consistent processing
- Character-level length monitoring for TTS compatibility
- Cultural context preservation for appropriate content adaptation
- Error handling for LLM API failures with graceful degradation

Error Handling:
- Lab file existence validation with detailed error reporting
- UTF-8 encoding enforcement for international character support
- LLM API failure detection with exception management
- File I/O error handling for transcript processing
- Directory structure validation for slice organization
```

### A.8.7 SVCADAPTER

The SVCAdapter agent provides sophisticated lyrics adaptation capabilities for music cover creation, maintaining original melody structure while enabling creative content transformation. It employs a multi-stage approach using dual-LLM processing for high-quality lyrical recreation with precise character count alignment and structural preservation.

Listing 36: SVCAdapter Structure

```
Input: user_requirements, midi_analysis_path, song_name
Output: adapted_lyrics_string

Algorithm:
1. Input validation and MIDI analysis processing:
   a. Validate user requirements and file path accessibility
   b. Load MIDI analysis JSON with UTF-8 encoding
   c. Extract original lyrics text from analysis data
   d. Initialize song name for output file generation

2. Lyrics structure parsing and segmentation:
   Function parse_lyrics_structure(lyrics):
   a. Split lyrics by 'AP' delimiter markers for phrase separation
```

```
   b. Create structure array with "LYRICS" placeholders and "AP" markers
   c. Extract lyrics_parts array containing actual lyrical content
   d. Maintain temporal alignment between structure and content

3. Template generation for controlled adaptation:
   Function generate_lyrics_template(lyrics_parts):
   a. Create numbered segments with original lyrics content
   b. Calculate character count constraints for each segment
   c. Generate structured format:
      "{index}. Original lyrics segment: {content}
       Character limit: {count}
       Recreation:"
   d. Ensure precise character count preservation

4. Full lyrics generation with Claude LLM:
   System context: "Professional lyrics adaptation AI for high-quality
↪  recreation"

   User prompt structure:
   "Perform recreation based on following points and user requirements:
   1. Strictly follow character count limits for each recreation segment
   2. Focus on rhyme and rhythm while maintaining narrative flow
   3. Ensure complete semantic integrity in lyrical content
   4. Use reasonable word combinations and inter-segment rhyming

   User requirements: {user_requirements}
   Original lyrics: {original_lyrics}
   Output format: {template}"

5. Content extraction and refinement with DeepSeek LLM:
   System context: "Lyrics extraction expert"

   User prompt structure:
   "Extract the **recreation** content from each lyrics segment:
   Output format:
   Recreation of segment 1
   Recreation of segment 2

   Text to extract: {generated_lyrics}"

6. Character count alignment and quality assurance:
   Function align_extract_parts():
   a. Compare extract_parts length with lyrics_parts requirements
   b. For mismatched character counts (max 5 retry attempts):
      - Generate context-aware alignment prompts with surrounding segments
      - Apply Claude LLM for precise character count correction
      - Maintain semantic coherence with previous/next segments
   c. Apply fallback padding/truncation if alignment fails:
      - Add Chinese character "la" characters for insufficient length
      - Truncate excess characters while preserving meaning

7. Structure reconstruction and output generation:
   a. Merge aligned lyrics parts with original structure markers
   b. Replace "LYRICS" placeholders with adapted content segments
   c. Preserve "AP" delimiters for proper phrase separation
   d. Generate multiple output formats:
      - raw_lyrics.txt: initial generation output
      - lyrics.txt: line-separated adapted segments
      - script.txt: complete restructured lyrics
```

```
        - {song_name}_cover.json: updated MIDI analysis with new lyrics

Technical Specifications:
- Character preservation: Exact count matching for musical timing
- Structural integrity: AP delimiter preservation for phrase boundaries
- Multi-format output: TXT and JSON formats for different use cases
- Quality assurance: Iterative refinement with context-aware correction
- Semantic coherence: Meaning preservation across segment boundaries

Adaptation Features:
- Rhyme scheme preservation with enhanced creativity
- Rhythmic pattern maintenance for musical compatibility
- Context-aware segment generation with surrounding lyrical flow
- Automated character count correction with intelligent padding
- Multi-stage validation for lyrical quality and structural accuracy

Error Handling:
- JSON parsing validation with encoding error management
- Character count mismatch detection with automated correction
- LLM API failure handling with graceful degradation
- File I/O error management for multiple output formats
- Structure length validation with detailed error reporting
```

### A.8.8 STANDUPADAPTER

The StandUpAdapter agent provides sophisticated comedy script adaptation capabilities for transforming reference content into structured stand-up comedy performances. It employs Claude LLM for creative content generation while maintaining professional comedy formatting standards including tone markers and atmosphere cues for enhanced delivery guidance.

Listing 37: StandUpAdapter Structure

```
Input: user_requirements, reference_script_path
Output: segmented_standup_script

Algorithm:
1. Input validation and reference script loading:
   a. Validate user requirements and script file path
   b. Load reference script content with UTF-8 encoding
   c. Extract data directory path for output file placement
   d. Strip whitespace and prepare content for adaptation

2. Stand-up comedy format specification:
   a. Define mandatory tone markers:
      - [Natural]: conversational delivery style
      - [Confused]: questioning or bewildered tone
      - [Empathetic]: understanding or sympathetic delivery
      - [Exclamatory]: energetic or surprised expression
   b. Configure atmosphere cues for audience interaction:
      - [Laughter]: comedic punchline moments
      - [Cheers]: celebratory or triumphant points

3. Creative adaptation with Claude LLM:
   System context: "Professional stand-up comedy adaptation specialist"

   User prompt structure:
   "Adapt the following reference script into stand-up comedy format.

   Content to adapt: {reference_script}
```

```
    Additional requirements: {user_requirements}

    Format specifications:
    1. Each line must begin with tone markers: [Natural] [Confused]
       [Empathetic] [Exclamatory]
    2. Add atmosphere cues [Laughter] or [Cheers] at key moments
    3. Keep each line independent with consistent structure

    Important notes:
    - NO titles, introductions, or conclusions
    - Preserve core humor while localizing cultural references
    - Incorporate English stand-up linguistic features and rhythm
    - Use atmosphere cues sparingly (3-4 total)
    - Generate 3-5 minute performance script"

4. Comedy structure optimization:
   a. Preserve original humor essence while adapting format
   b. Localize cultural references for target audience
   c. Incorporate stand-up comedy linguistic patterns:
      - Timing-based humor with pause indicators
      - Observational comedy structure
      - Callback references and running gags
   d. Apply rhythm and pacing appropriate for live performance

5. Tone marker integration and line formatting:
   a. Ensure every line begins with appropriate tone marker
   b. Match tone markers to content emotional context
   c. Structure independent lines for performance flexibility:
      "[Tone]content...\n[Tone]content...[Atmosphere]"
   d. Maintain comedic flow between individual segments

6. Atmosphere cue placement and timing:
   a. Identify punchline moments for [Laughter] placement
   b. Recognize triumph or celebration points for [Cheers]
   c. Apply strategic placement for maximum comedic impact:
      - Immediately after dialogue delivery
      - At peak comedic tension release points
   d. Limit to 3-4 cues total to avoid oversaturation

7. Output generation and file management:
   a. Generate complete stand-up script with title
   b. Format output structure:
      "# title\n[Tone]content...\n[Tone]content...\n..."
   c. Save to 'stand-up.txt' in reference script directory
   d. Return script string for immediate use or further processing

Technical Specifications:
- Content adaptation: Reference-based creative transformation
- Format compliance: Strict tone marker and atmosphere cue requirements
- Performance duration: 3-5 minute script generation target
- Linguistic adaptation: English stand-up comedy style integration
- Cultural localization: Reference adaptation for target audience

Comedy Adaptation Features:
- Humor preservation with format transformation
- Cultural reference localization for audience relevance
- Professional stand-up delivery guidance through tone markers
- Audience interaction planning with atmosphere cues
- Independent line structure for performance flexibility
```

```
Error Handling:
- Reference script file validation with encoding verification
- Claude LLM API error detection with graceful failure handling
- Output file creation with directory permission management
- UTF-8 encoding preservation for international character support
- Exception isolation with detailed error reporting for debugging
```

### A.8.9    STANDUPSYNTH

The StandUpSynth agent provides comprehensive audio synthesis for stand-up comedy perfor-
mances, combining advanced text-to-speech generation with audience reaction integration. It employs
CosyVoice2 for high-quality speech synthesis and implements intelligent script parsing to create
realistic comedy performances with appropriate vocal delivery and audience responses.

Listing 38: StandUpSynth Structure

```
Input: segmented_script, target_vocal_directory, reaction_directory
Output: merged_audio_path, segment_directory, metadata_path

Algorithm:
1. Input validation and environment setup:
   a. Validate script content and directory paths
   b. Convert target_vocal_dir and reaction_dir to absolute paths
   c. Navigate to tools/CosyVoice directory for model execution
   d. Initialize CosyVoice2 model with pretrained weights:
      - Model: CosyVoice2-0.5B for high-quality synthesis
      - Configuration: load_jit=False, load_trt=False, fp16=False

2. Script parsing and segment analysis:
   a. Split script by newlines and filter empty lines
   b. Skip first line (title) and process content lines
   c. Create segment directory structure for individual audio files
   d. Initialize counters and result arrays for processing tracking

3. Intelligent script analysis with DeepSeek LLM:
   System context: "Stand-up comedy segment analyzer"

   User prompt structure:
   "Analyze tone, text content, and atmosphere marker:
   {comedy_line}

   Output strictly in JSON format:
   1. tone: 'Natural', 'Empathetic', 'Confused', or 'Exclamatory'
   2. text: segment content without markers
   3. reaction: 'Laughter' or 'Cheers' (only if atmosphere marker present)
   4. Strict reliance on existing markers, no interpretation
   5. NO extra characters or explanations"

4. JSON response processing and validation:
   a. Strip markdown code block markers (```json, ```)
   b. Parse JSON response with error handling
   c. Extract tone, text, and optional reaction fields
   d. Validate tone values against supported categories
   e. Handle malformed JSON with graceful error reporting

5. Voice synthesis with tone-specific prompts:
   a. Load corresponding prompt files:
      - Prompt text: {tone}.lab file for speech context
```

```
      - Prompt audio: {tone}.wav file for voice characteristics
   b. Apply CosyVoice2 zero-shot synthesis:
      - Input: parsed text content
      - Context: tone-specific prompt text and audio
      - Output: high-quality speech audio at model sample rate
   c. Save individual segments: {segment_index}.wav

6. Audience reaction integration:
   a. Check for reaction field in parsed JSON
   b. Load corresponding reaction audio: {reaction}.wav
   c. Combine speech and reaction using AudioSegment:
      - Load original synthesized speech
      - Load reaction audio (laughter/cheers)
      - Concatenate: original_audio + reaction_audio
      - Export combined audio overwriting original segment

7. Audio consolidation and final merge:
   Function merge_audio_files():
   a. Initialize silent AudioSegment for accumulation
   b. Iterate through numbered segment files (0.wav to cnt.wav)
   c. Load and concatenate each segment with error handling
   d. Export final merged audio to ../final/stand_up.wav
   e. Return absolute path for downstream processing

8. Metadata generation and cleanup:
   a. Compile processing results into structured JSON
   b. Save metadata: tone, text, and reaction information per segment
   c. Export to stand-up.json in target vocal directory
   d. Restore original working directory after processing
   e. Return comprehensive output paths and metadata

Technical Specifications:
- Speech synthesis: CosyVoice2-0.5B with zero-shot voice cloning
- Audio processing: PyDub for segment manipulation and concatenation
- Script parsing: DeepSeek LLM for intelligent content analysis
- Output format: WAV files with model native sample rate
- Metadata: JSON structure with segment-level processing information

Comedy Performance Features:
- Tone-specific voice synthesis for natural delivery variation
- Audience reaction integration for realistic performance atmosphere
- Segment-based processing for precise timing control
- Automatic tone detection from script markers
- Professional audio quality with seamless segment transitions

Error Handling:
- CosyVoice2 model loading validation with detailed error reporting
- JSON parsing with markdown cleanup and format validation
- Audio file loading verification for segments and reactions
- Graceful handling of missing reaction files or invalid tones
- Comprehensive exception management with processing continuation
- Working directory restoration for environment consistency
```

### A.8.10 CROSSTALKADAPTER

The CrossTalkAdapter agent provides specialized script adaptation capabilities for transforming reference content into traditional Chinese crosstalk (xiangsheng) dialogue format. It employs Claude LLM

for cultural localization and dialogue structure optimization while maintaining authentic crosstalk performance characteristics including role-specific delivery patterns and traditional interactive elements.

Listing 39: CrossTalkAdapter Structure

```
Input: user_requirements, reference_script_path, dou_gen_directory,
↪ peng_gen_directory
Output: segmented_crosstalk_script

Algorithm:
1. Input validation and role configuration:
   a. Validate user requirements and reference script file path
   b. Extract performer directories for role-specific voice synthesis
   c. Generate role names from directory basenames:
      - dou_gen_name: comic lead performer (funny guy)
      - peng_gen_name: straight man performer (setup guy)
   d. Load reference script content with UTF-8 encoding

2. Traditional crosstalk role definition:
   a. Configure dou_gen (funny guy) characteristics:
      - Comic lead role delivering main jokes
      - Drives narrative progression and humor development
      - Primary responsibility for punchline delivery
   b. Configure peng_gen (setup guy) characteristics:
      - Straight man role providing reactions and setup
      - Creates opportunities for comic lead responses
      - Maintains dialogue rhythm and audience engagement

3. Cultural adaptation with Claude LLM:
   System context: "Professional cross talk (xiang sheng) adaptation
↪ specialist"

   User prompt structure:
   "Adapt English stand-up comedy material into authentic traditional
   Chinese crosstalk dialogue format.

   Material to adapt: {reference_script}

   Crosstalk roles:
   - {dou_gen_name}: Comic lead (funny guy), delivers main jokes and drives
↪ narrative
   - {peng_gen_name}: Straight man (setup guy), reacts and plays off comic
↪ lead

   Format Requirements:
   1. Each performer's lines on separate lines starting with their name
   2. Begin each line with tone marker: [Natural] [Confused] [Emphatic]
   3. Same tone should not appear consecutively for more than two lines

   Additional requirements: {user_requirements}"

4. Dialogue structure optimization:
   a. Implement traditional crosstalk format requirements:
      - Line-by-line role alternation for dynamic interaction
      - Tone marker integration: [Natural], [Confused], [Emphatic]
      - Consecutive tone limitation to prevent monotony
   b. Preserve cultural authenticity through:
      - Traditional crosstalk speech patterns and rhythm
      - Common xiangsheng phrases and interactive elements
      - Cultural reference localization for Chinese audience
```

```
5. Performance format standardization:
   a. Apply consistent dialogue structure:
      "[tone] Role_name: dialogue_content"
   b. Ensure proper role distribution between performers:
      - Balanced dialogue allocation
      - Appropriate setup-punchline timing
      - Natural conversational flow maintenance
   c. Integrate traditional crosstalk elements:
      - Call-and-response patterns
      - Misunderstanding-based humor
      - Cultural wordplay and linguistic jokes

6. Content localization and cultural adaptation:
   a. Transform English humor into Chinese comedy traditions
   b. Adapt cultural references for Chinese audience familiarity
   c. Incorporate traditional crosstalk performance elements:
      - Verbal sparring between performers
      - Audience-directed asides and commentary
      - Regional dialect influences and speech patterns
   d. Maintain original humor essence while ensuring cultural relevance

7. Output generation and file management:
   a. Generate complete crosstalk script with title header
   b. Format output structure: "Title\n[tone] Role: content...\n..."
   c. Save to 'cross-talk.txt' in reference script directory
   d. Return script string for immediate use or further processing
   e. Ensure UTF-8 encoding preservation for Chinese character support

Technical Specifications:
- Content adaptation: Reference-based cultural transformation
- Role management: Dual-performer dialogue optimization
- Format compliance: Traditional crosstalk structure requirements
- Cultural localization: English-to-Chinese humor translation
- Performance guidance: Tone markers for delivery optimization

Crosstalk Adaptation Features:
- Authentic xiangsheng dialogue patterns and timing
- Cultural reference localization for target audience
- Traditional performer role dynamics (funny guy/setup guy)
- Balanced dialogue distribution for engaging performance
- Professional comedy structure with setup-punchline optimization

Error Handling:
- Reference script file validation with encoding verification
- Claude LLM API error detection with graceful failure handling
- Output file creation with directory permission management
- UTF-8 encoding preservation for Chinese character integrity
- Exception isolation with detailed error reporting for debugging workflow
```

### A.8.11 CROSSTALKSYNTH

The CrossTalkSynth agent provides comprehensive audio synthesis for crosstalk performances, combining dual-performer voice generation with intelligent dialogue analysis. It employs CosyVoice2 for high-quality speech synthesis and DeepSeek LLM for script parsing to create authentic crosstalk performances with role-specific vocal characteristics and appropriate delivery styles.

Listing 40: CrossTalkSynth Structure

```
Input: segmented_script, funny_guy_directory, setup_guy_directory
Output: merged_audio_path, segment_directory, metadata_path

Algorithm:
1. Input validation and role setup:
   a. Validate script content and performer directory paths
   b. Convert directories to absolute paths for cross-platform compatibility
   c. Extract performer names from directory basenames:
      - dou_gen_name: funny guy performer (comic lead)
      - peng_gen_name: setup guy performer (straight man)
   d. Initialize CosyVoice2 model in tools/CosyVoice environment

2. CosyVoice2 model initialization:
   a. Navigate to CosyVoice working directory for proper execution
   b. Load pretrained model: CosyVoice2-0.5B with configuration:
      - load_jit=False, load_trt=False, fp16=False
      - High-quality synthesis settings for crosstalk dialogue
   c. Handle model loading errors with graceful failure reporting

3. Script parsing and line-by-line analysis:
   a. Split script by newlines and filter empty content
   b. Skip title line (first line) and process dialogue content
   c. Create segment directory for individual audio file storage
   d. Initialize counters and result arrays for processing tracking

4. Intelligent dialogue analysis with DeepSeek LLM:
   System context: "Crosstalk dialogue line analyzer"

   User prompt structure:
   "Analyze crosstalk dialogue line for performer role, tone, text content:
   {dialogue_line}

   Output JSON format with STRICT rules:
   1. role: either {funny_guy_name} or {setup_guy_name}
   2. tone: 'Natural', 'Emphatic', or 'Confused'
   3. text: dialogue content without formatting
   4. reaction: 'Laughter' or 'Cheers' (only if markers present)
   5. NO extra characters before/after JSON

   Example: {'role': '{performer}', 'tone': 'Natural', 'text': '...'}"

5. JSON response processing and validation:
   a. Strip markdown code block markers (```json, ```)
   b. Parse JSON response with comprehensive error handling
   c. Extract role, tone, text, and optional reaction fields
   d. Validate performer names against configured roles
   e. Handle malformed JSON with graceful error recovery

6. Role-specific voice synthesis:
   a. Load performer-specific prompt files:
      - Prompt text: {data_dir}/{role}/{tone}.lab
      - Prompt audio: {data_dir}/{role}/{tone}.wav at 16kHz
   b. Apply CosyVoice2 zero-shot inference:
      - Input: extracted dialogue text
      - Context: role and tone-specific prompts
      - Output: high-quality speech synthesis
   c. Save individual segments: {segment_index}.wav
```

```
7. Audio consolidation and performance assembly:
   Function merge_audio_files():
   a. Initialize silent AudioSegment for sequential concatenation
   b. Load numbered audio segments (0.wav to cnt.wav)
   c. Concatenate segments maintaining dialogue flow:
      - Preserve timing between performer exchanges
      - Maintain audio quality throughout concatenation
   d. Export final performance: ../final/cross_talk.wav

8. Metadata generation and workflow completion:
   a. Compile processing results into structured JSON metadata
   b. Include role, tone, text, and reaction information per segment
   c. Export metadata to cross-talk.json in data directory
   d. Restore original working directory after processing
   e. Return comprehensive output paths for downstream processing

Technical Specifications:
- Speech synthesis: CosyVoice2-0.5B with zero-shot voice cloning
- Audio processing: PyDub for segment manipulation and merging
- Script analysis: DeepSeek LLM for intelligent dialogue parsing
- Role management: Dual-performer voice synthesis with tone variation
- Output format: WAV files with model native sample rate

Crosstalk Performance Features:
- Dual-performer role-specific voice synthesis
- Tone-aware delivery variation (Natural, Emphatic, Confused)
- Authentic crosstalk dialogue timing and rhythm
- Professional audio quality with seamless performer transitions
- Intelligent script parsing for automated role and tone detection

Error Handling:
- CosyVoice2 model initialization validation with detailed error reporting
- JSON parsing with markdown cleanup and format validation
- Audio file loading verification for role-specific prompts
- Graceful handling of missing prompt files or invalid roles
- Comprehensive exception management with processing continuation
- Working directory restoration for environment consistency
```

### A.8.12 VIDEOCONVERSION

The VideoConversion agent transforms audio content with JSON timestamps into comprehensive visual scene descriptions optimized for video generation workflows. It processes timestamped audio segments and converts narrative content into actionable visual scenes through intelligent content analysis.

Listing 41: VideoConversion Structure

```
Input: timestamp_path
Output: video_scene_path

Algorithm:
1. Timestamp file processing:
   a. Load JSON timestamp data from cut_points.json
   b. Validate file structure and content availability
   c. Extract content segments from sentence_data.chunks array
   d. Handle file format validation and error reporting

2. Content formatting and preparation:
   a. Extract content from each timestamped chunk
```

```
b. Format content with ///// separators between segments
c. Create unified content string: "/////\n" + join(contents)
d. Validate segment count and content integrity

3. LLM-based scene description generation with embedded prompts:
   System prompt: "You are a visual scene descriptor. Follow exact
   requirements:
   - Keep number of ///// marks unchanged
   - Deduce English visual-scene descriptions only
   - Keep same sentence separators and spacing
   - Max 1 sentence per scene section
   - Don't directly translate sentences
   - Describe character appearances when names mentioned"

   User prompt template:
   "Content to process: {content}

   Example Input: /////\n[Emily] and [Jackson] stood together, ocean
   breeze ruffling hair, surrounded by vastness of ocean.
   \n\n/////\nLeader increased Xiao Wang's business freedom.

   Example Output: /////\nRed hair girl Emily and brown hair boy Jackson
   standing on sunset seaside with wind-blown hair
   \n\n/////\nWhite t-shirt young employees in office environment"

4. Scene description processing:
   a. Apply LLM processing with DeepSeek model
   b. Generate visual scene keywords and descriptions
   c. Maintain temporal alignment with original timestamps
   d. Preserve character appearance details and environmental context

5. Output structuring and validation:
   a. Create structured JSON with segment_scene field
   b. Preserve original content_created for reference
   c. Validate scene coherence and visual consistency
   d. Save to video_scene.json in scene_output directory

6. Quality assurance and reporting:
   a. Verify segment count matches input timestamps
   b. Display processing statistics and completion status
   c. Handle error cases with detailed error messaging
   d. Return structured output for downstream processing

Error Handling:
- Timestamp file existence validation
- JSON structure validation with KeyError handling
- Empty content segment detection and reporting
- Directory creation with proper permissions
- LLM response parsing with fallback mechanisms
```

### A.8.13 VIDEOQA

The VideoQA agent provides comprehensive video content analysis through automated transcription and interactive question-answering capabilities. It processes multiple videos simultaneously, generates combined transcripts, and enables users to query video content through natural language interactions.

Listing 42: VideoQA Structure with System Prompt

```
Input: video_directory, output_path, save_history_flag
```

```
Output: transcript_path, qa_session_results

Algorithm:
1. Initialize video processing environment:
   a. Load Whisper model for speech recognition
   b. Configure device (CUDA/CPU) and memory allocation
   c. Set up processing directories and file paths

2. Video discovery and validation:
   a. Scan directory for supported video formats
   b. Validate file integrity and accessibility
   c. Sort files for consistent processing order

3. Batch transcription pipeline:
   For each video file:
   a. Load video using Whisper pipeline
   b. Apply automatic speech recognition:
      - Chunk audio into 30-second segments
      - Process with batch size optimization
      - Extract timestamps and text content
   c. Format transcript with video identification
   d. Append to combined transcript buffer

4. Transcript consolidation:
   a. Merge individual transcripts with video markers
   b. Save combined transcript to designated path
   c. Generate processing statistics and metadata

5. Interactive Q&A session initialization:
   a. Load combined transcript for query processing
   b. Truncate content if exceeding API limits (50K chars)
   c. Initialize conversation history tracking

6. Q&A processing with embedded prompt:
   While user input != 'quit':
   a. Receive user question via command line interface
   b. Process question through QA agent with prompt:

   System Message: "You are a helpful assistant who answers questions
   about video content based strictly on the provided transcripts from
   multiple videos."

   User Prompt Template:
   "Answer the user's question carefully based only on the information
   contained in the video transcripts. The transcript contains content
   from multiple videos, each marked with '=== VIDEO: filename ==='.
   If the answer cannot be found in the transcripts, state that clearly.

   User's question: '{user_question}'

   Video transcripts content: {content}

   Requirements:
   1. Be concise and direct in your answer
   2. Only use information from the transcripts
   3. If the answer is not in the transcripts, say 'I don't have enough
      information from the videos to answer this question'
   4. When referencing information, mention which video file it came
      from if relevant
```

```
5. Don't make up information not present in the transcripts"

   c. Query LLM with constructed prompt and transcript context
   d. Generate factual answer based on video content
   e. Display answer and log interaction
   f. Update conversation history with timestamp

7. Session finalization:
   a. Save Q&A history to persistent storage
   b. Generate session summary statistics
   c. Return processing results and file paths

8. Error handling and recovery:
   - Fallback to existing transcripts on failure
   - Graceful session termination on interruption
   - Comprehensive logging for debugging
```

### A.8.14   VIDEOSUMMARIZATIONGENERATOR

The VideoSummarizationGenerator creates comprehensive video summarization content by processing video files through automatic speech recognition and applying user-specified presentation styles. It combines advanced transcription capabilities with intelligent content adaptation to generate structured video summaries.

Listing 43: VideoSummarizationGenerator Structure

```
Input: user_idea, video_directory, presentation_style_path, output_path
Output: content_output, status, processed_videos, transcript_source

Algorithm:
1. Whisper model initialization with retry logic:
   a. Load AutoModelForSpeechSeq2Seq from whisper-large-v3-turbo
   b. Configure device detection (CUDA/CPU) and dtype optimization
   c. Initialize AutoProcessor with tokenizer and feature extractor
   d. Create ASR pipeline with optimized parameters:
      - max_new_tokens=128, chunk_length_s=30, batch_size=16
      - generate_kwargs={"language": "en", "task": "transcribe"}

2. Content source processing:
   a. Analyze video_dir input type (file/directory/text)
   b. For video files:
      - Scan supported extensions (.mp4, .avi, .mov, .mkv, etc.)
      - Execute batch transcription with progress tracking
      - Combine transcripts with video identification markers
   c. For text files:
      - Load with multi-encoding support and validation
      - Handle binary reading fallback for encoding issues

3. Batch video transcription pipeline:
   For each video file:
   a. Apply Whisper ASR with retry logic (3 attempts)
   b. Process with chunk-based approach for large files
   c. Generate combined transcript with video markers:
      "=== VIDEO: filename.mp4 ===\n[transcript_content]"
   d. Handle transcription errors with graceful degradation

4. Presenter agent processing with embedded prompts:
   System: "You are an experienced expert in writing transcripts
   summarization. Pay special attention to user's words count
```

```
    requirements."

    User prompt template:
    "Create content summarization, strictly following user's ideas and
    presentation methods, answer using user's idea language.

    User's idea: '{user_idea}'
    Grounded text content: {content}
    Follow this presentation method: {present_content}

    Requirements:
    1. Format: Less point forms
    2. Content: Follow word count requirements, use original dialogues
    3. Language: Third-person perspective, clear narrative flow"

5. Content adaptation and optimization:
    a. Load presentation style from file or direct string input
    b. Apply content truncation for oversized inputs (15K limit)
    c. Generate statistics: word count, character count, source type
    d. Process through presenter agent with comprehensive retry logic

6. Output generation and validation:
    a. Create structured content output with user_idea and content_created
    b. Save to specified output path as plain text
    c. Generate processing metadata:
       - processed_videos: list of successfully transcribed files
       - transcript_source: source type classification
       - status: success/error with detailed error information

Error Handling:
- Comprehensive retry logic for all major operations (3-5 attempts)
- Exponential backoff with configurable wait times
- Graceful fallback for transcription failures
- Multi-encoding file reading with robust error recovery
- Device fallback for CUDA unavailability
- Path validation and directory creation
```

A.9 PROMPT OF INTENT ANALYSIS

To mitigate the memory burden on LLMs and achieve more precise intent-tool alignment, we conduct intent analysis of user requirements and provide a detailed prompt as presented in Listing 44. Additionally, VideoAgent adaptively refines its intent analysis based on the feedback provided during the self-reflection phase, with the corresponding prompt shown in Listing 45.

Listing 44: System Prompt of Intent Analysis

```
You are an intent analyst.
I will provide a set of candidate intents and a user's requirements.
Please select as many relevant candidate intents as possible that match the
↪ user's requirements. Consider factors such as keywords, audio types (speech
↪ , song, music, etc.), and other relevant dimensions.

Candidate intents:
{intents}

User requirements:
{reqs}

Please Only output pure List format:
```

```
['intent1', 'intent2', ...]

Note! Don't output any analysis and explanations!
```

Listing 45: System Prompt of Intent Analysis

```
You are an intent analyst.
Previous analysis attempt failed with the following reflection:
{reflection}

Previous selected intents:
{previous_intents}

Please re-analyze the user's requirements with the candidate intents below,
↪ considering the reflection.
Select as many relevant candidate intents as possible.

Candidate intents:
{intents}

User requirements:
{reqs}

Please Only output pure List format:
['intent1', 'intent2', ...]

Note! Don't output any analysis and explanations!
```

## A.10 GRAPH CONSTRUCTION RULES

We provide a comprehensive prompt for the agent graph, which guides the collaboration among tooluse agents and also allows for a high degree of flexibility in constructing agent workflows, not being limited to predefined pipelines, as presented in Listing 46.

Listing 46: System Prompt of Agent Graph

```
You are an Agent Graph Designer. I will provide User Requirement and
↪ Registered Agents (Name, Description and Parameters information):

Your task is to:
1. Judge Feasibility:
   - Evaluate implementation feasibility of User Requirements
   - Output: "Feasible" or "Infeasible" (strictly one of these)
2. Design Executable Agent Graph:
   - Format: List
   - Agent Graph shall contain metadata for each Agent Node including:
     * name: (string)
     * inputs: (list of input parameter objects with):
     * parameter: input parameter name
     * description: brief parameter description
     * outputs: (list of output parameter objects with):
     * parameter: output parameter name
     * description: brief parameter description
     * links: (list of dictionaries) where each dictionary specifies:
         - key of dictionaries: target agent name
         - value of dictionaries: target agent's input parameter name that
↪ this output connects to
3. Generate Agent Chain:
```

```
    - Format: List
    - Generate the Agent Chain based on the description of the Agent and the
↪ sequential information contained in the designed Agent Graph
4. Generate User Input Graph
    - Format: List
    - Parameter nodes with no in-degree (no incoming edges) are uniformly
↪ considered to require user input.
    - Parameter nodes with no in-degree may have different names but share the
↪ same user input, meaning a single user input parameter can point to
↪ multiple such nodes.
    - Parameter nodes with no in-degree that are linked to user input should be
↪  represented in the format **AgentName.input_parameter**
    - Generate the User Input Graph based on the Agent descriptions and
↪ parameter passing information in the designed Agent Graph.
5. Output Reasoning:
    - If Feasible, Provide concise reasoning (<200 words) explaining the entire
↪  workflow logic
    - If Infeasible, Specify exact failure reasons (<200 words)

In addition to the above formatting requirements, please also note the
↪ following:
1. For each element of **outputs** in each Agent Node:
    - Ensure that the **links** in the **outputs** point to an input parameter
↪ that actually exists in the next Agent Node.
    - The output parameter name does not need to match the input parameter name
↪  in the next Agent Node.
    - Ensure the output parameter's description and type match the input
↪ parameter requirements of the next Agent Node. For example, a file path
↪ output cannot be passed to a directory path input.
2. Final JSON Output Format Specification:
{
    "Feasibility": "Feasible" or "Infeasible",
    "Agent Graph": ...,
    "Agent Chain": ...,
    "User Input Graph": ...,
    "Reasoning": ...
}
Strictly follow JSON output format!
```

## A.11 TWO-STEP REVIEWER

During the self-reflection phase, we propose a two-step reviewer strategy to maximize the accuracy of the evaluation. The corresponding prompts can be found in Listing 47 and Listing 48 respectively.

Listing 47: System Prompt for Initial Evaluation

```
You are an agent graph validation system.
I will provide:
1. User Requirement
2. Registered agent metadata
3. Candidate Agent Graph
4. An Agent Chain derived from the Candidate Agent Graph
5. Required User Inputs

User Requirements:
{reqs}

Metadata of registered agents:
{tools}
```

```
Task: Evaluate the candidate agent graph:

Candidate Agent Graph:
{agent_graph}

Agent Chain:
{agent_chain}

Required User Inputs:
{user_inputs}

Evaluation Criteria:
1. Based on the Metadata of registered agents and parameter passing in the
↪ Agent Graph, determine from multiple aspects whether the user requirements
↪ can be fulfilled:
   - Execution sequence of agents in the Agent Graph
   - For parameter nodes with no incoming edges, they are uniformly considered
↪  as user inputs, but it is necessary to determine whether they should be
↪ provided by the user or by the parent agent
   - Validate that the necessary output parameters are correctly routed to the
↪  intended agent and the expected input parameters.
2. There should be no functionally redundant agents (e.g., repeatedly adding
↪ audio tracks to a video).
3. For vaguely mentioned requirements in user needs, lenient evaluation is
↪ acceptable. For example, if the user requests audio quality improvement, it
↪ 's sufficient as long as at least one relevant agent in the graph meets
↪ this requirement.

Please Only output pure JSON format:
{{
"Result": '0' if correct else '1',
"Reasoning": Concisely state the key reasons why a score of '0' or '1' was
↪ assigned (<100 words).
}}
```

Listing 48: System Prompt for Secondary Evaluation

```
You are an agent graph reflection system.
I will provide:
1. User Requirement
2. Registered agent metadata
3. Candidate Agent Graph
4. An Agent Chain and User Input Graph derived from the Candidate Agent Graph
5. Previous validation result

User Requirements:
{reqs}

Metadata of registered agents:
{tools}

Task: Evaluate the candidate agent graph:

Candidate Agent Graph:
{agent_graph}

Agent Chain:
{agent_chain}
```

```
Required User Input Graph:
{user_inputs}

Previous validation result:
{judge_res}

Reflection Task:
1. If the previous validation result is '0', please reflect on whether there
↪ were any overlooked aspects based on the **Evaluation Criteria** and the
↪ reasoning behind the previous validation result.
2. If the previous validation result is '1', please reflect on whether the
↪ reasoning behind the previous validation result was correct.

Evaluation Criteria:
1. Based on the Metadata of registered agents and parameter passing in the
↪ Agent Graph, determine from multiple aspects whether the user requirements
↪ can be fulfilled:
   - Execution sequence of agents in the Agent Graph
   - For parameter nodes with no incoming edges, they are uniformly considered
↪  as user inputs, but it is necessary to determine whether they should be
↪ provided by the user or by the previous agent
   - Validate that the necessary output parameters are correctly routed to the
↪  intended agent and the expected input parameters.
   - Validate that the output parameters' description and type match the input
↪  requirements of the next agent.
   - Not all output parameters are necessarily mapped to the input
↪ requirements of the next agent. Redundant output parameters may exist, but
↪ they should not interfere with the fulfillment of user requirements.
2. There should be no functionally redundant agents (e.g., repeatedly adding
↪ audio tracks to a video).
3. For vaguely mentioned requirements in user needs, lenient evaluation is
↪ acceptable. For example, if the user requests audio quality improvement, it
↪ 's sufficient as long as at least one relevant agent in the graph meets
↪ this requirement.

Please Only output pure JSON format:
{{
"Result": '0' if correct else '1',
"Reasoning": Concisely state the key reasons why a score of '0' or '1' was
↪ assigned (<100 words).
}}
```

### A.12 CASES OF MULTI-MODAL AGENT WORKFLOW

To demonstrate VideoAgent's high sensitivity to user requirements and its generalization capability, we also provide several constructed agent workflow cases. VideoAgent can be extensively customized to tailor agent workflows according to user needs. For example, as shown in Listing 49, it performs loudness normalization on audio. Users can further extend this functionality with additional audio processing tasks, such as vocal separation or adding background music. Moreover, VideoAgent supports customized audio and video creation, allowing users to freely propose diverse creative requirements, including voice cloning, song generation, and music video production.

Listing 49: User Requirement of Audio Preprocessing

```
User Requirement: I need to extract the audio from a video file, improve its
↪ sound quality by normalizing the loudness, and then mix it with some
↪ background music. The result should be a clean audio track that balances
↪ the original audio with background music.
```

Listing 50: User Requirement of Audio Preprocessing

```
{
"Feasibility": "Feasible",
"Intent List": ["Loudness Normalization", "Add BGM", "Audio Optimization"],
"Agent Graph": [
    {"node": "LoudnessNormalizer", "inputs": [{"name": "data_dir", "
↪ description": "Directory of audio files to be normalized"}], "outputs": [{"
↪ name": "status", "description": "Execution status (success/error)", "links
↪ ": []}]},
    {"node": "Mixer", "inputs": [{"name": "bgm_path", "description": "Path to
↪ the BGM file"}, {"name": "audio_path", "description": "Audio to be mixed
↪ into the background music"}], "outputs": [{"name": "audio_path", "
↪ description": "Path to the synthesized audio", "links": []}]}
],
"Agent Chain": ["LoudnessNormalizer", "Mixer"],
"User Input Graph": [
    {"node": "raw_audio_directory", "description": "Directory containing the
↪ user's raw audio recording that needs to be normalized", "links": [{"
↪ LoudnessNormalizer": "data_dir"}]},
    {"node": "normalized_audio_path", "description": "Path to the normalized
↪ audio file after processing by LoudnessNormalizer", "links": [{"Mixer": "
↪ audio_path"}]},
    {"node": "background_music_path", "description": "File path to the
↪ background music to be mixed with the voice recording", "links": [{"Mixer":
↪  "bgm_path"}]}
],
"Reasoning": "This workflow normalizes the user's raw audio recording and then
↪  mixes it with background music. First, LoudnessNormalizer processes the
↪ raw audio files to ensure consistent volume levels. Since
↪ LoudnessNormalizer doesn't output file paths (only a status), we require
↪ the user to provide the path to the normalized audio file after processing.
↪  Then, the Mixer agent combines this normalized audio with background music
↪ , ensuring the music doesn't overpower the voice. The workflow addresses
↪ the requirement to add background music to a recording while maintaining
↪ proper audio balance."
}
```

Listing 51: User Requirement of Storytelling Video

```
User Requirement: I want to create a commentary video about a novel. The video
↪  should have professional voiceover with background music and appropriate
↪ visuals that match the commentary content.
```

Listing 52: Case of Storytelling Video

```
{
"Feasibility": "Feasible",
"Intent List": ["Text-to-Speech", "Commentary", "Add BGM", "Video Edit"],
"Agent Graph": [
    {"node": "CommentaryContentGenerator", "inputs": [{"name": "reqs", "
↪ description": "User's idea for the commentary video including word count
↪ requirements"}, {"name": "source_text", "description": "File path to the
↪ novel source text"}, {"name": "comm_present_style", "description": "File
↪ path to commentary presentation style for content generation"}], "outputs":
↪  [{"name": "video_scene_path", "description": "File path storing scene
```

```
4752    ↪ semantics for video storyboard sound synthesis.", "links": [{"
4753    ↪ VoiceGenerator": "video_scene_path"}, {"VideoSearcher": "video_scene_path
4754    ↪ "}]}]},
4755      {"node": "VoiceGenerator", "inputs": [{"name": "video_scene_path", "
4756    ↪ description": "Path to a custom scene JSON file"}, {"name": "
4757    ↪ target_vocal_path", "description": "Path to the target timbre for voice
4758    ↪ generation"}], "outputs": [{"name": "audio_path", "description": "Path to
4759    ↪ the synthesized audio", "links": [{"Mixer": "audio_path"}]}, {"name": "
4760    ↪ timestamp_path", "description": "Path to video frame timestamp", "links":
4761    ↪ [{"VideoEditor": "timestamp_path"}]}]},
4762      {"node": "Mixer", "inputs": [{"name": "bgm_path", "description": "Path to
4763    ↪ the BGM file"}, {"name": "audio_path", "description": "Audio to be mixed
4764    ↪ into the background music"}], "outputs": [{"name": "audio_path", "
4765    ↪ description": "Path to the synthesized audio", "links": [{"VideoEditor": "
4766    ↪ audio_path"}]}]},
4767      {"node": "VideoPreloader", "inputs": [{"name": "video_dir", "description":
4768    ↪  "Directory containing the source MP4 video files to be processed"}], "
4769    ↪ outputs": [{"name": "status", "description": "Execution status (success/
4770    ↪ error)", "links": []}]},
4771      {"node": "VideoSearcher", "inputs": [{"name": "video_scene_path", "
4772    ↪ description": "File path storing scene semantics for video storyboard sound
4773    ↪  synthesis."}], "outputs": [{"name": "status", "description": "Execution
4774    ↪ status (success/error)", "links": []}]},
4775      {"node": "VideoEditor", "inputs": [{"name": "video_dir", "description": "
4776    ↪ Directory containing source video files"}, {"name": "audio_path", "
4777    ↪ description": "Path to the audio"}, {"name": "timestamp_path", "description
4778    ↪ ": "JSON File path used to store and load the timestamp of the end of each
4779    ↪ video segment"}], "outputs": [{"name": "video_path", "description": "Path
4780    ↪ to the generated video file", "links": []}]}
4781    ],
4782    "Agent Chain": ["CommentaryContentGenerator", "VoiceGenerator", "Mixer", "
4783    ↪ VideoPreloader", "VideoSearcher", "VideoEditor"],
4784    "User Input Graph": [
4785      {"node": "commentary_requirements", "description": "User's specific
4786    ↪ requirements for the commentary video including word count and style", "
4787    ↪ links": [{"CommentaryContentGenerator": "reqs"}]},
4788      {"node": "novel_file", "description": "File path to the novel text that
4789    ↪ will be the subject of commentary", "links": [{"CommentaryContentGenerator
4790    ↪ ": "source_text"}]},
4791      {"node": "commentary_style", "description": "File path to the desired
4792    ↪ presentation style for the commentary", "links": [{"
4793    ↪ CommentaryContentGenerator": "comm_present_style"}]},
4794      {"node": "voice_timbre", "description": "Path to the target vocal timbre
4795    ↪ file for professional voiceover", "links": [{"VoiceGenerator": "
4796    ↪ target_vocal_path"}]},
4797      {"node": "background_music", "description": "Path to the background music
4798    ↪ file", "links": [{"Mixer": "bgm_path"}]},
4799      {"node": "visuals_directory", "description": "Directory containing source
4800    ↪ video files for visual content", "links": [{"VideoPreloader": "video_dir"},
4801    ↪  {"VideoEditor": "video_dir"}]}
4802    ],
4803    "Reasoning": "The workflow begins with generating commentary content about the
4804    ↪  novel based on user requirements. This content is structured with scene
4805    ↪ semantics. The VoiceGenerator then creates professional voiceover audio and
         ↪  provides timestamps for visual syncing. In parallel, VideoPreloader
         ↪ prepares video files from the user's library while VideoSearcher identifies
         ↪  appropriate visual clips based on the commentary content. The voiceover is
         ↪  mixed with background music by the Mixer agent. Finally, VideoEditor
         ↪ combines the mixed audio track with matching visuals using the timestamps
```

```
4806  ↪ to synchronize content, producing a complete commentary video about the
4807  ↪ novel with professional narration, appropriate background music, and
4808  ↪ relevant visuals."
4809  }
```

### A.13 MLLM/VLM FOR VIDEO EDITING EVALUATION

This subsection presents the standardized system prompt methodology for MLLM integration in video editing benchmark tasks, focusing on precise clip-to-caption matching and automated video content analysis workflows.

Listing 53: MLLM Video Clip Matching Prompt with Frame Integration

```
Prompt Construction with Frame Integration:

1. Text prompt construction:
prompt = f"""
Here is a video broken down into {len(chunks)} clips of 3 seconds each.
Each clip shows multiple frames from that time segment.
The clips are structured as follows:
"""

for chunk in chunks:
    prompt += f"Clip {chunk['chunk_idx']}\n"

prompt += f"""
Identify {num_periods} specific clips that best correspond to the following
↪ captions:
"""

for i, caption in enumerate(shuffled_captions):
    prompt += f"{i+1}. {caption}\n"

prompt += f"""
Response with a JSON structure containing the clip numbers that best match
↪ each caption.

Rules:
1. Each clip_id should be a number (not a string)
2. Include exactly {num_periods} selections
3. Choose from the clip numbers shown (0 to {len(chunks)-1})
4. Keep the reason concise (1-2 sentences)
5. Ensure your output is valid JSON - no trailing commas, proper quotes, etc.
6. Do not include the ```json prefix or ``` suffix around your response
7. Do not answer anything unrelated

Return only the JSON object with no additional text.

Output Example:
{{
"selections": [
    {{
    "caption": "[caption text]",
    "clip_id": [clip number],
    "reason": "[brief description of what is seen in this clip]"
    }},
    ...
]
}}
```

```
"""

2. Frame integration into multimodal message structure:
messages = [{"role": "user", "content": []}]

# Add text prompt as first content element
messages[0]["content"].append({"type": "text", "text": prompt})

# Sequential frame insertion for visual analysis
for img_base64 in frames:
    messages[0]["content"].append({
        "type": "image_url",
        "image_url": {
            "url": f"data:image/jpeg;base64,{img_base64}"
        }
    })
```

