# OpenReview forum: "VideoAgent: All-in-One Agentic Framework for Video Understanding and Editing"
_ICLR.cc/2026/Conference — Submitted to ICLR 2026_

### Official Review · Reviewer_pQcp · 2025-10-23

**Soundness:** 2
**Presentation:** 3
**Contribution:** 2
**Rating:** 4
**Confidence:** 4

**Summary:**

This paper identifies the two main problems in automated video editing (lack of coherence in long-form narratives and inability to handle diverse tasks), and proposes VideoAgent, an all-in-one agentic framework with these main contributions:

* a method for video shot creation with cross-model and context on the full narrative of a target output video
* an orchestration framework that can coordinate a large set of agents to generate a final video edit out of the created video shots
* a new video edit benchmark (VideoEdit)

Complete evaluation follows using both VideoEdit and Shot2Story, which display high performance against the proposed baselines (87-98% success rate), reduced API costs (60% lower costs), and comparable results to those produced by human editors (just 4% below human edits).

The paper also provides detailed prompts and pseudocode for each of the agents used by VideoAgent, and includes source code for this task.

**Strengths:**

The main strengths of this work are:

* Sound engineering, combining a large amount of agents with a solid orchestration method
* Great quantitative results, generally superior to those of the baselines and, especially, very close to human-created videos (with the caveats discussed in the weaknesses section)
* Novel approach that combines multiple-agents with a long-narrative aware video shot process, and an orchestration framework with self-aware elements
* Clear descriptions of the methods used
* Exhaustive details on prompts and pseudocode used in the agents, and open source code, both of which allow for high reproducibility of the work

**Weaknesses:**

The main weaknesses of this work are:

* Lack of details about the human baselines. Just 4% below human performance is a very impressive result, however, which is greatly diminished by lack on details about how this human performance is measured. For example, who are the humans, what is their expertise, which tools have they used, for how long, etc.
  * this point is really critical because the non-human baselines are based on systems that aren't designed to handle multi-modal video editing. So it's difficult to understand what the quality of this system is without a valid human comparison.
  * (minor suggestion): Besides an ad-hoc human baseline of videos edited just for the purpose of this evaluation, one can also wonder how the system would compare against video edits seen on the wild, for different categories. What is the success rate against, for example, against fan edits of existing works.

* Lack of actual examples in video format. The paper displays many examples as frame sequences, but given the nature of this work, the addition of examples in video format would be beneficial to this work. Being able to actually watch and listen to the videos produced by the system (and to compare them to the raw input materials) would provide a better understanding of the system quality.

* While this work details the creation of a significant engineering system, with many agents and a solid orchestration method, the research contributions appear more incremental. To make the research contributions clearer, the paper could describe in more detail how novel aspects in the introduced orchestration system differ from those in other multi agent systems based on LLMs that also deal with graphs. The Related Work section at this time merely describes the application to multimodal video editing workflows which I don't think is enough novelty.

* (minor) Lack of details regarding system latency (though API costs are provided), especially when compared against other baselines. Ideally the paper would include a plot with an axis for latency and another for each key metric, such that the tradeoffs between quality and performance can be better understood. (A similar plot for API costs would be interesting too, and given the reduced API costs of this system, good evidence in favor of this work)

* (minor): Lack of mentions of key downsides for this approach, and potential future work. What do the failure modes look like? What is insightful about them?

* (minor): The appendix may be excessive. The paper could improve by just listing a short summary of each agent behavior, and pointing to the supplementary materials for details.

**Questions:**

* Have you considered using video generation models too, as an agent, for adding shots not included in the input materials?

* Could a much simpler version of this work approach the same quality? For example, could the graph be non-dynamic but fixed, with each node gated on a selector for whether the node agent needs to be applied or not? (this has been explored to some extent in Table 3 which removes Intent Parsing and Agent Graph elements, but I wonder specifically about an existing but fixed graph)

* Have you considered ablating the specific list of agents, to measure how they rank compared to each other? this could inform which other agents are possible future additions for the system

**Details Of Ethics Concerns:**

* The paper displays copyrighted work (e.g. SpongeBob in page 16)

* The voice synthesis and voice cloning features could be used for impersonation

* Possible impact on employment is not discussed; could this work be done in such a way that it allows human video editors to collaborate with the system?

---

> ### Author Response · Authors · 2025-11-20
>
> We sincerely appreciate your constructive feedback. We have carefully considered each of your comments, and we hope our clarifications below elleviate doubts about our work.
>
> **Comment 1 (Weakness 1)**: Discussion on the human evaluation and baselines.
>
> **Response**: Thank you for pointing out this issue. We provide details regarding the human evaluation experiment as follows:
>
> We produced 19 demos, and together with the baseline versions, there are 49 demos in total. They were organized into 18 pairs and evaluated by a total of 26 participants. These participants rated each demo based on three criteria: consistency, audio quality, and scene diversity. The demos were uploaded to a shared drive and organized into a Google Form, with the order randomized and the files anonymized. This evaluation was conducted within our VideoAgent community group, which includes over 200 members from diverse backgrounds such as researchers, content creators, and hobbyists. Collecting all the ratings took about one week, and each participant spent approximately 1 hour to complete the questionnaire. To ensure the transparency of our human evaluation, we provide here the anonymized links to the demos used in the assessment: https://drive.google.com/file/d/1liKL6q10E-sCe1XTnNhtv9G4sAff04g_/view?usp=drive_link.
>
> Moreover, we notice some questions raised in the **minor suggestions** regarding the human-edited works. We apologize for any confusion caused. We would like to clarify this point in more detail: the handiwork in **the human evaluation was not created by us**. Instead, these edits were sourced from video platforms with over 100 million users, with the selected videos having playback counts in the tens of thousands or more. Some of the demos produced by VideoAgent are inspired by the genuine editing ideas of these users—for example, edits related to Ma Baoguo and General Fan—with a success rate of around 90%. We hope this helps address your concern regarding the question: "What is the success rate against, for example, fan edits of existing works?"
>
> **Comment 2 (Weakness 2)**: Examples in Video Format.
>
> **Response**: Thank you for pointing out this issue. For a more intuitive understanding of VideoAgent, we have created anonymous accounts on YouTube and Bilibili to facilitate viewing of our demos.
>
> YouTube link: https://www.youtube.com/@xxxx-w3z7x; Bilibili link: https://space.bilibili.com/341654352.

---

> ### Author Response · Authors · 2025-11-20
>
> **Comment 3 (Weakness 3): Discussion on the innovation and contributions**
>
> **Response:** Thank you for your valuable feedback. We provide clarifications below on how our approach differs from existing multi-agent systems for graph-based workflows.
>
> Our methodology innovation and contributions are three-fold:
>
> **All-in-one Agentic Framework**: Unlike existing methods that are specially designed for specific video creation types (e.g., short clips or domain-specific tasks), we propose a comprehensive agentic framework that unifies video understanding and editing capabilities within a single system, enabling automated video creation across diverse genres and production workflows.
>
> **Global-Aware Video Shot Creation Algorithm**: Existing work directly performs visual information retrieval based on user-input queries, which cannot meet the demands of long-form video creation. By perceiving complete visual summaries and shot storyboard generation, our method can perform shot-level planning that doesn't violate existing visual material content while generating detailed visual expression schemes, thereby enabling coherent narrative creation for extended video content.
>
> **Dynamic Graph Workflow Orchestration Algorithm**: Existing video creation agent systems mostly predefine sequential or static graph workflows, unable to handle diverse video creation demands. Alternatively, some adopt reinforcement learning to train agents, suffering from significant efficiency limitations. Therefore, we propose a dynamic graph-based orchestration method. To accurately coordinate and utilize dozens of agents with complex parameters, we employ an intent parsing mechanism to categorize numerous agents based on functionality, and adopt a dual-stage self-reflection mechanism that further reduces workflow planning failure through multi-round iterative refinement.
>
> We compare the performance of VideoAgent with the supplementary TeaserGen (Xu et al., 2025) in shot planning. In addition, we also add shot retrieval experiments on the VideoRepurpose (Wu et al., 2025) benchmark. Regarding workflow orchestration, we also supplement our comparisons with other graph-based agent frameworks, including GPTSwarm (Zhuge et al., 2024), an agent trained using reinforcement learning; GraphCounselor (Gao et al., 2025), which constructs an agent graph in advance and selects an agent chain using embedding similarity. The results are as follows:
>
> **Table 1: Shot retrieval on Shot2Story benchmark**
>
> |Method|Recall|EM|IoU|Cost|Time|
> |:-:|:-:|:-:|:-:|:-:|:-:|
> |Claude-Sonnet-3.7|46.03|27.95|23.91|0.374|43s|
> |Gemini-2.5-pro|45.98|27.78|25.91|0.349|42s|
> |VideoRAG|31.03|15.84|14.35|0.100|67s|
> |VideoMind-7B|38.26|27.75|19.67|-|21s|
> |TeaserGen|41.88|27.62|20.91|0.079|25s|
> |Ours-Claude-Sonnet-3.7|44.27|28.18|24.81|0.147|37s|
> |Ours-Gemini-2.5-pro|47.24|28.21|25.74|0.136|37s|
>
> **Table 2: Shot retrieval on VideoRepurpose benchmark**
>
> |Method|Recall|IoU|Method|Recall|IoU|
> |:-:|:-:|:-:|:-:|:-:|:-:|
> |GPT-4o|41.27|19.31|TeaserGen|18.46|16.41|
> |Gemini-2.5-pro|43.12|20.68|Ours-GPT-4o|47.13|19.75|
> |Claude-3.7-sonnet|42.19|17.64|Ours-Gemini-2.5-pro|43.18|19.68|
> |VideoMind-7B|40.11|16.89|Ours-Claude-3.7-sonnet|44.64|19.58|
>
> **Table 3: Workflow orchestration of performance comparison in terms of success rate**
>
> |Backbone|Claude-4|Claude-4|Claude-3.7|Claude-3.7|GPT-4o|GPT-4o|Deepseek-v3|Deepseek-v3|
> |:-:|:-:|:-:|:-:|:-:|:-:|:-:|:-:|:-:|
> |Data|Audio|Video|Audio|Video|Audio|Video|Audio|Video|
> |Flow|0.62|0.64|0.84|0.83|0.68|0.60|0.66|0.61|
> |GPTSwarm|0.69|0.83|0.68|0.86|0.64|0.80|0.73|0.81|
> |GraphCounselor|0.83|0.82|0.85|0.81|0.86|0.84|0.82|0.84|
> |**VideoAgent**|**0.93±0.02**|**0.87±0.01**|**0.95±0.03**|**0.93±0.02**|**0.90±0.02**|**0.88±0.01**|**0.92±0.02**|**0.89±0.02**|
>
> These results validate that our shot retrieval and agentic orchestration approach significantly outperforms unified models and alternative agent frameworks in handling complex video creation workflows.
>
> **Comment 4 (Question 1)**: Discussion on adding video generation models as agents
>
> **Response**: Thank you for this insightful suggestion. We agree that incorporating video generation models would be valuable, and our ongoing work following VideoAgent is exploring exactly this direction—using generative models to synthesize missing shots.
>
> However, integrating generative models introduces distinct technical challenges including precise control over generation outputs, automated quality assessment, and seamless integration with existing footage. These challenges differ significantly from the video understanding and workflow orchestration problems we address in this work.
>
> Therefore, we chose to focus this paper on automated coordination of video editing workflows using existing materials, which allows us to thoroughly investigate multi-agent orchestration mechanisms and deep understanding of existing video content. Our modular design provides a solid foundation for incorporating generative agents in future work.

---

> ### Author Response · Authors · 2025-11-20
>
> **Comment 5 (Question 2)**: Feasibility analysis of fixed graph structures
>
> **Response**: We find that this approach bears considerable similarity to the recently proposed GraphCounselor (Gao et al., 2025). GraphCounselor constructs an agent graph in advance based on existing agents, and selects an agent chain by leveraging the embedding similarity. However, the construction of this agent graph heavily depends on the quality of existing agents—for instance, whether the inputs and outputs between agents are standardized, and whether parameter naming is consistent. This poses a significant engineering challenge, especially when dealing with a larger number of agents, and also limits scalability. In contrast, VideoAgent dynamically generates agent graphs according to user requirements, which largely avoids these issues. Our supplementary experiments further demonstrate that the performance of GraphCounselor falls short of that of VideoAgent. Therefore, we believe that the fixed-graph approach is feasible, but it still has certain limitations.
>
> Additionally, we notice that there are some questions regarding the agent graph generated by VideoAgent. In response, we update a file named **graph_demo.txt** in the anonymous github, which contains an existing but relatively fixed graph structure: https://anonymous.4open.science/r/VideoAgent-DC33. Based on the demo, we find when predefining a static graph, precise alignment of the input and output identifiers for each neighboring agent is required. If an agent’s output consists solely of a state variable, and the subsequent agent does not incorporate this state as an input, constructing a complete and coherent static graph becomes challenging. We hope this will help clarify any concerns you might have. We hope this will help clarify any concerns you might have.
>
> **Comment 6 (Question 3)**: Discussion on the ablation of specific agents
>
> **Response**: Thank you for your very valuable suggestions. Performing ablation on specific agents to assess their importance indeed helps with adding other agents in the future as well as understanding the orchestration among agents. Accordingly, we have added the corresponding ablation experiments, and the results are as follows:
>
> |Backbone|Claude-3.7|Claude-3.7|GPT-4o|GPT-4o|Deepseek-v3|Deepseek-v3|
> |:-:|:-:|:-:|:-:|:-:|:-:|:-:|
> |Data|Audio|Video|Audio|Video|Audio|Video|
> |- LoudnessNormalizer|0.84|0.82|0.79|0.77|0.91|0.95|
> |- AudioExtractor|0.65|0.80|0.72|0.83|0.61|0.82|
> |- StandUpSynth|0.04|0.05|0.05|0.04|0.02|0.01|
>
> We selected three editing agents and measured the relative success rates after ablation of each. It can be observed that ablating certain auxiliary-function agents has a much less negative impact on the success rate of workflow orchestration compared to ablating core-function agents. This finding provides important guidance for the future expansion of agents.

---

> > ### Comment · Reviewer_pQcp · 2025-11-21
> >
> > Thank you very much for your detailed answers. In a first read of your replies my main question is about the following claim:
> >
> > > However, the construction of this agent graph heavily depends on the quality of existing agents—for instance, whether the inputs and outputs between agents are standardized, and whether parameter naming is consistent. This poses a significant engineering challenge, especially when dealing with a larger number of agents, and also limits scalability. In contrast, VideoAgent dynamically generates agent graphs according to user requirements, which largely avoids these issues. (...) Based on the demo, we find when predefining a static graph, precise alignment of the input and output identifiers for each neighboring agent is required. If an agent’s output consists solely of a state variable, and the subsequent agent does not incorporate this state as an input, constructing a complete and coherent static graph becomes challenging.
> >
> > It is my understanding that this proper alignment of input and output types, names, etc. is a requirement for both fixed and dynamic graphs. I assume the reason why it would not be seen as a problem for dynamic graphs is because you have a method for filtering nodes based on such signatures. In which case such method could be used during the manual construction of the fixed graph too. Or one dynamically constructed graph structure could be used to simulate how a fixed graph would behave.
> >
> > Even if the creation of one fixed graph approaches the cost of generation of any graph at runtime, the run-time latency and the long term maintenance of a fixed graph is diminished which makes me think this alternative could be worth considering.
> >
> > Please let me know if I am misunderstanding why the input / output signatures would be a problem for fixed graphs but not for dynamic graphs.

---

> > > ### Author Response · Authors · 2025-11-22
> > >
> > > We are glad to clarify this important point. In our dynamic graph framework, alignment is only required at the level of parameter types between inputs and outputs—there is no requirement for the parameter names themselves to match. This greatly simplifies the engineering complexity and enhances scalability, especially when dealing with many agents. For both researchers and developers, it means you only need to be aware of the limited parameter types in the system to add new agents, without needing to know or strictly adhere to the specific input/output names. They can be freely defined.
> > >
> > > Specifically, when VideoAgent dynamically constructs the graph, each output from an agent node explicitly points to the input of the next agent as long as their types match. There is no need for the parameter names themselves to be consistent, which significantly reduces the engineering burden. However, for pre-defined static graphs, it is necessary to strictly standardize all agents’ input and output types and names. In scenarios involving many agents, this becomes a massive engineering effort and the scalability is very poor. High flexibility and scalability are also key advantages of VideoAgent. We hope this helps to clarify any concerns you may have.

---

> > > > ### Comment · Reviewer_pQcp · 2025-11-23
> > > >
> > > > > However, for pre-defined static graphs, it is necessary to strictly standardize all agents’ input and output types and names.
> > > >
> > > > Why is this true?
> > > >
> > > > I think of these as two independent factors: 1) the graph structure itself, 2) the input and output signatures.
> > > >
> > > > You can take the dynamic system you currently have (for which both 1 and 2 are dynamic), and change only 1 by making the actual structure of the graph constant, while keeping 2 dynamic by keeping the rest of the code just the same. Unless I'm missing something this shows that it is not necessary to strictly standardize the signatures for fixed graphs.

---

> ### Author Response · Authors · 2025-11-20
>
> **Comment 7 (Weakness 4&5&6)**: Discussion on the minor suggestions
>
> **Response**: Thank you for your feedback. Although they are minor suggestions, we find them very valuable. We will respond point by point in hopes of addressing your concerns.
>
> - **Details regarding system latency**: We add latency information for the shot retrieval and workflow orchestration here. In the revised version, we will update this section into the suggested graphical format. The results are as follows:
>
> **Table: System latency of shot retrieval**
>
> |Method|Recall|EM|IoU|Cost|Time|
> |:-:|:-:|:-:|:-:|:-:|:-:|
> |Claude-Sonnet-3.7|46.03|27.95|23.91|0.374|43s|
> |Gemini-2.5-pro|45.98|27.78|25.91|0.349|42s|
> |VideoRAG|31.03|15.84|14.35|0.100|67s|
> |VideoMind-7B|38.26|27.75|19.67|-|21s|
> |TeaserGen|41.88|27.62|20.91|0.079|25s|
> |Ours-Claude-Sonnet-3.7|44.27|28.18|24.81|0.147|37s|
> |Ours-Gemini-2.5-pro|47.24|28.21|25.74|0.136|37s|
>
> **Table: System latency of workflow orchestration (in seconds)**
>
> |Backbone|Claude-4|Claude-4|Claude-3.7|Claude-3.7|GPT-4o|GPT-4o|Deepseek-v3|Deepseek-v3|
> |:-:|:-:|:-:|:-:|:-:|:-:|:-:|:-:|:-:|
> |Data|Audio|Video|Audio|Video|Audio|Video|Audio|Video|
> |Flow|112.31|91.76|103.75|89.36|98.47|102.88|109.18|115.17|
> |GPTSwarm|79.63|101.32|76.84|97.45|43.29|51.57|76.87|60.32|
> |GraphCounselor|24.78|38.43|26.31|33.85|37.02|30.85|45.08|21.46|
> |**VideoAgent**|**38.14**|**56.73**|**39.90**|**53.42**|**23.21**|**39.45**|**50.48**|**36.67**|
>
> As shown in the above tables, VideoAgent achieves a well-balanced performance in both effectiveness and efficiency, demonstrating clear advantages over most baselines by ensuring high task completion quality while also improving computational efficiency.
>
> - **Limitations and Future Work**: Thank you for the reminder. We provide a brief supplement here and will include a "Limitations and Future Work" section in the revised version. We argue that the current framework relies to some extent on the quality of multimodal input materials, which directly impacts the final product. To bridge the gap, we intend to integrate supplementary resources, including outputs generated by multimodal models as well as curated external material repositories, and systematically align these with users' multimodal inputs along multiple feature dimensions. This multi-faceted alignment is expected to substantially improve the robustness and quality of the resulting product. Additionally, during our research, we found that some existing foundational models are still unable to perform end-to-end conversion of user ideas into high-quality entertainment products—such as the Meme Video in VideoAgent. In the future, it would be worthwhile to develop such foundational models that can directly accept raw input materials and generate creative derivative works based on user ideas. We recognize that this presents significant technical challenges, but it is an interesting and promising direction deserving further attention.
>
> - **Length of the Appendix**: Thank you for your valuable suggestion. In the revised version, we will provide a concise overview of the agents’ behaviors and functions in Appendix sections A5–A8, and include clear references in the main text to guide readers to the relevant agents, aiming to minimize any potential reading burden.

---

> ### Author Response · Authors · 2025-11-23
>
> We apologize for the misunderstanding regarding your point. The strict specification of input and output types and names of static graphs that we mentioned before refers to the approach where static graphs are predefined through automatically identifying paths by loading all agents programmatically (without relying on LLMs). However, it is also possible to define these graphs manually or with the help of LLMs, in which case strict standardization of input and output signatures is not necessary.
>
> We carefully considered the user experience. In practice, users can define some static graphs themselves. In addition, we have pre-defined the six most commonly used static graph types that are highly popular on video platforms: Beat-synced Edits, Storytelling Video, Video Overview, Meme Video Remaking, Song Remixes, and Cross-lingual Adaptations.
>
> The dynamic graphs we propose are intended to address the need for flexibility among all creators, which is particularly important in scenarios with a large number of agents and diverse user demands. Users can flexibly choose between static and dynamic graphs within our framework to best meet their individual needs.

---

> > ### Comment · Reviewer_pQcp · 2025-11-24
> >
> > Thank you for your detailed answers which satisfy my main doubts about this paper. Specifically your responses about human baselines, video examples and also latency are solid. I will raise my rating accordingly.
> >
> > Thank you for your research on this topic.

---

> > > ### Author Response · Authors · 2025-11-25
> > >
> > > We sincerely appreciate your careful consideration and positive reassessment of our paper during the rebuttal phase. Your thoughtful comments and increased score provide us with great encouragement and motivation. Thank you very much for your time and constructive feedback, which have helped us improve the quality and clarity of our work.

---

### Official Review · Reviewer_CE6n · 2025-10-31

**Soundness:** 3
**Presentation:** 3
**Contribution:** 3
**Rating:** 6
**Confidence:** 4

**Summary:**

This paper presents VideoAgent, an multi-agent framework for automated video understanding and editing, aiming to enable general-purpose video creation with coherent narratives and long-video reasoning. The framework consists of two major components including automated video shot creation and multi-agent orchestration. Besides, a new VideoEdit benchmark is introduced for evaluation. Experiments on video understanding, video retrieval, and workflow orchestration show that VideoAgent outperforms existing multimodal LLMs and agentic systems.

**Strengths:**

1.	The presentation is clear and easy to follow.

2.	The experiments and visualizations are reasonable and well done.

3.	Over 30 tool agents support a wide range of operations (audio, visual, translation, meme creation, etc.), suggesting high practical applicability.

**Weaknesses:**

1.	While the system integration is impressive, most modules (retrieval, trimming, intent parsing) adapt existing methods rather than proposing new algorithms.

2.	The reliance on proprietary or external APIs (e.g., GPT-4o, Claude-Sonnet, Gemini-2.5) may limit true reproducibility and comparability.

3.	The new VideoEdit benchmark seems self-curated and may not fully represent real-world creative diversity.

4.	The paper lacks discussion on computational efficiency or latency of the full multi-agent pipeline — an important factor for large-scale or real-time production.

**Questions:**

1.	The paper mentions multi-agent orchestration – integrating more than 30 specialized editing agents for diverse operations (e.g., rhythm detection, voice cloning, translation, and trimming). Can the entire system operate end-to-end automatically, or does it require manual intervention between stages? If so, how efficient is the end-to-end pipeline in real use cases? What is the average generation time and computational cost for producing a multi-scene video?

2.	Given so many external or API-based agents, how reproducible are the results if other researchers attempt to re-run the same pipeline?

3.	How well does the system handle long-form or multi-hour videos? Are there memory or latency constraints when orchestrating dozens of agents?

4.	How does the self-reflective orchestration prevent error propagation between dependent agents, and can failed subgraphs be re-executed automatically?

5.	What is the maximum number of characters/scenes that VideoAgent can process simultaneously while maintaining quality and coherence?

---

> ### Author Response · Authors · 2025-11-21
>
> Thank you for your recognition of our work and for the valuable suggestions. We will carefully address each of the questions to alleviate your concerns about our work, hoping to further improve your impression of our efforts. In order to give a more authentic experience of VideoAgent’s performance, we create anonymous accounts on both YouTube and Bilibili, making it more convenient to watch our demos.
>
> YouTube link: https://www.youtube.com/@xxxx-w3z7x; Bilibili link: https://space.bilibili.com/341654352.
>
> **Comment 1 (Weakness 1):**  Discussion on the innovation and contributions
>
> **Response:** Thank you for your insightful comments. To clarify and highlight the distinctive aspects of our work, we would like to briefly elaborate on the key innovations and contributions presented in our approach.
>
> - **All-in-one Agentic Framework**. Our framework stands out by integrating video understanding and editing into a unified agentic framework. In contrast to prior methods tailored to specific video styles or domains, our approach supports a broad range of video creation scenarios within a single architecture.
>
> - **Global-Aware Video Shot Creation Algorithm**. We introduce a global-aware shot creation strategy. Unlike existing techniques that retrieve visual content based solely on user queries without holistic context, our method analyzes comprehensive visual summaries and shot storyboards. This enables deliberate shot-level planning that respects original material, producing coherent long-form narratives with rich visual detail.
>
> - **Dynamic Graph Workflow Orchestration Algorithm**. We propose a dynamic graph-based orchestration algorithm. While many existing agent systems depend on fixed or linear workflows, our method dynamically orchestrates multiple heterogeneous agents through intent parsing and a dual-stage self-reflection process. This adaptive mechanism effectively handles workflow complexity and significantly reduces failure rates during video production planning.
>
> **Comment 2 (Question 1 & Weakness 4):** Analysis of the system's operational performance
>
> **Response:** We will carefully address each sub-question and hope to resolve any concerns you may have. 1) VideoAgent is capable of running completely in an end-to-end manner, requiring no human intervention during the process. We also provide YouTube and Bilibili links at the beginning to directly watch the generated videos. 2) We test the system in real-world scenarios using an NVIDIA RTX 4090 GPU. When a user inputs a 10-minute 1080p video, generating a roughly 1-minute highlight clip takes about 5 minutes, with a computational cost of approximately $0.03472. Additionally, to further demonstrate the efficiency of VideoAgent, we supplement our work with a series of efficiency experiments, as follows:
>
> **Table 1: System latency of shot retrieval**
>
> | Method | Recall | EM | IoU | Cost | Time |
> |:-:|:-:|:-:|:-:|:-:|:-:|
> | Claude-Sonnet-3.7 | 46.03 | 27.95 | 23.91 | 0.374 | 43s |
> | Gemini-2.5-pro | 45.98 | 27.78 | 25.91 | 0.349 |  42s |
> | Qwen-2.5-VL-72B-Instruct | 18.89 | 27.99 | 10.51 | - | 45s |
> | VideoRAG | 31.03 | 15.84 | 14.35 | 0.100 | 67s |
> | VideoMind-7B | 38.26 | 27.75 | 19.67 | - | 21s |
> | TeaserGen | 41.88 | 27.62 | 20.91 | 0.079 | 25s |
> | Ours-Claude-Sonnet-3.7 | 44.27 | 28.18| 24.81| 0.147 | 37s |
> | Ours-Gemini-2.5-pro | 47.24 | 28.21 | 25.74 | 0.136 | 37s |
>
> **Table 2: System latency of workflow orchestration (in seconds)**
>
> |Backbone|Claude-4|Claude-4|Claude-3.7|Claude-3.7|GPT-4o|GPT-4o|Deepseek-v3|Deepseek-v3|
> |:-:|:-:|:-:|:-:|:-:|:-:|:-:|:-:|:-:|
> |Data|Audio|Video|Audio|Video|Audio|Video|Audio|Video|
> |Flow|112.31|91.76|103.75|89.36|98.47|102.88|109.18|115.17|
> |GPTSwarm|79.63|101.32|76.84|97.45|43.29|51.57|76.87|60.32|
> |GraphCounselor|24.78|38.43|26.31|33.85|37.02|30.85|45.08|21.46|
> |**VideoAgent**|**38.14**|**56.73**|**39.90**|**53.42**|**23.21**|**39.45**|**50.48**|**36.67**|
>
> We measure the average time spent on each workflow orchestration and shot retrieval. The results presented in the tables above indicate that VideoAgent attains a balanced trade-off between effectiveness and efficiency. It consistently outperforms most baseline methods by delivering high-quality task completion outcomes while simultaneously enhancing computational efficiency.

---

> ### Author Response · Authors · 2025-11-21
>
> **Comment 3 (Weakness 2 & Question 2):** Discussion on the reproducibility of experimental results
>
> **Response:** Thank you very much for your valuable suggestions. We incorporate standard deviation measurements into the workflow orchestration and shot retrieval experiments to further ensure the reproducibility of the results, hoping this can address your concerns. The experimental results are as follows:
>
> **Table 3: Video understanding and retrieval performance comparison in terms of multiple metrics**
>
> | Method | Recall | EM | IoU | Cost | Time |
> |:-:|:-:|:-:|:-:|:-:|:-:|
> | Ours-Claude-Sonnet-3.7 | 44.27±0.0048 | 28.18±0.02 | 24.81±0.47 | 0.147 | 37s |
> | Ours-Claude-Sonnet-3.5 | 38.70±0.0035 | 28.45±0.02 | 21.32±0.53 | 0.147 | 36s |
> | Ours-Gemini-2.5-pro | 47.24±0.0039 | 28.21±0.02 | 25.74±0.49 | 0.136 | 37s |
> | Ours-Gemini-2.5-flash | 44.93±0.0041 | 28.25±0.02 | 25.07±0.51 | 0.028 | 35s |
> | Ours-GPT-4o | 48.85±0.0038 | 28.26±0.02 | 26.99±0.49 | 0.099 | 37s |
>
> **Table 4: Workflow orchestration of performance comparison in terms of success rate**
>
> |Backbone|Claude-4|Claude-4|Claude-3.7|Claude-3.7|GPT-4o|GPT-4o|Deepseek-v3|Deepseek-v3|
> |:-:|:-:|:-:|:-:|:-:|:-:|:-:|:-:|:-:|
> |Data|Audio|Video|Audio|Video|Audio|Video|Audio|Video|
> |Flow|0.62|0.64|0.84|0.83|0.68|0.60|0.66|0.61|
> |GPTSwarm|0.69|0.83|0.68|0.86|0.64|0.80|0.73|0.81|
> |GraphCounselor|0.83|0.82|0.85|0.81|0.86|0.84|0.82|0.84|
> |**VideoAgent**|**0.93±0.02**|**0.87±0.01**|**0.95±0.03**|**0.93±0.02**|**0.90±0.02**|**0.88±0.01**|**0.92±0.02**|**0.89±0.02**|
>
> As shown in the tables above, VideoAgent not only demonstrates superior performance but also exhibits good reproducibility.
>
> **Comment 4 (Weakness 3):** Discussion on VideoEdit benchmark
>
> **Response:** We also give careful consideration to how to reflect the diversity of real-world creativity. To that end, we generated user ideas based on videos from Bilibili platform, which has over 100 million users, selecting those that ranked in the top 20 of the weekly charts and had over one million views. These ideas were then used as input for our VideoAgent to create a series of derivative videos, encompassing six different types of products. You are welcome to click the anonymous account below to view these demos. We hope this can help alleviate any concerns you may have.
>
> YouTube link: https://www.youtube.com/@xxxx-w3z7x; Bilibili link: https://space.bilibili.com/341654352.
>
> **Comment 5 (Question 3):** Discussion on handling long videos and multiple agents
>
> **Response:** We will address all the questions point by point in hopes of resolving your confusion.
>
> - One major advantage of VideoAgent is its support for raw input videos lasting several hours, whereas other baselines such as Director and FunClip not only have limited video editing types but also do not support such long videos as input. In real-world scenarios, for example, the Spider-Man mashup demo we created uses a two-hour movie as the input material. We have also provided anonymous YouTube and Bilibili links in the beginning so that all our demos can be directly viewed for convenience.
>
> - Our dataset includes cases where more than ten agents coordinate together. During our experiments, there was no limitation on system memory, and the model can run properly with an 8GB GPU. We also supplement efficiency experiments in Tables 1 and 2 earlier in the response, demonstrating that VideoAgent achieves an excellent balance between effectiveness and efficiency.

---

> ### Author Response · Authors · 2025-11-21
>
> **Comment 6 (Question 4):** Discussion on how the self-reflection mechanism prevents erroneous propagation of agent information.
>
> **Response:** We design a dual-stage self-reflection mechanism. Specifically, we first use intent recognition to classify and filter the potentially relevant agents, which effectively reduces the information burden when the number of agents is large. Next, through our proposed graph-based orchestration method, we precisely guide the LLMs to autonomously select neighboring nodes of agents, gradually constructing an agent graph. To reduce judging errors made by the LLM during evaluation, we designed a two-step evaluation mechanism to assess the generated agent graph under strict criteria. If the LLM judges that the agent graph cannot be generated, it will reflect on the set of agents filtered by intent recognition and attempt some intent adjustments. However, if the LLM judges that the agent graph can be generated but is insufficient to meet user requirements or contains internal defects, it will specifically reflect on the graph construction process and provide adjustment suggestions. We believe that such a targeted self-reflection mechanism can more effectively prevent the erroneous propagation of agent information, and experiments have demonstrated the superiority of our workflow orchestration.
>
> Moreover, our VideoAgent supports re-execution of failed subgraphs. This is mainly reflected in the LLM judgment phase: whether it determines that the agent graph cannot be generated, or that the graph is generated but does not satisfy user needs or has internal defects, our VideoAgent will actively identify the issues and attempt different solutions until a complete agent graph is produced or the reflection limit is reached. It does not stop running simply due to the failure of agent graph construction.
>
> **Comment 7 (Question 5):** Discussion on limits regarding number of characters and scenes
>
> **Response:** VideoAgent is capable of handling an unlimited number of characters and scenes while maintaining quality and coherence. For example, a 50-second Spider-Man mashup includes around 12 scenes and 7 characters, and a 3.5-minute narrative explanation video of the novel "Joy of Life" contains approximately 39 scenes and over ten characters. This demonstrates the advantages of the framework. The YouTube and Bilibili links mentioned at the beginning provide a more intuitive understanding of these capabilities.

---

### Official Review · Reviewer_oTRV · 2025-11-01

**Soundness:** 3
**Presentation:** 3
**Contribution:** 2
**Rating:** 4
**Confidence:** 4

**Summary:**

This paper proposes VideoAgent, an agent-based framework for automated video editing. By introducing a global-aware video shot creation mechanism and a self-reflective agent graph orchestration strategy, VideoAgent demonstrates promising results. Nevertheless, the paper still has several aspects that could be further improved.

**Strengths:**

1. The paper focuses on the task of video editing and content creation, which holds significant practical value in real-world applications.

2. The paper is well-written and easy to follow, with a comprehensive appendix that provides detailed explanations of the technical aspects of the proposed work.

**Weaknesses:**

1. The definition and research scope of the task are not clearly articulated. Video editing is a highly broad concept, and the authors should explicitly specify which sub-tasks are covered by this work.

2. In Section 2.3.1, the authors mention functionalities such as face swapping and lip synchronization, yet there appears to be no corresponding agent described in Appendix A.5.

3. The paper lacks methodological novelty and sufficient contribution; the proposed system is largely built upon existing techniques and relies heavily on prompt engineering rather than introducing new algorithmic insights.

4. The overall framework appears redundant and overly complicated. Constructing a dedicated dataset to train a more compact and unified model would likely be more effective.

5. The paper makes extensive use of LLMs, but does not include a dedicated section “Usage of LLMs”.

**Questions:**

Please refer to Weaknesses.

---

> ### Author Response · Authors · 2025-11-18
>
> We sincerely appreciate your valuable feedback and have carefully addressed all your concerns in our responses. We hope our clarifications help alleviate any doubts about our work. For a more intuitive understanding of VideoAgent, we have created anonymous accounts on YouTube and Bilibili to facilitate viewing of our demos.
>
> YouTube link: https://www.youtube.com/@xxxx-w3z7x; Bilibili link: https://space.bilibili.com/341654352
>
> **Comment 1:**  Discussion on the definition of the task and the scope of the study
>
> **Response:** Traditional video editing approaches are often limited to processing short video segments or are specialized in domain-specific tasks, which restricts their ability to meet diverse real-world requirements.
>
> We define video editing as creating fully coherent videos from natural language instructions and multimodal inputs. Our approach addresses two core operations: 1) selecting and assembling segments from raw video materials to construct narrative-driven sequences, and 2) utilizing audio-visual tools for content remaking and post-production enhancement. VideoAgent automatically executes both operations to handle diverse video production demands.
>
> The scope of our study specifically targets two key challenges:
> - **Coherent video planning**: We develop a globally-aware shot planning agent that employs cross-modal retrieval to organize shots into consistent long-form narratives.
> - **Multi-agent workflow orchestration**: We build a framework with over 30 specialized editing agents, using intent parsing and self-reflective graph construction to dynamically assemble complex editing pipelines.
>
>
> **Comment 2:**  Discussion on the absence of face swapping and lip synchronization agents
>
> **Response:** Thank you for bringing this to our attention. We will add the description of the face swapping and lip synchronization agents to the appendix in the revised version of the paper. The face swapping agent is implemented by Viggle AI and the lip synchronization agent by Kling AI. We will incorporate detailed information in the revised version.
>
> **Comment 3:**  Analysis of the innovation and contributions
>
> **Response:** Thank you for your valuable feedback. We provide clarifications below to better demonstrate our research contributions and methodological innovations. Our methodology innovation and contributions are three-fold:
>
> - **All-in-one Agentic Framework**. Unlike existing methods that are specially designed for specific video creation types (e.g., short clips or domain-specific tasks), we propose a comprehensive agentic framework that unifies video understanding and editing capabilities within a single system, enabling automated video creation across diverse genres and production workflows.
>
> - **Global-Aware Video Shot Creation Algorithm**. Existing work directly performs visual information retrieval based on user-input queries, which cannot meet the demands of long-form video creation. By perceiving complete visual summaries and shot storyboard generation, our method can perform shot-level planning that doesn't violate existing visual material content while generating detailed visual expression schemes, thereby enabling coherent narrative creation for extended video content.
>
> - **Dynamic Graph Workflow Orchestration Algorithm**. Existing video creation agent systems mostly predefine sequential or static graph workflows, unable to handle diverse video creation demands. Alternatively, some adopt reinforcement learning to train agents, suffering from significant efficiency limitations. Therefore, we propose a dynamic graph-based orchestration method. To accurately coordinate and utilize dozens of agents with complex parameters, we employ an intent parsing mechanism to categorize numerous agents based on functionality, and adopt a dual-stage self-reflection mechanism that further reduces workflow planning failure through multi-round iterative refinement.

---

> ### Author Response · Authors · 2025-11-18
>
> **Comment 4:**  Discussion on the complexity of the overall framework.
>
> **Response:** We appreciate your feedback on the framework design. We respectfully disagree that our approach is redundant or overly complicated for the following reasons:
>
> **Unified Yet Flexible Architecture**: In our unified all-in-one VideoAgent framework, our architecture is actually simple and flexible to handle the diverse video creation demands. We have only two core algorithms: automated video shot creation and self-reflective agent orchestration. The perceived complexity stems from the inherent diversity of video production requirements, not architectural redundancy.
>
> **Necessity of Specialized Tools**: Video production inherently requires highly specialized capabilities including voice cloning, song generation, scriptwriting, and dialogue creation. Training a single unified model to master all these domain-specific functions would be both computationally prohibitive and technically challenging, as each requires distinct expertise and training paradigms. Our agentic approach leverages state-of-the-art specialized models for each function while providing intelligent coordination.
>
> **Empirical Validation**: We compare the performance of VideoAgent with general-purpose LLMs and the trained models (i.e., VideoMind and the supplementary TeaserGen (Xu et al., 2025)) in shot planning. In addition, we also add shot retrieval experiments on the VideoRepurpose (Wu et al., 2025) benchmark. Regarding workflow orchestration, we also supplement our comparisons with other graph-based agent frameworks, including GPTSwarm (Zhuge et al., 2024), an agent trained using reinforcement learning; GraphCounselor (Gao et al., 2025), which constructs an agent graph in advance and selects an agent chain using embedding similarity. The additional results are as follows:
>
> **Table 1: Video understanding and retrieval performance on Shot2Story Benchmark**
>
> |Method|Recall|EM|IoU|Cost|Time|
> |:-:|:-:|:-:|:-:|:-:|:-:|
> |Claude-Sonnet-3.7|46.03|27.95|23.91|0.374|43s|
> |Gemini-2.5-pro|45.98|27.78|25.91|0.349|42s|
> |VideoRAG|31.03|15.84|14.35|0.100|67s|
> |VideoMind-7B|38.26|27.75|19.67|-|21s|
> |TeaserGen|41.88|27.62|20.91|0.079|25s|
> |Ours-Claude-Sonnet-3.7|44.27|28.18|24.81|0.147|37s|
> |Ours-Gemini-2.5-pro|47.24|28.21|25.74|0.136|37s|
>
> **Table 2: Video understanding and retrieval performance on VideoRepurpose Benchmark**
>
> |Method|Recall|IoU|Method|Recall|IoU|
> |:-:|:-:|:-:|:-:|:-:|:-:|
> |GPT-4o|41.27|19.31|TeaserGen|18.46|16.41|
> |Gemini-2.5-pro|43.12|20.68|Ours-GPT-4o|47.13|19.75|
> |Claude-3.7-sonnet|42.19|17.64|Ours-Gemini-2.5-pro|43.18|19.68|
> |VideoMind-7B|40.11|16.89|Ours-Claude-3.7-sonnet|44.64|19.58|
>
> **Table 3: Workflow orchestration of performance comparison in terms of success rate**
>
> |Backbone|Claude-4|Claude-4|Claude-3.7|Claude-3.7|GPT-4o|GPT-4o|Deepseek-v3|Deepseek-v3|
> |:-:|:-:|:-:|:-:|:-:|:-:|:-:|:-:|:-:|
> |Data|Audio|Video|Audio|Video|Audio|Video|Audio|Video|
> |Flow|0.62|0.64|0.84|0.83|0.68|0.60|0.66|0.61|
> |GPTSwarm|0.69|0.83|0.68|0.86|0.64|0.80|0.73|0.81|
> |GraphCounselor|0.83|0.82|0.85|0.81|0.86|0.84|0.82|0.84|
> |**VideoAgent**|**0.93±0.02**|**0.87±0.01**|**0.95±0.03**|**0.93±0.02**|**0.90±0.02**|**0.88±0.01**|**0.92±0.02**|**0.89±0.02**|
>
> These results validate that our shot retrieval and agentic orchestration approach significantly outperforms unified models and alternative agent frameworks in handling complex video creation workflows.
>
> **Comment 5:**  Discussion on the usage of LLMs
>
> **Response:** Thank you for your suggestion. Here, we provide a brief overview of the "Usage of LLMs": In this work, LLMs are leveraged as key functions in multiple editing agents to enhance understanding, planning, and execution capabilities:
>
> - **Shot Planning Agent**: LLMs compress visual keyframe captions and combine them with user instructions to generate coherent shot-level storyboards, ensuring global narrative consistency.
>
> - **Fine-grained Video Trimming**: Vision Language Models adaptively select precise video segments according to storyboard text and target shot duration, achieving temporal alignment with narrative rhythm.
>
> - **Efficient Agent Selection via Intent Parsing**: LLMs parse user instructions to extract intents that map to relevant agents, enabling efficient agent filtering and prioritization from a registry.
>
> - **Self-Reflective Agent Graph Orchestration**: LLMs iteratively refine intent understanding through self-reflection and evaluate the agent graph workflow, improving agent selection and mitigating failures.
>
> Overall, LLMs provide high-level reasoning, multimodal integration, and self-reflective orchestration to enable adaptive and coherent video generation aligned with user goals. In addition, we make limited use of LLMs for language refinement during the preparation of this manuscript.
>
> We will add the "Usage of LLMs" section in the revised version.

---

> > ### Comment · Reviewer_oTRV · 2025-11-26
> >
> > Thank you for your response. However, several issues remain unresolved:
> >
> > **Regarding Comment 1:**
> > Thank you for providing the definition of video editing. I would like to clarify whether the scope you define encompasses all video editing tasks, or only a subset of them.
> >
> > **Regarding Comments 2 and 5:**
> > Given that ICLR allows revisions to the submitted manuscript, would it be possible for you to incorporate and submit the additional content addressing these points?
> >
> > **Regarding Comment 3:**
> > I understand the methodological design presented. However, as mentioned in my initial review, these components appear to be incremental modifications to existing models, which makes the contributions of the paper relatively limited.

---

> > > ### Author Response · Authors · 2025-11-26
> > >
> > > Thank you very much for your valuable feedback. We carefully addressed each question, and hope this helps to alleviate any doubts you may have.
> > >
> > > **Comment 1**: The scope of video editing tasks
> > >
> > > The scope we defined encompasses all video editing tasks. Our VideoAgent is capable of handling a wide range of complex and diverse video editing tasks, and can achieve satisfactory results. To provide a more intuitive sense of VideoAgent’s performance in handling various editing tasks, we created anonymous accounts on YouTube and Bilibili, featuring nearly 20 videos edited by VideoAgent across different scenarios. The links are as follows:
> > >
> > > YouTube link: https://www.youtube.com/@xxxx-w3z7x; Bilibili link: https://space.bilibili.com/341654352
> > >
> > > **Comment 2**: Revisions to the additional content
> > >
> > > Thank you for your kind reminder. In response to **Comments 2**, regarding the **Face Swapping and Lip Synchronization** section, we accordingly updated Appendix sections A.5, A.6.4, and A.6.5 to provide a detailed explanation of the functionalities and implementation of these two agents. Furthermore, in regard to **Comments 5**, concerning the "Usage of LLMs" part, we have added a new section titled "Usage of LLMs" after the main text and before the references to clearly clarify the role of LLMs in our work.
> > >
> > > **Comment 3**:  Discussion on innovation and contributions
> > >
> > > We are pleased to further elaborate on the contributions of our work to existing models and agent systems, and hope this may help address any concerns you may have.
> > >
> > > - **Unified Agentic Framework for Video Editing**: We propose an all-in-one agentic framework that consolidates video comprehension and editing functionalities within a single cohesive architecture. Unlike existing approaches that are often confined to particular video genres or application domains, like FunClip (ModelScope, 2024) and NarratoAI (linyqh, 2025), our framework generalizes across a wide spectrum of video creation scenarios. This unification significantly improves flexibility and broadens practical applicability by eliminating the need for domain-specific systems.
> > >
> > > - **Global-Aware Video Shot Creation Strategy**: Our method introduces a novel shot creation algorithm incorporating global contextual awareness. Distinct from conventional retrieval-based techniques that rely solely on user queries without exploiting broader visual context (e.g., VideoMind (Liu et al., 2025) and TeaserGen (Xu et al., 2025)), our approach leverages holistic visual summaries and storyboard representations to guide shot-level planning. This comprehensive strategy ensures that the reconstructed video narratives maintain coherence and preserve the integrity of the original material, resulting in long-form videos with enhanced visual richness and narrative consistency.
> > >
> > > - **Dynamic Graph-Based Workflow Orchestration Algorithm**: We develop an adaptive workflow orchestration algorithm grounded in a dynamic graph structure. Whereas traditional agent-based systems often depend on fixed or linear task execution sequences—for example, GPTSwarm (Zhuge et al., 2024) and GraphCounselor (Gao et al., 2025), our approach flexibly coordinates multiple heterogeneous agents in response to parsed user intents. Incorporating a dual-stage self-reflection mechanism, this method dynamically adjusts the workflow to manage complexity effectively and reduce failure rates. Notably, it achieves efficiency and robustness without relying on reinforcement learning or predefined graphs, addressing key limitations found in prior agent orchestration frameworks.

---

> ### Author Response · Authors · 2025-11-26
>
> We sincerely appreciate your valuable feedback and have thoughtfully reviewed and addressed each point raised. If there are any remaining questions or if you would like us to clarify anything further, please feel free to reach out. We would be more than happy to assist. We look forward to hearing from you at your convenience.

---

### Official Review · Reviewer_bw9K · 2025-11-01

**Soundness:** 2
**Presentation:** 2
**Contribution:** 3
**Rating:** 4
**Confidence:** 3

**Summary:**

The paper presents VideoAgent, an all-in-one agentic framework that integrates video understanding, editing, and workflow orchestration within a unified system. It introduces a shot planning agent for coherent long-form video generation and a self-reflective agent graph orchestration module that dynamically assembles workflows using specialized agents. Evaluations on the new VideoEdit benchmark show significant improvements over baselines such as VideoRAG and VideoMind. Human evaluations rate its outputs close to professional-level quality, demonstrating strong potential for scalable, automated video creation.

**Strengths:**

1. Comprehensive End-to-End Pipeline:
The manuscript describes a full pipeline for generating video content from multi-modal inputs. By integrating narrative planning with an execution engine, it effectively links high-level creative intent with concrete video editing and synthesis tasks, offering a coherent workflow from intention to output.

2. Flexible Multi-Agent Orchestration:
The multi-agent orchestration framework constitutes a strong systems contribution. Its graph-based, self-reflective architecture that dynamically assembles workflows from over thirty specialized agents demonstrates scalability and adaptability. This design is well suited to the non-linear and modular nature of video editing tasks.

3. Benchmark and Evaluation Framework:
The introduction of the VideoEdit benchmark is a valuable contribution to the community, offering a standardized resource for comparison in future work. The empirical validation includes ablation studies and performance analyses that provide credible evidence of the system’s efficiency and effectiveness.

**Weaknesses:**

1. Novelty and Positioning:
The shot-planning module and graph-based orchestration are conceptually related to existing systems such as TeaserGen (Xu et al., 2025) and GPTSwarm (Zhuge et al., 2024). The paper would benefit from clearer articulation of domain-specific innovations tailored to video editing.

2. Evaluation Methodology:
The human evaluation lacks detail on criteria, sample selection, and scoring consistency. The use of a single quality metric limits interpretability, and missing information about excluded baselines and cost calculations weakens transparency.

3. Benchmarking Scope:
The evaluation compares only against general-purpose frameworks. Including domain-specific systems such as ReelDeal or VideoRepurpose would better contextualize performance claims.

**Questions:**

1. How does the proposed system fundamentally differ from existing narration-driven or graph-based orchestration frameworks like TeaserGen or GPTSwarm?

2. Could the authors provide more detail on the human evaluation protocol, including rating criteria and inter-rater agreement?

3. Does the reported cost-efficiency include the entire pipeline or only selected phases?

4. Are there plans to extend evaluation to additional video categories or domain-specific baselines for a fairer comparison?

---

> ### Author Response · Authors · 2025-11-20
>
> We sincerely appreciate your recognition of our work. We will carefully address each question, hoping to alleviate any concerns you may have. Additionally, to provide a more intuitive experience of VideoAgent’s capabilities, we create anonymous accounts on YouTube and Bilibili that contain all of our generated demos. Kindly refer to the links below:
>
> YouTube link: https://www.youtube.com/@xxxx-w3z7x; Bilibili link: https://space.bilibili.com/341654352
>
> **Comment  1 (Weakness1 & Question 1):** Analysis of the differences between existing systems.
>
> **Response**: Thank you for your valuable suggestions. We would like to provide clarifications regarding how our approach distinguishes itself from existing multi-agent systems designed for graph-based workflows.
>
> **i) Domain-specific innovations compared to baseline methods**
>
> Relative to **TeaserGen** (Xu et al., 2025), our key innovations include: First, while TeaserGen focuses specifically on documentary teaser generation using pre-trained highlight detection modules with local learning-to-retrieval, our approach provides an **all-in-one agentic framework** that unifies video understanding and editing capabilities for diverse video creation types rather than being limited to specific domains. Second, we introduce a **global-aware video shot creation algorithm** that perceives complete visual summaries and generates coherent shot storyboards, whereas TeaserGen directly performs visual information retrieval based on user queries, which cannot adequately support long-form video creation demands.
>
> Compared to **GPTSwarm** (Zhuge et al., 2024), our primary advantages are: First, while GPTSwarm relies on reinforcement learning to train agents, which suffers from significant efficiency limitations, we propose a **dynamic graph workflow orchestration algorithm** that employs intent parsing mechanisms to categorize agents by functionality without requiring extensive training. Second, our dual-stage self-reflection mechanism enables multi-round iterative refinement to reduce workflow planning failures, whereas GPTSwarm's approach lacks this sophisticated error correction capability for complex video production workflows.
>
> **ii) Experimental comparison and analysis**
>
> We supplemented our experiments by comparing with TeaserGen for shot retrieval tasks and incorporating both GPTSwarm and GraphCounselor (Gao et al., 2025) for workflow orchestration evaluation. The experimental results demonstrate our method's superiority:
>
> **Table 1: Video understanding and retrieval performance comparison in terms of multiple metrics**
>
> |Method|Recall|EM|IoU|Cost|Time|
> |:-:|:-:|:-:|:-:|:-:|:-:|
> |Claude-Sonnet-3.7|46.03|27.95|23.91|0.374|43s|
> |Gemini-2.5-pro|45.98|27.78|25.91|0.349|42s|
> |VideoRAG|31.03|15.84|14.35|0.100|67s|
> |VideoMind-7B|38.26|27.75|19.67|-|21s|
> |TeaserGen|41.88|27.62|20.91|0.079|25s|
> |Ours-Claude-Sonnet-3.7|44.27|28.18|24.81|0.147|37s|
> |Ours-Gemini-2.5-pro|47.24|28.21|25.74|0.136|37s|
>
> **Table 2: Workflow orchestration of performance comparison in terms of success rate**
>
> |Backbone|Claude-4|Claude-4|Claude-3.7|Claude-3.7|GPT-4o|GPT-4o|Deepseek-v3|Deepseek-v3|
> |:-:|:-:|:-:|:-:|:-:|:-:|:-:|:-:|:-:|
> |Data|Audio|Video|Audio|Video|Audio|Video|Audio|Video|
> |Flow|0.62|0.64|0.84|0.83|0.68|0.60|0.66|0.61|
> |GPTSwarm|0.69|0.83|0.68|0.86|0.64|0.80|0.73|0.81|
> |GraphCounselor|0.83|0.82|0.85|0.81|0.86|0.84|0.82|0.84|
> |**VideoAgent**|**0.93±0.02**|**0.87±0.01**|**0.95±0.03**|**0.93±0.02**|**0.90±0.02**|**0.88±0.01**|**0.92±0.02**|**0.89±0.02**|
>
> The results show that VideoAgent consistently outperforms baseline methods across different backbone models. Specifically, our approach achieves higher recall and IoU scores compared to TeaserGen while maintaining competitive efficiency, and demonstrates substantially higher success rates than both GPTSwarm and GraphCounselor in workflow orchestration tasks.
>
> **iii) Summary of contributions**
>
> To summarize, our methodology innovation and contributions are three-fold: **(1) All-in-one Agentic Framework** that unifies video understanding and editing capabilities within a single system, enabling automated video creation across diverse genres unlike existing domain-specific approaches; **(2) Global-Aware Video Shot Creation Algorithm** that performs shot-level planning through complete visual summary perception and storyboard generation, addressing the limitations of query-based retrieval methods for long-form content creation; and **(3) Dynamic Graph Workflow Orchestration Algorithm** that employs intent parsing and dual-stage self-reflection mechanisms to efficiently coordinate multiple agents without requiring reinforcement learning, thereby overcoming the efficiency limitations of existing training-based approaches.

---

> ### Author Response · Authors · 2025-11-20
>
> **Comment 2 (Weakness 2 & Question 2):** Discussion on the details of human evaluation
>
> We sincerely appreciate your feedback. We would like to share some additional details: We produced 19 demos using VideoAgent, and together with the baseline versions, there are 49 demos in total. They were organized into 18 pairs and evaluated by a total of 26 participants. These participants rated each demo based on three criteria: consistency, audio quality, and scene diversity. The demos were uploaded to a shared drive and organized into a Google Form, with the order randomized and the files anonymized. This evaluation was conducted within our VideoAgent community group, which includes over 200 members from diverse backgrounds such as researchers, content creators, and hobbyists. Collecting all the ratings took about one week, and each participant spent approximately 1 hour to complete the questionnaire. All demos cost less than $0.01 to produce. To ensure the transparency of our human evaluation, we provide here the anonymized links to the demos used in the assessment:
>
> https://drive.google.com/file/d/1liKL6q10E-sCe1XTnNhtv9G4sAff04g_/view?usp=drive_link.
>
> Additionally, we provide the specific scores for the three dimensions. Due to the limitations of the table, the video categories have been omitted here, and the average scores for each dimension were calculated. In the revised version, we will design a chart that includes the scores for all video categories. The results are as follows:
>
> |Method|FunClip|Director|NotebookLM|NarratoAI|Handiwork|VideoAgent|
> |:-:|:-:|:-:|:-:|:-:|:-:|:-:|
> |Consistency|1.9|2.0|1.5|1.1|3.5|3.2|
> |Scene diversity|1.6|1.8|-|1.3|4.2|3.8|
> |Audio quality|1.4|1.2|2.3|2.8|3.9|4.1|
> |Total Score|1.6|1.7|1.9|1.7|3.9|3.7|
>
> VideoAgent closely matches human-produced videos (all videos were selected from platforms with over 100 million users and have more than 10,000 views) in terms of consistency, scene diversity, and audio quality.
>
>
> **Comment 3 (Question 3):** Discussion on the cost-efficiency
>
> The cost-efficiency mentioned in the paper include the entire process, namely workflow orchestration, video understanding, and shot retrieval. In addition, we also gathered statistics from real-world application scenarios: for a 2-hour movie input, producing a 1-minute finished video with VideoAgent takes about 10 minutes and costs approximately $0.01. We hope this helps clarify any confusion you may have.
>
> **Comment 4 (Question 4 & Weakness 3)** Discussion on extending the benchmark
>
> Thank you for your valuable suggestions. We added a new VideoRepurpose (Wu et al., 2025) benchmark and conducted experiments on shot retrieval. 50 videos were randomly sampled from VideoRepurpose, each approximately 1 minute long and containing 1-2 highlight time intervals. The experimental results are as follows:
>
> |Method|Recall|IoU|Method|Recall|IoU|
> |:-:|:-:|:-:|:-:|:-:|:-:|
> |GPT-4o|41.27|19.31|TeaserGen|18.46|16.41|
> |Gemini-2.5-pro|43.12|20.68|Ours-GPT-4o|47.13|19.75|
> |Claude-3.7-sonnet|42.19|17.64|Ours-Gemini-2.5-pro|43.18|19.68|
> |VideoMind-7B|40.11|16.89|Ours-Claude-3.7-sonnet|44.64|19.58|
>
> The results indicate that VideoAgent continues to achieve the best performance, thereby further validating the effectiveness and superiority of the proposed framework.

---

> ### Author Response · Authors · 2025-11-26
>
> Thank you very much for the insightful feedback. We carefully considered and addressed each point. Should any questions remain or additional clarifications be required, please do not hesitate to let us know; we would be pleased to provide further information. We kindly look forward to your response.

---

> ### Author Response · Authors · 2025-11-27
>
> Thank you very much for your valuable comments. We have carefully addressed each of your questions. As the timeline is somewhat tight, please do not hesitate to let us know if you have any further questions or require additional information—we would be happy to assist you at any time. We sincerely look forward to your valuable feedback and greatly appreciate your time and effort.

---

### Author Response · Authors · 2025-11-25

Thank you very much to all the reviewers for your valuable feedback. We thoughtfully considered each of your comments and endeavored to address your concerns to the best of our ability. If you have any additional questions or need further clarification, we would be delighted to assist. We look forward to your continued feedback.

To facilitate a better understanding of our work, here is a brief overview. We develop an **All-in-one Agentic Framework** that seamlessly integrates both video understanding and editing capabilities within a unified system. Unlike prior methods focused on specific domains or video genres, our framework supports diverse video creation tasks under a single architecture, greatly enhancing flexibility and applicability. Second, we introduce a **Global-Aware Video Shot Creation Algorithm** which transcends traditional query-based retrieval by performing shot-level planning informed by holistic visual summaries and storyboard generation. This enables the production of coherent, long-form video narratives that are visually rich and contextually consistent. Third, we design a **Dynamic Graph Workflow Orchestration Algorithm** that dynamically coordinates multiple heterogeneous agents by utilizing intent parsing combined with a dual-stage self-reflection mechanism. This adaptive orchestration approach efficiently manages complex workflows without relying on reinforcement learning and pre-defined graphs, thereby overcoming the efficiency and robustness limitations faced by existing agent systems.

---

### Author Response · Authors · 2025-11-29
**Summary of rebuttal**

Dear Area Chair

**Summary of rebuttal:** We have carefully addressed Reviewer 4’s concerns by enhancing human evaluation details, clearly articulating our paper’s innovations and contributions, and adding system latency and targeted ablation studies. Reviewer 2’s score was very likely to improve if not for the unforeseen circumstance. We have also thoroughly responded to all comments from Reviewers 1 and 3 with additional clarifications and experiments, and are confident this will positively influence their evaluations, despite no further feedback being received.

Before the event, our overall score was 6644, and we believe it would very likely have risen to **6664** had this situation not occurred.

Thank you for your consideration.

Best regards

---

### Meta-Review · Area_Chair_g5eK · 2026-01-04

**Summary:**

The reviewers' concerns primarily focused on the novelty and technical justification of the proposed framework, with Reviewers bw9K, OTRV, and pQcp questioning its differentiation from existing systems like TeaserGen and GPTSwarm, while Reviewer OTRV and Reviewer pQcp raised doubts about the necessity of the complex dynamic graph orchestration and the incremental nature of the algorithmic contributions. Significant attention was also drawn to the evaluation rigor, where Reviewers bw9K and pQcp demanded more detailed human evaluation protocols and visual evidence (video examples), and Reviewers CE6n and pQcp requested comprehensive data on system latency, computational efficiency, and the implications of relying on proprietary APIs. Furthermore, Reviewer bw9K and Reviewer OTRV sought better contextualization through domain-specific baselines and clearer task definitions, which the authors addressed through additional benchmarks and clarifications during the rebuttal process.

**Reviewer Concerns:**

The authors successfully addressed Reviewer pQcp's and Reviewer OTRV's requests for visual evidence by creating anonymous YouTube and Bilibili channels containing nearly 20 demo videos, which significantly improved the reviewers' confidence. Reviewer CE6n's concerns regarding the computational cost and latency of the multi-agent pipeline were effectively addressed by providing detailed tables showing the system can process a 10-minute video in approximately 5 minutes for roughly $0.03, along with breakdown latencies for orchestration.The authors responded to Reviewer bw9K's request for domain-specific baselines by adding the VideoRepurpose benchmark and comparing against TeaserGen and GPTSwarm, demonstrating superior recall and success rates. Through an extensive back-and-forth with Reviewer pQcp, the authors clarified the trade-offs between static and dynamic graphs. While they initially argued strongly for dynamic graphs, they ultimately addressed the concern by integrating six pre-defined static graph templates (e.g., "Meme Video," "Storytelling") into the framework to support standard workflows.

Despite the authors' claims of an "All-in-One" framework, Reviewer OTRV explicitly maintained the critique that the contributions appear to be "incremental modifications to existing models" rather than significant algorithmic innovations. The rebuttal confirms the system relies heavily on prompt engineering and existing APIs rather than novel architectural developments. While the authors provided the details requested by Reviewer bw9K and Reviewer pQcp (26 participants, 49 demos), the methodology remains questionable. The human baseline consisted of "viral videos" from Bilibili  rather than professional edits performed on the same source material. This introduces selection bias, as high view counts do not necessarily equate to the specific editing quality (coherence/narrative) being tested. Reviewer CE6n's concern about reliance on closed-source, proprietary APIs (GPT-4o, Claude)  remains fundamentally unresolved. While the authors tested across multiple models to show robustness, the framework is not a self-contained model that can be fully reproduced or archived independent of commercial service availability.

**Reviewer Scores:**

Reviewer pQcp would most likely increase the score from 4 to 6, as they explicitly stated, "I will raise my rating accordingly" in their final comment. Their initial hesitation stemmed from the lack of visual evidence and a strong skepticism regarding the necessity of "dynamic graphs" versus simpler static ones. The authors successfully resolved these core issues by providing anonymous YouTube/Bilibili links with nearly 20 demo videos and clarifying that the framework supports both dynamic orchestration and pre-defined static templates. This comprehensive response satisfied the reviewer's main doubts about the system's practicality and flexibility.


Reviewer bw9K may choose to maintain the score of 4. While the authors added the requested VideoRepurpose benchmark and human evaluation details, a strict reviewer would likely find these additions methodologically flawed. The "human baseline" relied on viral Bilibili videos rather than controlled professional edits, introducing significant selection bias that undermines the claim of "professional-quality" performance. Furthermore, the core critique regarding novelty that the system is merely a combination of existing tools like TeaserGen was defended but not structurally disproven. Without explicit confirmation of satisfaction, it is highly probable this reviewer will maintain their original score of 4, viewing the rebuttal as volume over substance.


Reviewer CE6n  was already leaning positively with a score of  but expressed valid concerns regarding system latency, computational costs, and reproducibility due to the reliance on proprietary APIs. The authors solidified this positive stance by providing concrete efficiency metrics: a 10-minute video can be processed in approximately 5 minutes for a cost of just $0.039. Furthermore, they included standard deviation measurements across multiple runs to demonstrate reproducibility despite the use of APIs. With these efficiency and stability concerns alleviated, the reviewer is likely to maintain their "Marginally above acceptance" rating.

Reviewer OTRV represents the highest risk of maintaining a low score of 4. While the authors addressed the specific missing technical details regarding face-swapping and lip-sync agents, the reviewer remained unconvinced about the fundamental novelty of the work. In their final comments, OTRV commented that the contributions appeared to be "incremental modifications to existing models" rather than significant algorithmic innovations. Unlike the other reviewers who expressed satisfaction, OTRV's core critique regarding the "incremental" nature of the research was not fully resolved by the rebuttal, suggesting their score may remain at 4.

---

### Decision · Program_Chairs · 2026-01-26

Reject